# Gut micro-organisms associated with health, nutrition and dietary interventions

Francesco Asnicar[1✉], Paolo Manghi[1,6], Gloria Fackelmann[1], Gabriel Baldanzi[1], Elco Bakker[2], Liviana Ricci[1], Gianmarco Piccinno[1], Elisa Piperni[1,3], Katarina Mladenovic[1], Federica Amati[2,4], Alberto Arrè[2], Sajaysurya Ganesh[2], Francesca Giordano[2], Richard Davies[2], Jonathan Wolf[2], Kate M. Bermingham[2,5], Sarah E. Berry[2,5,7], Tim D. Spector[2,4,7] & Nicola Segata[1,3,4,7✉]

The incidence of cardiometabolic diseases is increasing globally, and both poor diet and the human gut microbiome have been implicated[1]. However, the field lacks large-scale, comprehensive studies exploring these links in diverse populations[2]. Here, in over 34,000 US and UK participants with metagenomic, diet, anthropometric and host health data, we identified known and yet-to-be-cultured gut microbiome species associated significantly with different diets and risk factors. We developed a ranking of species most favourably and unfavourably associated with human health markers, called the 'ZOE Microbiome Health Ranking 2025'. This system showed strong and reproducible associations between the ranking of microbial species and both body mass index and host disease conditions on more than 7,800 additional public samples. In an additional 746 people from two dietary interventional clinical trials, favourably ranked species increased in abundance and prevalence, and unfavourably ranked species reduced over time. In conclusion, these analyses provide strong support for the association of both diet and microbiome with health markers, and the summary system can be used to inform the basis for future causal and mechanistic studies. It should be emphasized, however, that causal inference is not possible without prospective cohort studies and interventional clinical trials.

Cardiometabolic diseases (CMDs) are the leading causes of morbidity and mortality in Western countries and constitute a heavy burden on global healthcare systems[1]. The most predominant CMDs are cardio-vascular disease (CVD) and type 2 diabetes (T2D)[3], which are connected with the increased consumption of calorie-dense, high-risk processed foods observed over the past few decades[4]. Habitual diet is not only among the known risk factors associated with CMDs, but also the primary modifiable target for prevention and treatment[5]. Well established anthropometric and intermediary measures of CMDs, ranging from clinical measurements (for example, blood pressure) to lipid profiles (such as triglycerides, cholesterol and lipoproteins), glucose levels (for example, fasting and postprandial glucose, and haemoglobin A1c (HbA1c)), inflammatory markers (for example, glycosylated proteins, the systemic inflammation marker GlycA[21] and high-sensitivity C-reactive protein), and known risk factors such as body mass index (BMI), can be used to study the diet–CMD axis[6–8] but do not consider the biochemical mechanisms occurring in the human gut.

The human gut microbiome has emerged as a cofactor on the same axis as it is associated with diet and cardiometabolic conditions[9–12] and is a modifiable element[13–15]. A change in dietary patterns can shift the species-level composition of the microbiome, with knock-on effects on host health[16]. However, individual responses to dietary interventions vary, and precision nutrition aims at identifying host-specific factors that modulate the interaction between diet and host health[17], but it is currently not possible to disentangle the effects diet plays to improve cardiometabolic health via the microbiome. Furthermore, the composition of the gut microbiome displays high individuality and variation depending on different demographics, ethnicity, sex and age; hence, defining or identifying universal biomarkers of a healthy gut microbiome has proven difficult[18–20].

Nutritional intervention studies usually involve low sample-size cohorts at the population level and are often limited by their statistical power and specificity to local lifestyle and dietary habits, which are all critical aspects, especially given the microbiome's complexity and variability. Large-scale comprehensive studies with multi-national populations can help disentangle some of the complex interplays between dietary patterns and the gut microbiome to develop personalized interventions to prevent and treat CMDs. Accordingly, we collated, sampled and analysed five of the largest metagenomic cohorts available to date, comprising more than 34,000 people and spanning two continents, paired with dietary data, detailed anthropometric and health markers. We identified microbiome species consistently associated with more favourable and (inversely) unfavourable health markers across continents. These species were organized into two microbiome rankings, representing host health and diet quality, respectively, that can be the basis for future causal and mechanistic studies.

[1]Department CIBIO, University of Trento, Trento, Italy. [2]Zoe Ltd, London, UK. [3]IEO, Istituto Europeo di Oncologia IRCSS, Milan, Italy. [4]Department of Twins Research and Genetic Epidemiology, King's College London, London, UK. [5]Department of Nutritional Sciences, King's College London, London, UK. [6]Present address: Research and Innovation Center, Fondazione Edmund Mach, San Michele all'Adige, Italy. [7]These authors contributed equally: Sarah E. Berry, Tim D. Spector, Nicola Segata. ✉e-mail: f.asnicar@unitn.it; nicola.segata@unitn.it

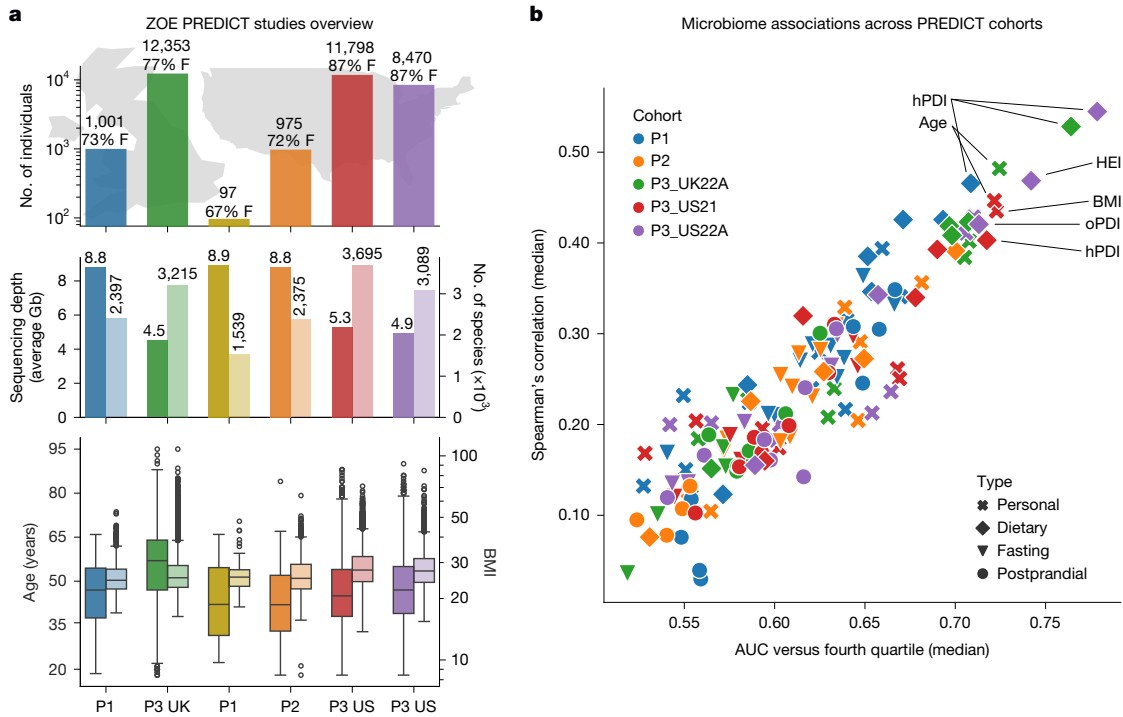

**Fig. 1 | The ZOE PREDICT studies comprise over 34,000 healthy people from five cross-sectional studies from the UK and the USA with gut microbiome samples, detailed individual information and dietary habits. a**, In this study, we considered and harmonized five cross-sectional ZOE PREDICT cohorts with participants from the UK and the USA (Supplementary Fig. 1). For each cohort, sample size and the percentage of female participants (% F) are reported in the upper bar plots, with sequencing depth (left-hand columns, darker colour, average size in gigabases) and the total number of detected species (right-hand columns, lighter colour) are reported in the middle bar plots, showing that cohorts with lower sequencing depths do not have fewer total numbers of detected species. Bottom box plots, distributions of age (left-hand columns, darker colour) and BMI (right-hand columns, lighter colour) in the five PREDICT

cohorts (the PREDICT 1 (P1) cohort is split into its UK and US parts, but considered as a single cohort). Box plots show first and third quartiles (boxes) and the median (middle line); whiskers extend up to 1.5 × interquartile range (IQR). **b**, Random forest classification (discriminating the first three quartiles against the fourth quartile) and regression machine learning models (Methods) trained on the whole microbiome SGB-level relative abundance values with a cross-validation approach, show moderately strong and consistent associations with different categories of clinical data available across the five cross-sectional ZOE PREDICT cohorts (full machine learning results are reported in Extended Data Fig. 1 and Supplementary Tables 2 and 3). HEI, healthy eating index; PDI, plant-based diet index; hPDI, healthful PDI; oPDI, overall PDI.

## Metagenomics of the ZOE PREDICT cohorts

We used four large-scale microbiome cohorts from the ZOE PREDICT studies ($n = 33,596$; Fig. 1a, Supplementary Fig. 1, Supplementary Table 1 and Methods) to assemble an extensive microbiome dataset of people with detailed dietary records along with anthropometric measures. Together with the previously available ZOE PREDICT 1 cohort[9] ($n = 1,098$), the PREDICT cohorts comprise 34,694 participants from both the USA ($n = 21,340$) and the UK ($n = 13,354$; Methods). Collected data comprise common health risk factors such as BMI, triglycerides, blood glucose and HbA1c, as well as several dietary indices and clinical markers that are intermediary measures of cardiometabolic health, such as the atherosclerotic CVD (ASCVD) risk, high-density lipoproteins (HDL) and GlycA[21] (Supplementary Table 2 and Methods).

A systematic machine learning validated approach[9,22] (Methods) revealed strong associations consistent across the five ZOE PREDICT cohorts between the microbiome and surrogate health markers and nutrition (Fig. 1b, Supplementary Table 2 and Methods). Markers that were classified accurately by the gut microbiome included glycemia, blood cholesterol, triglycerides and inflammation (both fasting and postprandial; Extended Data Fig. 1), with age, BMI, the healthy eating index[23] and the healthful plant-based diet index (PDI)[24] also correlated with microbiome machine learning regression estimates (Spearman's correlation > 0.4; Fig. 1b and Supplementary Table 2). The top predicted markers from both machine learning regression and classification showed consistent associations across PREDICT cohorts, with average

area under the receiver operating characteristic curve (AUC) ranging from 0.64 to 0.73, and an average Spearman's correlation ranging from 0.30 to 0.46 for regression (Fig. 1b and Supplementary Table 3).

## Ranking gut species to host and health

We next set out to identify which gut microbial species were most responsible for the microbiome's associations with host markers. To do so, we grouped the 37 markers into three categories: (1) anthropometric-derived and accessible health-related measures (hereafter called 'personal' and including, for example, ASCVD and blood pressure), (2) fasting (for example, GlycA, triglycerides, HDL, cholesterol and glucose) and (3) postprandial markers, which are surrogate measures of cardiometabolic health. As expected, some markers tended to correlate quantitatively (Supplementary Table 4 and Methods).

We considered 661 non-rare microbial species (greater than 20% prevalence; Methods) according to the definition of species-level genome bins (SGBs)[19,20], and computed the partial Spearman's correlations (corrected for sex, age and BMI) between the relative abundance of each micro-organism and the value of each marker. Correlations were ranked, and correlations' ranks were averaged within each category and then averaged among the three categories in each cohort (Methods). The five resulting cohort-level average rankings were averaged to derive a single ranking that we called the 'ZOE Microbiome Health Ranking 2025' (ZOE MB health-rank). This resulted in a ranking for 661 microbial species in which the lowest ranking (closer to 0) species are the most

positively associated with the considered panel of host markers and vice versa for the highest ranking (closer to 1) species (Fig. 2, Extended Data Figs. 2 and 3, Supplementary Fig. 2 and Supplementary Tables 5 and 6).

Most SGBs ranked within the 50 most favourably or unfavourably linked to host anthropometry belong to the Firmicutes phylum (92 out of 100) and, in particular, to the Clostridia class ($n = 80$; Supplementary Table 7). Within this class, in the ZOE MB health-ranks, most SGBs belonged to the Clostridiales order, with 32 unfavourably ranked SGBs (of which $n = 27$ *Lachnospiraceae* out of 50) and 31 favourably ranked SGBs ($n = 13$ *Lachnospiraceae* and $n = 12$ *Ruminococcaceae*) assigned to this order. Collectively, the average total relative abundance of the 50 most favourably ranked SGBs is 5.98%, whereas the 50 most unfavourably ranked SGBs account for 13.64% (Supplementary Table 7).

## Uncharacterized health-linked bacteria

A large portion of the 50 most favourably ZOE MB health-ranked SGBs are unknown ($n = 22$), meaning that these are uncultured species represented solely by microbial genomes reconstructed from metagenomic data. Of the 28 known SGBs (with available isolate genomes), 24 are still uncharacterized species without phenotypic descriptions and recognized taxonomic names (Supplementary Table 7). *Eubacterium siraeum* (SGB4198) and *Faecalibacterium prausnitzii* (SGB15317) are among the few exceptions with previous support for their favourable role[9,25].

By contrast, the 50 unfavourably ZOE MB health-ranked SGBs are generally species with cultured isolates and established taxonomic labels (Supplementary Table 7). Among the 44 known SGBs, several species were already linked with detrimental effects on the host, including *Ruminococcus gnavus*[26], *Flavonifractor plautii*[27], *Ruminococcus torques*[28,29] and *Enterocloster bolteae*[30]. Overall, the most prevalent favourably ranked health-associated micro-organisms in the human gut belong to under-investigated species, highlighting gaps in our knowledge of the potential beneficial role of the human microbiome in promoting and maintaining non-pathogenic conditions.

## Gut species ranked by diet quality

Similarly to the ZOE MB health-ranks, we defined a species ranking on the basis only of dietary markers across all five PREDICT cohorts, which we called the 'ZOE Microbiome Diet Ranking 2025' (ZOE MB diet-rank; Supplementary Table 5). As markers of a generally healthier diet, we adopted five validated indices (Methods) computed starting from validated food frequency questionnaires (FFQs) or logged diet data (logged using a mobile phone app), reflecting long- and short-term dietary habits, respectively (Extended Data Figs. 4 and 5 and Methods).

The ZOE MB health- and diet-rankings showed, as expected, general concordance (Spearman's $\rho = 0.72$; Extended Data Fig. 6a and Supplementary Table 5). Although the large majority of the SGBs highlighted by high or low ZOE MB health-ranks and diet-ranks belong to unknown taxa, reported phenotypic characteristics of known species agree with our analysis. For example, *R. torques* (SGB4608) and *F. plautii* (SGB15132), discussed previously as unfavourable species according to the ZOE MB health-ranks, were also concordantly unfavourably ranked in the ZOE MB diet-ranks (0.991–0.904 and 0.981–0.901, respectively). On the other hand, the favourably ranked *Blautia glucerasea* (SGB4816) was described to reduce visceral fat accumulation, blood glucose and triglycerides in mice[31] (ZOE MB health-ranks and diet-ranks of 0.267 and 0.062, respectively). As another example, in a dietary fibre supplementation trial involving individuals with T2D, *Lachnospira eligens* (SGB5082) was increased selectively and associated negatively with postprandial glucose and insulin, body weight and waist circumference[32] (ZOE MB health-ranks and diet-ranks of 0.276 and 0.115, respectively), indicating that precise dietary interventions aimed at stimulating beneficial bacterial growth can contribute to treating or managing metabolic disorders symptoms.

Despite the overall agreement between the ZOE MB health- and diet-rankings, 65 out of the 661 ranked SGBs showed discordant rankings (absolute rank difference at least 0.3; Extended Data Fig. 6a and Supplementary Table 8). Generally, the different trends may be due to the different capacities of certain bacteria (for example, generalists) to use a variety of substrates, including those derived from unhealthy diets, while releasing functional metabolites with protective or health-promoting effects. Among these, for example, *Harryflintia acetispora* (SGB14838) was found associated with favourable cardio-metabolic markers and unfavourable diets (ZOE MB health-rank = 0.363 and ZOE MB diet-rank = 0.879) in this study. This strict anaerobe can use readily available monosaccharides such as maltose, glucose and fructose, but can also produce short-chain fatty acids[33], which are regulatory and anti-inflammatory mediators[34].

Across the US and UK populations, the ZOE MB health-rankings showed high consistency (Spearman's $\rho = 0.61$; Extended Data Fig. 6b), whereas country-specific ZOE MB diet-rankings were more heterogeneous (Spearman's $\rho = 0.26$; Extended Data Fig. 6c). The intraclass correlation coefficients (ICC)[35] also suggest that the ZOE MB health-ranks are more consistent across countries than the ZOE MB diet-ranks (ICC = 0.5929 and 0.2623, respectively; Extended Data Fig. 6b,c). Across cohorts, we obtained an ICC = 0.63 and 0.46 for the ZOE MB health-ranks and diet-ranks, respectively, indicating that health rankings were more able to capture cohorts and countries differences, whereas the most favourably ranked species appeared to match across populations with similar levels of industrialization and lifestyle.

## Species rankings stratify by BMI

BMI is an imperfect but widely adopted and easy-to-obtain anthropometric marker of health risk. As BMI was not included among the markers of the ZOE MB health- and diet-rankings, and we corrected for it in the partial correlation analysis, we set out to evaluate how the two rankings can stratify people according to their BMI to assess how health signatures in the gut microbiome are reflected in body mass.

We correlated the 661 ZOE MB health-ranked species with BMI (corrected for sex and age), in each PREDICT cohort, and found that, overall, the ranks were associated positively with BMI (Spearman's $\rho = 0.72$), with the favourably ranked SGBs correlated negatively with BMI, whereas unfavourably ranked SGBs correlated positively with BMI (Fig. 3a). These results were confirmed when considering the ZOE MB diet-ranks and discrete BMI categories (Extended Data Fig. 7a–c; all intra-dataset comparisons statistically significant at $Q < 0.2$ and all 30 except 7 at $Q < 0.01$) as well as the cumulative abundance of the species in the two 50-species sets (Fig. 3b,c; all intra-dataset comparisons statistically significant at $Q < 0.2$ and all 30 except 5 at $Q < 0.01$).

To generalize these associations, we leveraged a total of 5,348 healthy individuals from 27 public cohorts divided into three BMI categories, healthy weight ($n = 2,837$), overweight ($n = 1,562$) and obese ($n = 949$; Supplementary Table 9 and Methods). In 47 pairwise comparisons, 34 had a higher median richness for the 50 most favourably ranked ZOE MB health SGBs in lower BMI groups versus higher BMI groups (binomial $P = 0.003$; Supplementary Table 10 and Supplementary Fig. 3a), and this was not dependent on country effects or sequencing depth (Supplementary Table 11), highlighting the generalization of the identified ranks. Meta-analysis based on linear regression on single cohorts (Methods) showed that individuals with healthy weight carried, on average, 5.2 more of the 50 favourably ZOE MB health-ranked SGBs than people with obesity ($P = 0.0003$; Fig. 3d and Supplementary Table 12), which corresponded to a normalized difference in the cumulative abundances of unfavourably and favourably ranked SGBs of Cohen's $d = -0.59$ ($P < 0.0001$; Supplementary Tables 10 and 13 and Methods). Correspondingly, individuals with obesity carried, on average, 1.95 more of the unfavourably ranked SGBs than people of healthy weight ($P = 0.0005$; Fig. 3d, Supplementary Tables 14 and 15; Cohen's $d$

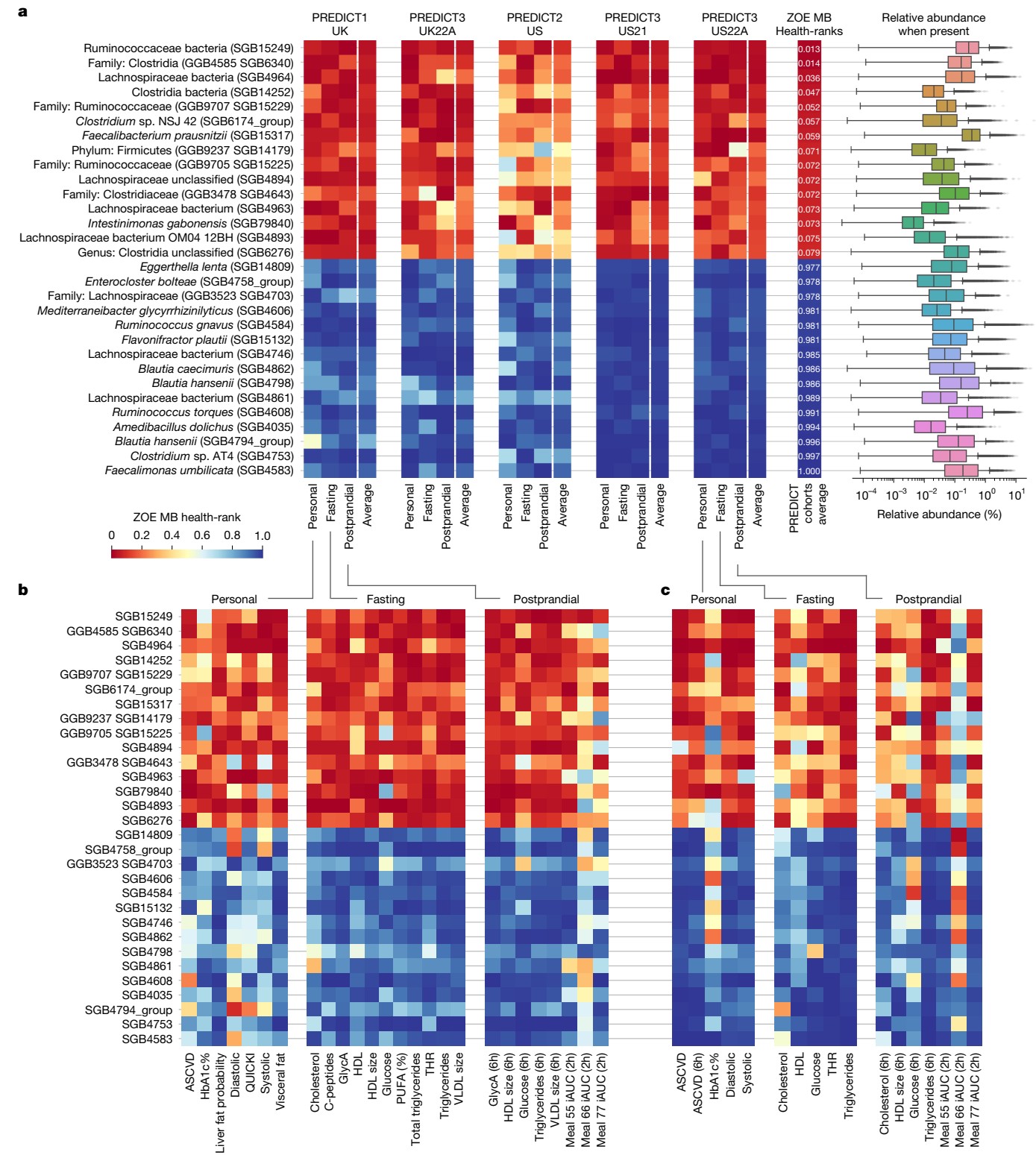

**Fig. 2 | The 15 top and bottom health-ranked SGBs show consistent associations across the five PREDICT cohorts. a**, Average percentiles for the 15 most favourably (top) and unfavourably (bottom) ranked SGBs (selected for visualization purposes) across all five PREDICT cohorts. Percentiles are computed from the ranking of the correlations between SGBs and the different markers in each clinical data category. Percentiles close to 0 reflect SGBs consistently correlated positively with positive markers and negatively with negative markers, and vice versa for percentiles close to 1. For each cohort, the average percentiles for three clinical data categories are shown (personal, fasting and postprandial) and the cohort-level average. The rightmost column of the heatmap reports the ZOE MB health-ranks with the distribution of their relative abundance values when present (derived from $n$ = 34,694 participants spanning the five PREDICT cohorts). Box plots as in Fig. 1. **b,c**, Detailed percentiles for the 15 most favourably and unfavourably ranked SGBs against the markers of the three clinical data categories of the PREDICT 1 (UK) (**b**) and PREDICT 3 US22A (US) (**c**) cohorts. Detailed panels of the percentiles for the other three cohorts can be found in Extended Data Fig. 2. iAUC, incremental area under the curve; PUFA, polyunsaturated fatty acid; QUICKI, quantitative insulin sensitivity check index; THR, total-cholesterol-to-HDL ratio; VLDL, very-low-density lipoprotein.

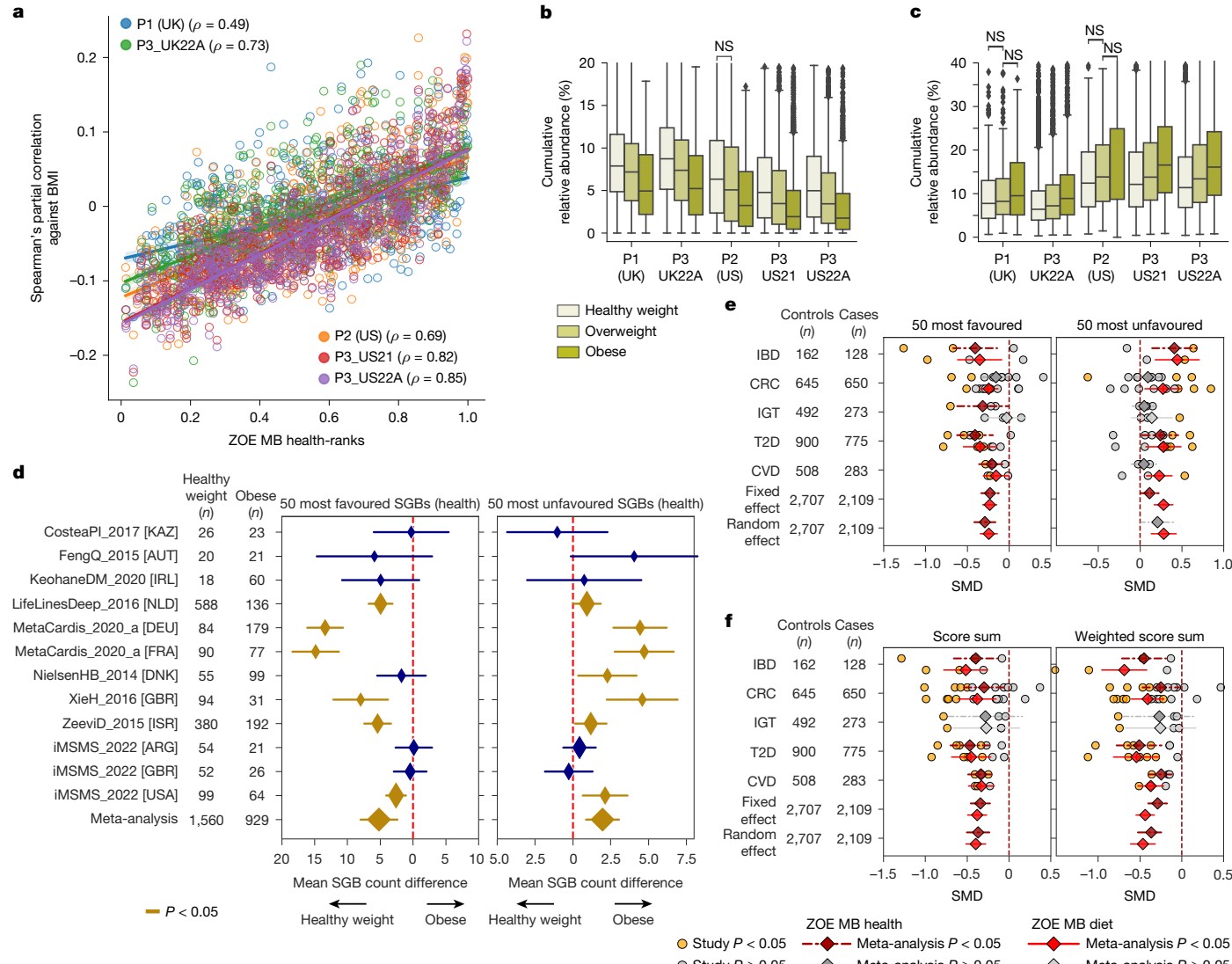

**Fig. 3 | ZOE MB health- and diet-ranked species show significant and reproducible associations with BMI and diseases. a**, Concordance of ZOE MB health-ranks with partial Spearman's correlations against BMI (corrected for sex and age) across PREDICT cohorts. Favourably ranked SGBs correlate negatively with BMI; unfavourably ranked SGBs correlate positively (ZOE MB diet-ranks in Extended Data Fig. 7a). Shading represents 95% confidence interval. **b,c**, Cumulative relative abundance of favourably (**b**) and unfavourably (**c**) ranked SGBs across BMI categories. As BMI increases, reflecting higher health risks, the abundance of favourable SGBs decreases whereas that of unfavourable SGBs increases. Similar patterns were seen for SGB richness (Extended Data Fig. 7b,c). Only non-significant (NS) false discovery rate (FDR)-corrected $P$ values ($Q > 0.01$, two-sided Mann–Whitney $U$-test) are annotated. Box plots as in Fig. 1. **d**, Meta-analysis of the 50 most favourable and unfavourable SGBs comparing participants of healthy weight with those with obesity from public cohorts. Lower BMI is associated with more favourable SGBs; people with higher BMI

carry more unfavourable SGBs. Meta-analysis on ranks defined on UK and US participants shows reproducibility across countries. Other comparisons are in Extended Data Fig. 8 and the diet-ranked SGBs meta-analysis in Extended Data Fig. 9. Country codes: ARG, Argentina; AUT, Austria; DEU, Germany; DNK, Denmark; FRA, France; GBR, United Kingdom of Great Britain and Northern Ireland; IRL, Ireland; ISR, Israel; KAZ, Kazakhstan; NLD, Netherlands; USA, United States of America. **e**, Meta-analysis of disease group (adjusted by sex, age and BMI) on standardized mean differences (SMD) of cumulative relative abundance of the 50 most favourable (left) and unfavourable (right) SGBs from both rankings (meta-analysis on SGB richness in Supplementary Fig. 5a; Methods). **f**, Meta-analysis of normalized ZOE MB health-ranks and diet-ranks, weighted by arcsin square-root of relative abundance values (right, weighted score sum; left, score sum (unweighted)). Horizontal lines in meta-analysis plots represent 95% confidence intervals. CRC, colorectal cancer; IBD, inflammatory bowel disease; IGT, impaired glucose tolerance.

on cumulative relative abundances = 0.29; $P = 0.0001$). Pairwise analysis of the other BMI categories confirmed these results (Extended Data Fig. 8 and Supplementary Tables 10 and 12–15).

Similarly, we tested the association of the 50 most favourably and unfavourably ZOE MB diet-ranked SGBs with BMI, and found similar but milder signals compared with the ZOE MB health-ranks (average Spearman's correlations between the two ranks and BMI of 0.61 and 0.72, respectively; Fig. 3a and Extended Data Fig. 7a). Using public datasets, 36 intra-dataset comparisons out of 47 showed a higher median cumulative abundance and a higher median richness of the

50 most favourable SGBs in lower BMI classes compared with higher BMIs (binomial $P = 0.0003$; Supplementary Fig. 3b). Conversely, 36 comparisons showed a higher median count of the least favourable 50 SGBs for the higher BMI classes compared with the lower BMI groups (binomial $P = 0.0003$; Supplementary Table 10). The contribution of diet-ranked SGBs in different BMI categories similarly showed a decreasing number and cumulative relative abundance of favourably ranked SGBs and an increase in unfavourably ranked SGBs (Extended Data Fig. 7d–g). In meta-analysis, healthy weight and overweight participants carried 3.5 and 1.5 more favourable diet-ranked SGBs, and 1.25

and 0.88 fewer unfavourable ZOE MB diet-ranked SGBs than obesity participants, respectively (Extended Data Fig. 9, Supplementary Fig. 3 and Supplementary Tables 16–19). All these analyses were confirmed when rankings were computed without adjusting for BMI (Extended Data Fig. 7h–k) and, altogether, these results suggest that the ZOE MB health- and diet-ranks can stratify people based on their obesity status regardless of geography.

## Species rankings and host diseases

Next, we assessed whether the ZOE MB health-ranked SGBs had a differential presence or abundance in control participants compared with participants with a defined disease condition, exploiting 25 case–control, publicly available microbiome studies (4,816 samples in total with $n = 2,707$ controls and $n = 2,109$ cases; Methods) investigating five diseases with variable levels of association with the gut microbiome (Supplementary Table 20). The number of the 50 most favourably ZOE MB health-ranked SGBs was higher in controls than cases for 21 of the 25 cohorts, whereas the count of the 50 most unfavourably ranked SGBs was correspondingly higher in cases for the same number of cohorts (binomial $P = 0.0004$).

We performed a meta-analysis on the count of the 50 most favourable and unfavourable SGBs from the ZOE MB health- and diet-rankings. Control samples carried, on average, 3.6 more favourably ranked SGBs than participants with disease (random-effect model, $P = 0.0002$; Methods) and 1.6 fewer unfavourable SGBs ($P = 0.0004$; Supplementary Fig. 5a and Supplementary Table 21). Similarly, for the ZOE MB diet-ranked SGBs, controls carried, on average, 3.8 more favourable SGBs and 1.3 fewer unfavourable SGBs, $P = 9.5 \times 10^{-6}$ and $P = 0.0006$, respectively; Supplementary Fig. 5a and Supplementary Table 21). Furthermore, meta-analyses of the cumulative abundance of the 50 most favourable and unfavourable SGBs confirmed a greater contribution from favourable species in control groups and of unfavourable SGBs in the corresponding disease groups (meta-analysis Cohen's $d = -0.29$, $P = 7.1 \times 10^{-6}$ and $d = 0.21$, $P = 0.054$ for the ZOE MB health-ranks; $d = -0.24$, $P = 3.1 \times 10^{-6}$ and $d = 0.28$, $P = 0.0002$ for the ZOE MB diet-ranks; Fig. 3e and Supplementary Table 22).

To assess how informative the rankings are in summarizing the health-associated status of a single sample, we scored all metagenomes from diseased and control participants by summing the normalized ZOE MB health-ranks of the SGBs present in the sample (Methods). We found a strong separation between diseased and control participants (meta-analysis Cohen's $d = -0.37$, $P = 8.3 \times 10^{-8}$), improving over the simple counting of the number of most favourable and unfavourable SGBs (Fig. 3f). Notably, T2D showed the strongest disease-specific association (meta-analysis Cohen's $d = -0.47$, $P = 6.78 \times 10^{-5}$; Fig. 3f and Supplementary Table 23) with the weighted version of this score showing an even stronger effect for T2D (meta-analysis Cohen's $d = -0.51$, $P = 0.0002$). People were also scored using the ZOE MB diet-ranks, and similar links with their health status emerged (Fig. 3f and Supplementary Table 23). Notably, standard alpha diversity measures such as gut SGBs richness and Shannon's entropy measures showed weaker and less consistent associations, with significant links only in the IBD and T2D comparisons (Supplementary Fig. 5b and Supplementary Table 24).

Although the ranking-based scoring of single samples cannot have the same predictive power for host phenotypes compared with condition-specific supervised learning approaches relying directly on labelled training data, our results showed how embedding the ranking system into a simple one-dimensional microbiome index provides a meaningful evaluation of microbiome health conditions.

## Diet changes effects on ranked species

To validate the effect of dietary changes on the presence and abundance of gut microbial species according to their ZOE MB health-rankings,

we analysed two dietary intervention studies, namely ZOE METHOD[36] and BIOME[37] (ClinicalTrials.gov registrations, NCT05273268 and NCT06231706, respectively). In brief, the ZOE METHOD cohort comprised $n = 347$ people assigned to a personalized dietary intervention programme (PDP; $n = 177$) arm versus an arm with general diet advice following the US Department of Agriculture recommendations (control, $n = 170$). People assigned to the PDP group showed lower energy intake and a significant decrease in triglycerides, HbA1c, weight and waist circumference after 18 weeks[36]. The ZOE BIOME cohort comprised $n = 349$ healthy adults (intention-to-treat) randomized into the primary intervention group (receiving a defined prebiotic blend, $n = 116$), the functional control group (receiving bread croutons to match the calories in the control group, $n = 120$) and the daily probiotic group (supplemented with 15 billion colony-forming units of *Lacticaseibacillus rhamnosus* per day, $n = 113$). Overall, weight, waist circumference, metabolites and gastrointestinal symptoms did not differ significantly between groups[37].

We identified which microbiome species were impacted significantly by the dietary interventions in the two cohorts. In the ZOE BIOME cohort, 57, 4 and 14 prevalent SGBs showed significant changes at the endpoint ($Q < 0.01$) for the prebiotic blend, probiotic and control arm, respectively (Fig. 4a). Among the species with a significant change in the prebiotic arm were beneficial fibre-degrading *Bifidobacterium adolescentis* (SGB17244), *Bifidobacterium longum* (SGB17248) and *Blautia obeum* (SGB4811)[38–40], as well as butyrate-producing *Agathobaculum butyriciproducens* (SGB14993), *Anaerobutyricum hallii* (SGB4532) and *Coprococcus catus* (SGB4670)[41,42]. By contrast, the species *Dysosmobacter welbionis* (SGB15078), among the top unfavourably associated SGBs in our study, was decreased significantly by the same dietary intervention (Supplementary Table 25). In the ZOE METHOD cohort, we found 46 SGBs differed significantly in their relative abundance in the PDP arm, and only two in the control arm (Fig. 4b and Supplementary Table 25; Wilcoxon signed-rank test $Q < 0.1$). Of note, the prominent butyrate producers *Roseburia hominis* (SGB4936) and *A. butyriciproducens* (SGB14993) were also found to increase in the PDP intervention.

The dietary intervention groups of both clinical trials that aimed at improving diet using different approaches (prebiotic blend for BIOME and PDP for METHOD) showed the highest number of significantly changing SGBs (Fig. 4a and Supplementary Table 25). Focusing on the most significant gut microbial SGBs with largest change in relative abundance after dietary interventions, we found increasing *Bifidobacterium animalis* (SGB17278)—a bacterium present in dairy-based foods and in the microbiome of people consuming larger amounts of them[43,44] (Fig. 5a,b and Supplementary Table 25), an unknown Lachnospiraceae bacterium (SGB4953, BIOME; Fig. 5a) and *R. hominis* (SGB4936, METHOD; Fig. 5b) both previously associated with a vegan diet[43], and another unknown Lachnospiraceae bacterium (SGB5200, BIOME; Fig. 5a) linked to a vegetarian diet[43]. *Butyricimonas paravirosa* (SGB1785, METHOD; Fig. 5b), *Phocea massiliensis* (SGB14837), a currently uncharacterized Ruminococcaceae species (SGB14899) and *Candidatus* Pararuminococcus gallinarum (SGB63327) (all found in the BIOME cohort; Fig. 5a), were instead decreasing in the intervention and reported to be associated with a mixed diet[43]. The *Streptococcus salivarius* (SGB8007, METHOD; Fig. 5b) species was also found in food microbiomes[45] and, together with an unknown Ruminococcaceae species (SGB14899, BIOME; Fig. 5a), were found associated with non-vegans[43].

We found that the SGBs with increased relative abundance at endpoint in the prebiotic blend arm of the BIOME trial (Fig. 4a and Supplementary Table 25) showed significantly more favourable ZOE MB health-ranks and diet-ranks than the decreasing SGBs (Fig. 5c,d, Mann–Whitney $U$-test $P = 7.78 \times 10^{-3}$ and $P = 3.00 \times 10^{-5}$, respectively). This was confirmed for the PDP arm of the METHOD cohort (Fig. 5e,f; Mann–Whitney $U$-test $P = 5.20 \times 10^{-5}$ and $P = 2.03 \times 10^{-3}$, respectively). No significant enrichment for ZOE MB health- and diet-rankings was instead detected for the significantly changing SGBs for the probiotic

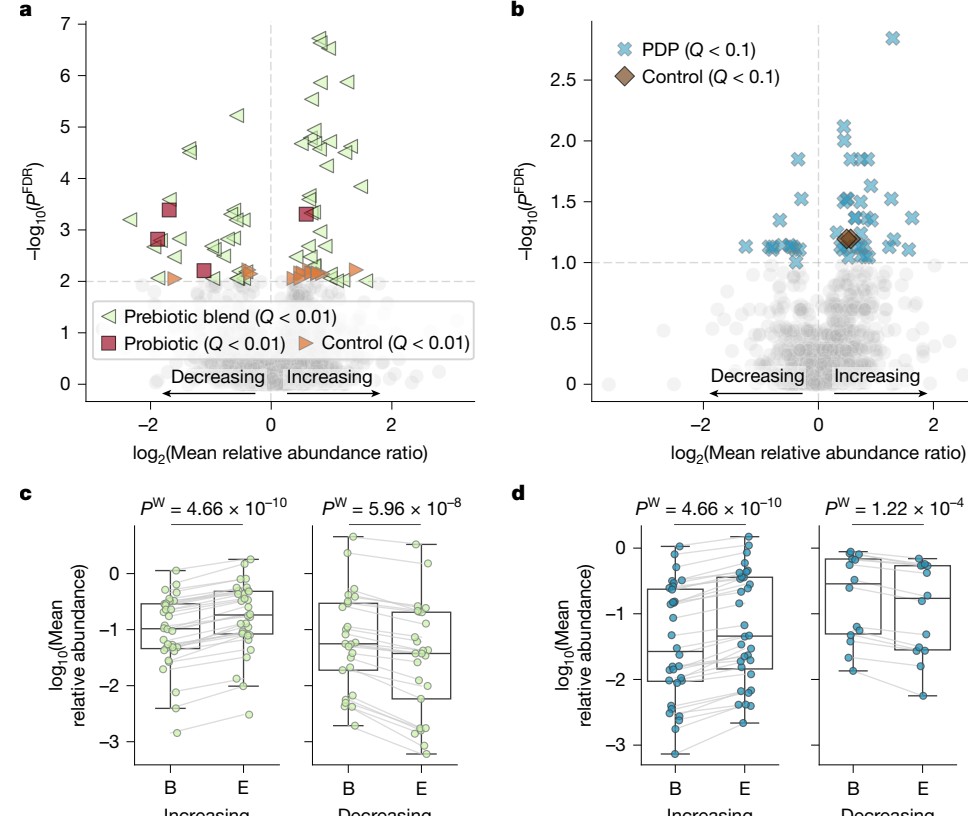

**Fig. 4 | Dietary interventions have a large impact on microbiome composition. a,b,** Pre–post dietary intervention variations in prevalent gut microbial SGBs (at least 10% at both time points). The plots show the effect size (log₂-transformed ratio of mean relative SGB abundance at endpoint over baseline) against the significance (*Q* values, FDR–Benjamini–Hochberg-corrected *P* values). **a,** BIOME cohort (ClinicalTrials.gov NCT06231706) with *n* = 321 healthy adults from the UK (*n* = 106 prebiotic blend, *n* = 106 probiotic and *n* = 109 control), significance threshold set to *Q* < 0.01. **b,** METHOD cohort (ClinicalTrials.gov NCT05273268)

with *n* = 347 US individuals (*n* = 177 PDP, *n* = 170 control), and significance threshold set to *Q* < 0.1. **c,** Change in relative abundance for the significant SGBs in the intervention arms of BIOME (prebiotic blend, *n* = 57). **d,** Change in relative abundance of METHOD (PDP, *n* = 46), separated into those that increase from those that decrease from baseline (B) to endpoint (E). Extended Data Fig. 10a–d reports the change in relative abundance and prevalence of the control and prebiotic arms. Two-sided Wilcoxon test; box plots as in Fig. 1).

group of the BIOME cohort or the control groups of both BIOME and METHOD cohorts (Extended Data Fig. 10e,f).

Together, these results show how dietary interventions or tailored prebiotic blends, both aiming at improving diet quality, positively modulate the microbiome composition. The SGBs' rankings (ZOE MB health and diet), which were defined on cross-sectional independent cohorts, were strongly and consistently predictive of the SGBs most associated with the dietary interventions in independent cohorts and countries, supporting the direct, reproducible and actionable link between diet and microbiome composition.

## Conclusions

Defining the baseline composition of the human gut microbiome in 'healthy' host conditions has been a long-standing challenge. This area has several problems, including defining general host health across age, as well as the inter-population microbiome variability, the existence of several distinct health-associated microbiome configurations[19,46], the personalized nature of diet's impact on the gut microbiome[16,47], the diversity of dietary regimes[48,49] and the effect of social interaction on microbiome transmission[44]. To address this challenge, we reformulated the question of what a health-associated microbiome is by scoring gut microbiome species for their tendency to be correlated with healthy diet scores and with the continuum of a panel of intermediate markers of cardiometabolic health, in large and generally healthy populations. By leveraging diet scores such as the healthy eating index or

the healthful PDI, and health estimators such as blood glucose, HDL and triglycerides, we identified species that are expected to characterize hosts in healthier conditions, as well as other species that are enriched in hosts with more unfavourable health risk factors. Most of the key health-associated species were from previously uncharacterized species, underlining the wide knowledge gap of the microbiome composition in non-diseased conditions. These rankings, named ZOE MB health-ranks and diet-ranks, are released and maintained publicly (Supplementary Table 5 and 'Data availability' section) and can be adopted by the research community to evaluate whether a given human gut microbiome sample is characterized by a more favourable or unfavourable diet and health-associated species.

Several factors were crucial in the robust definition of the proposed microbiome species ranking systems. First, the scale of our combined cohorts with consistent experimental protocols for metagenomic sequencing and analysis is unprecedented. Second, the geographic diversity spanning all US states and UK regions, although confined to typical Westernized lifestyles and diets, allowed us to overcome local lifestyle-associated microbiome configurations. Third, consistent long- and short-term diet logging data processed in an integrative quantitative approach and validated markers that are intermediary measures of cardiometabolic health, and more advanced postprandial metabolomic-derived markers, enabled a fine-grained definition of relative health gradients across the surveyed populations. Fourth, publicly available datasets that were processed and curated uniformly, permitted independent validation and generalization of the results

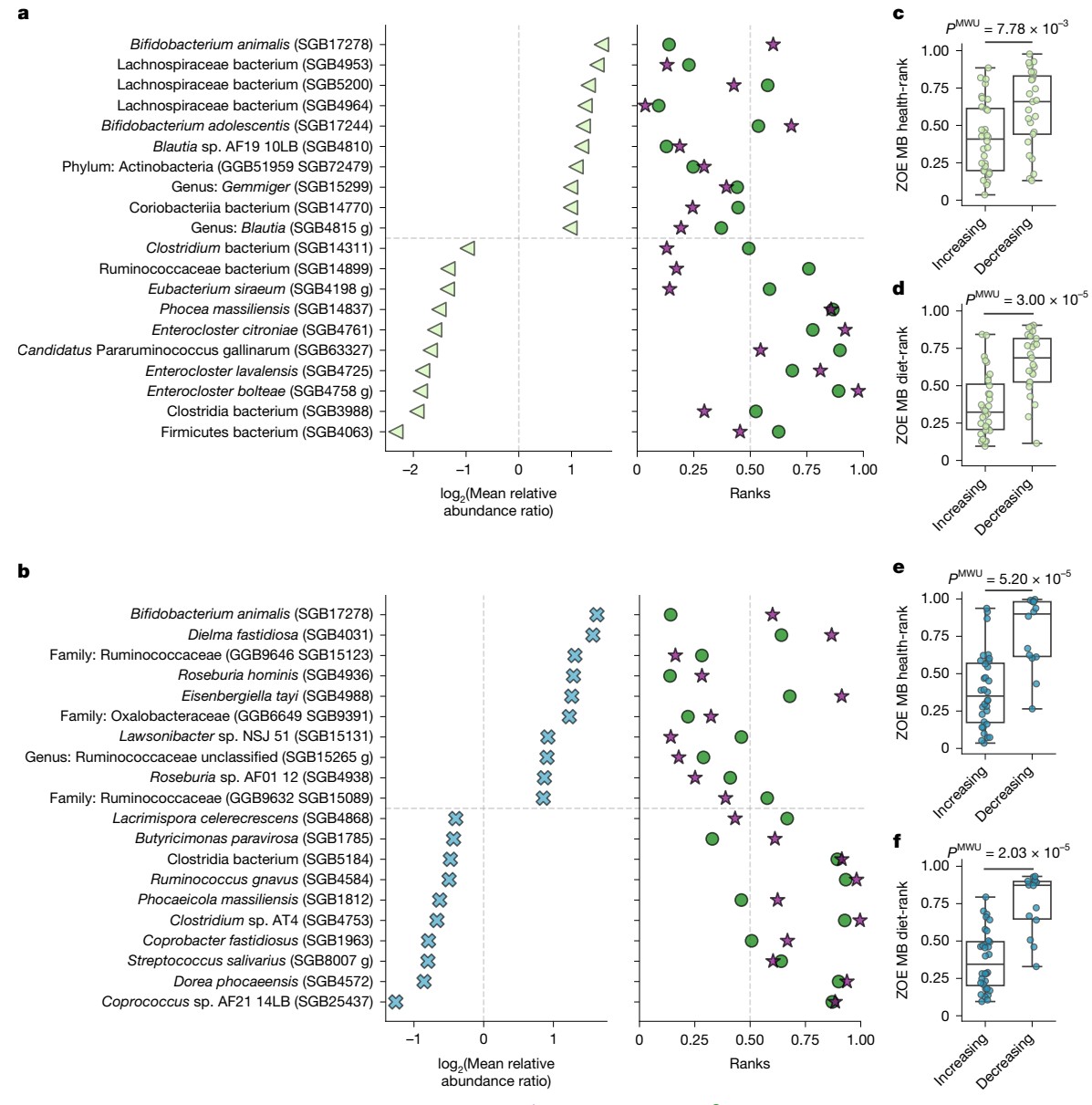

**Fig. 5 | Gut microbial SGBs that increase after dietary interventions are linked to more favourable ZOE MB health- and diet-ranks. a**, The 20 most significant gut microbial SGBs with the greatest effect sizes following the BIOME dietary intervention from Fig. 4a (left), paired with their ZOE MB health-ranks and diet-ranks, if available (right). **b**, The 20 most significant gut microbial SGBs with the greatest effect sizes following the METHOD personalized dietary intervention programme from Fig. 4b (left), paired with their ZOE MB health-ranks and diet-ranks, if available (right). The $x$ axis shows the $\log_2$-transformed ratio of mean relative abundance SGB values at endpoint over baseline. All values are reported in Supplementary Table 25. **c**,**d**, The distributions of the ZOE MB health-ranks (**c**) and diet-ranks (**d**) for the prebiotic blend arm of the BIOME cohort ($n = 57$ of tested SGBs). **e**,**f**, The distributions of the ZOE MB health-ranks (**e**) and diet-ranks (**f**) for the PDP arm of the METHOD cohort ($n = 46$ of tested SGBs). The distributions show that SGBs increasing in relative abundance have significantly more favourable ranks, whereas decreasing SGBs have more unfavourable ranks (two-sided Mann–Whitney $U$-test, $P = 7.78 \times 10^{-3}$, $P = 3.00 \times 10^{-5}$, $P = 5.20 \times 10^{-5}$ and $P = 2.03 \times 10^{-5}$, respectively). Distributions of the ZOE MB health-ranks and diet-ranks for the other arms are reported in Extended Data Fig. 10e,f. Box plots as in Fig. 1.

and showed the relevance of the species rankings toward additional conditions and diseases not evaluated in the original populations. We acknowledge that the demographic composition of the cohorts may influence some associations, and we are continuing to expand in both population scale and precision of each host-associated readout.

Our microbiome species ranking system proved accurate in reflecting changes induced by large-scale dietary intervention trials with associated host marker improvements (Figs. 4 and 5). Indeed, the cross-sectional associations were reflected in a significant and substantial increase of health-associated microbiome species and

a reduction or depletion of unfavourably ranked species. Many health-associated host markers are co-correlated because they are nutritional indicators, and disentangling their direct interactions from those mediated by the microbiome will remain elusive until large-scale microbiome interventions become possible in humans. In this respect, one key limitation of our study design is that it does not allow directly disentangling of the effect that diet exerts on the microbiome to improve cardiometabolic health from the impact of diet only. This is particularly important as diet-based ranks were more dependent on country-related differences compared with health ranks, and further

studies should explore food-specific links with gut microbial species and cardiometabolic outcomes in greater detail[50]. This would entail designing large-scale interventions in which both the introduction of single foods and alterations of specific microbiome characteristics (for example, by administration of specific microbiome members) are tested, which are ultimately required to provide causal evidence that personalized nutritional interventions targeting the microbiota have a robust and reproducible impact on cardiometabolic health. Nonetheless, the confirmation of the cross-sectional patterns along the diet–microbiome–health axis in longitudinal nutrition intervention trials not only increases the intrinsic value of the rankings but confirms that the human gut microbiome can be modulated successfully by dietary intervention and that the effects on the microbiome of such interventions are both predictable and reproducible. By providing the full list of ranked microbial species, this work can be exploited in future research on microbiome-powered precision nutrition and can be expanded in the future to more diverse populations and lifestyles that are currently underrepresented in microbiome, nutritional and health studies.

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

# Methods

## ZOE PREDICT cohorts definition

The ZOE PREDICT programme comprises several distinct studies that together constitute one of the largest multi-omic health initiatives, linking diet, person-specific metabolic responses to foods, and the gut microbiome. In this work, we considered and harmonized five ZOE PREDICT cohorts: PREDICT 1, PREDICT 2, PREDICT 3 US21, PREDICT 3 US22A, and PREDICT 3 UK22A. The PREDICT 1 cohort (NCT03479866) was described previously[9,51]. In brief, PREDICT 1 enrolled 1,098 participants ($n = 1,001$ from the UK and $n = 97$ from the USA) who underwent a clinical visit to collect anthropometric information and blood samples, followed by an at-home phase during which postprandial responses to both standardized tests and ad libitum meals were recorded. Stool samples were collected at home before the in-person clinical visit. The PREDICT 2 study (NCT03983733) had a similar collection protocol to PREDICT 1 but was conducted entirely remotely and included data from 975 people from 48 US states (including the federal District of Columbia and without participants from North Dakota and Hawaii). The PREDICT 3 cohorts (US21, US22A and UK22A) are research cohorts (NCT04735835) embedded within the ZOE commercial product. Participants provide informed written consent for their data to be used for scientific research purposes. In total, 32,621 samples ($n = 11,798$ for US21, $n = 8,470$ for US22A and $n = 12,353$ UK22A) were collected and retrieved. The studies were fully remote, participants completed health and food questionnaires at baseline, and self-collected and shipped stool samples. Cardiometabolic markers were collected as described below. Furthermore, we considered and analysed two registered clinical nutritional intervention studies, namely METHOD[36] (NCT05273268) and BIOME[37] (NCT06231706), focusing on the microbiome changes and their links with the two derived SGB-level rankings (ZOE MB health-ranks and diet-ranks). All study protocols are registered and available on clinicaltrials.gov through the clinical trials number and link affiliated with each trial.

## Sample collection, DNA extraction and sequencing

For the PREDICT 1 cohort, sample collection, DNA extraction and sequencing were described previously[9]. The PREDICT 2 samples were collected in Zymo buffer, DNA extraction was performed at QIAGEN Genomic Services using DNeasy 96 PowerSoil Pro, and sequencing was performed on the Illumina NovaSeq 6000 platform using the S4 flow cell and targeting 7.5 Gb per sample. The PREDICT 3 samples were self-collected into tubes containing the DNA-Shield Zymo buffer. Sample processing was performed by Zymo and Prebiomics. In brief, DNA extraction by Zymo used the ZymoBIOMICS-96 MagBead DNA kit, whereas Prebiomics used the DNeasy 96 PowerSoil Pro kits. Sequencing libraries were prepared using the Illumina DNA Prep Tagmentation kit, following the manufacturer's guidelines. Whole-genome shotgun metagenomic sequencing on the Illumina NovaSeq 6000 platform used the S4 flow cell and targetted 3.75 Gb per sample.

All raw sequenced data were quality controlled using the preprocessing pipeline available at https://github.com/SegataLab/preprocessing, which comprises three steps: (1) removal of reads with low-quality ($Q < 20$), too short (length under 75 nt), or with more than two ambiguous bases; (2) removal of host contaminant DNAs (Illumina's spike-in phiX 174 and human genomes, hg19); and (3) synchronization of paired-end and unpaired reads.

## Dietary data processing

In the PREDICT cohorts, we assessed long-term food intakes using FFQs, which were largely consistent across cohorts. Specifically, for PREDICT 1 participants (UK), we used a modified 131-item European Prospective Investigation into Cancer and Nutrition (EPIC) FFQ[52]. Participants in PREDICT 2 (USA) were surveyed using a similarly validated Diet History Questionnaire-III FFQ, including 135 items about food and beverages, as well as 26 questions about dietary supplements[53]. In PREDICT 3 UK22A and US22A, we developed and used a 264-item FFQ adapted from the EPIC-Norfolk Study FFQ and the Diet History Questionnaire-III. Consequently, there is a large overlap between the food items collected across the FFQs; for example, 90% of questions in the EPIC FFQ are included in the PREDICT 3 FFQ. This FFQ also includes additional food items to accurately capture modern eating habits—a limitation of older FFQ versions[54]. In the PREDICT 3 US21 cohort, FFQs were not collected, and only short-term logged dietary data collected using the ZOE mobile phone app were used instead.

Starting from both long- and short-term dietary data, we computed three versions of the PDI[55], namely, the overall PDI, the healthful PDI (measuring the adherence to a healthier plant-based foods diet) and the unhealthy PDI (measuring the intake of unhealthful plant-based foods), as well as the healthy eating index[23] (measuring how consumed foods align with dietary guidelines), the alternative Mediterranean diet score (measuring the adherence to a Mediterranean diet)[56] and the Healthy Food Diversity (HFD) index (measuring the number, distribution and health value of consumed foods)[57]. Specifically, to calculate PDIs and the healthy eating index, food items were first assembled into food groups by mapping them onto a 'food tree' consisting of a database of nutrient information arranged according to a hierarchical tree structure: level 1 (9 food groups), level 2 (52 food groups) and level 3 (195 food groups). UK foods were mapped onto the Composition of Foods Integrated Dataset (CoFID)[58] using food categories or sub-group codes, whereas US foods were similarly mapped onto the US Department of Agriculture Food and Nutrient Database for Dietary Studies database. Level 3 foods were aggregated and harmonized by nutrition scientists to allow for comparisons across cohorts. The Mediterranean diet and HFD scores were calculated as described previously[9].

## Host health and anthropometric marker collection

In PREDICT 1, sex and age were self-reported, whereas height, weight and blood pressure were measured at a clinic visit (day 0). At the clinic visit, participants were also fitted with wearable continuous glucose monitor CGM) devices (Abbott Freestyle Libre Pro (FSL)), visceral fat mass was measured using dual-energy X-ray absorptiometry scans following standard manufacturer's recommendations (DXA; Hologic QDR 4500 plus) and fasting GlycA was measured using a high-throughput NMR metabolomics (Nightingale Health) 2016 panel. Fasting and postprandial venous blood samples were also collected at the clinic; plasma glucose and serum total cholesterol, HDL-C and triglycerides were measured using Affinity 1.0, and whole blood HbA1c% was measured using Viapath. The ten-year ASCVD risk was calculated as per the 2019 American College of Cardiology (ACC) and American Heart Association (AHA) clinical guidelines[59]. Additional data were collected over the subsequent 13-day period at home; postprandial responses to eight standardized meals (seven in duplicate) of differing macronutrient (fat, carbohydrate, protein and fibre) content were measured using CGMs and dried-blood-spot analysis as described previously[13]. T2D and hyperlipidemia were self-reported via health questionnaires. The PREDICT 2 and PREDICT 3 studies were fully remote. Sex, age, height, weight and blood pressure were self-reported, and fasting and postprandial responses for total cholesterol, HDL-C, triglycerides and HbA1c were assessed using whole blood finger-prick samples collected at home using dried-blood-spot analysis by commercial laboratories (CRL, Eurofins Biomnis). CGMs were fitted at home by participants. A selection of standardized meals smaller than in PREDICT 1 was tested in PREDICT 2 and PREDICT 3 (a metabolic challenge meal, and medium-fat and carbohydrate breakfast and lunch meals). Some of the considered markers represent the same metabolic function over time and showed positive correlations between their fasting and postprandial measurements, whereas others represent opposite types of the same biomolecular pathway and showed negative correlations among them (Supplementary Table 4).

## Public human microbiome datasets

We leveraged 27,011 public metagenomic samples from 107 cohorts available through the curated MetagenomicData 3 (cMD3) resource[60,61] to define the cohorts used for the meta-analyses on BMI and healthy–diseased comparison ('Statistical and meta-analyses'). For the meta-analysis on BMI, we selected cohorts with stool microbiome samples from healthy participants (self-assessed, not reporting a diagnosis), aged at least 16 years, BMI ≥ 18.5 and sex information available. Cohorts with fewer than 30 people were excluded. Furthermore, the ThomasAM_2018_c and LeChatelier_2013 cohorts were excluded as duplicates in the YachidaS_2019 and NiesenHB_2014 cohorts, respectively. Overall, 6,182 samples from 34 different cohorts and 20 countries were retrieved. Participants were classified into three categories: healthy weight (BMI ≥ 18.5 and <25), overweight (BMI ≥ 25 and <30) and obese (BMI ≥ 30). Then, each combination of country, dataset and two BMI categories was tested if at least 15 samples were retained. These led to analysing a total of 5,348 samples from 27 cohorts (2,837 healthy weight, 1,562 overweight and 949 obese participants; Supplementary Table 9). For the health–diseased meta-analyses, we selected from cMD3 participants aged at least 16 years, BMI ≥ 18.5 and the sex information available that were part of a case–control study of one of the following diseases: CRC, IBD (including ulcerative colitis and Crohn's disease), T2D, IGT and ASCVD. Studies with fewer than 30 people were excluded. In total, we considered ten datasets of CRC (650 cases and 645 controls), two datasets of IGT (273 cases and 492 controls), five datasets of T2D (775 cases and 900 controls), three datasets of IBD (103 controls, 59 of which used in two different comparisons, 60 individuals with Crohn's disease and 68 individuals with ulcerative colitis) and three datasets of CVD (283 cases and 508 controls). Notably, German and French participants of the MetaCardis cohort were separated, and this led to a set of 449 controls used in both the T2D and the IGT analyses, whereas only the 176 controls from France were used in the CVD analysis. Overall, the total number of samples analysed was $N = 4,816$ (2,707 controls and 2,109 cases) from 25 cohorts and 10 countries (Supplementary Table 20). The cohort selection for the two analyses used the script https://github.com/waldronlab/curatedMetagenomicDataAnalyses/blob/main/python_tools/meta_analysis_data.py available in cMD3.

## Microbiome taxonomic profiling

All microbiome samples from the PREDICT cohorts were profiled using MetaPhlAn 4 (v.4.beta.2, database vJan21_CHOCOPhlAnSGB_202103), without performing read subsampling, as the benefit of occasionally detecting a few additional low-abundance species in samples with a higher number of reads outweighs the potential noise from uneven sequencing depth. Samples retrieved from cMD3 (described in 'Public human microbiome datasets') were profiled with MetaPhlAn 4 (v.4.beta.1, database vJan21_CHOCOPhlAnSGB_202103) using default parameters in both cases (among default parameters, the stat_q is set to 0.2 by default, which defines the quantiles for the robust average coverage calculation), which precludes the necessity for additional prevalence filters considering its default parameters are tailored for the taxonomic profiling of human microbiome samples[19]. MetaPhlAn 4 is a publicly available taxonomic profiler for metagenomic samples (Github repository: https://github.com/biobakery/MetaPhlAn) that leverages medium and high-quality genomes from isolates and metagenome-assembled genomes (MAGs). Isolate genomes and MAGs are clustered at 95% average nucleotide identity to define SGBs, as described previously[20]. If an SGB cluster contains a genome isolate, then it is referred to by that isolate's taxonomic label. If an SGB contains only MAGs, then it represents an unknown species cluster and is assigned the taxonomic label of a genus, family or phylum, according to which is the genomically closest to a taxonomic label from isolate genomes. As the taxonomic classification of MetaPhlAn depends on species-specific marker genes, sometimes there are several SGBs of very closely related genomes for which the identification of SGB-specific markers is not feasible. In this case, more than one SGB can be considered together, and the label '_group' is appended to the representative SGB ID. In this way, MetaPhlAn 4 improves the resolution of the taxonomic profiling task[62].

## Rankings definition

We first identified a subset of prevalent SGBs to ensure a minimum number of non-zero relative abundance values. In each PREDICT cohort, we selected markers that are intermediary measures of host health or diet health, and they were organized into four categories: personal, dietary, fasting and postprandial (Supplementary Table 2). Second, we calculated the partial Spearman's correlation between each SGB and health markers, adjusting for sex, age and BMI, using the 'pingouin' Python package (v.0.5.4, https://github.com/raphaelvallat/pingouin) (Extended Data Figs. 3 and 5). The relative abundance values of SGBs (including zeros) were used as input for the correlations. Third, the SGB-marker partial correlations were sorted ascending if the marker was considered as positive with respect to health, or descending if the marker was considered as negative. These sorted partial correlations were ranked and normalized according to cohort sample sizes into percentiles ranging from 0 to 1 (function pandas.DataFrame.rank with param pct=True from pandas v.2.1.3) (Fig. 2b,c and Extended Data Figs. 2 and 4). Fourth, for each category of markers, we computed the average percentiles across markers (Fig. 2a and Extended Data Fig. 4). SGBs were retained in the overall rankings if they were ranked in at least two different cohorts, leading to a final ranking of 661 SGBs. Finally, the ZOE MB health-rank 2025 was defined by first averaging the personal, fasting and postprandial category percentiles within each cohort, and then averaging these cohort-specific averages. The ZOE MB diet-rank 2025 instead was defined by averaging the dietary percentiles across all cohorts (Fig. 2a, Extended Data Fig. 4 and Supplementary Table 5). The ZOE Microbiome Rankings are also available at https://zoe.com/our-science/microbiome-ranking.

## Machine learning

To assess the link to the human gut microbiome composition, we developed and used a machine learning framework based on random forest classification and regression algorithms from the scikit-learn (v.1.3.2) Python package (as implemented in the RandomForestClassifier and RandomForestRegressor functions, respectively), both with 'n_estimators=1000' and 'max_features=sqrt' parameters[63]. We trained random forest classifiers and regressors on MetaPhlAn 4-estimated SGB-level relative abundances (arcsine square-root transformed) to assess the extent to which the outcome variable was predictable from the microbiome as a proxy of the strength of the microbiome–variable association. This framework was used and described originally in ref. 9 and accounts for the presence of twin pairs in the data, which avoids biases due to identical values in twins. In brief, the framework uses a cross-validation approach, splitting the dataset randomly into training and testing folds with an 80:20 ratio, respectively, and repeated 100 times (as implemented in the StratifiedShuffleSplit function). Folds are also constructed to maintain a similar ratio of the two classes to predict as they appear in the full data. For target variables with continuous values, classification was performed by contrasting the first against the fourth quartile, the first three against the fourth quartile and the first against the last three quartiles. Performances were evaluated using the AUC for the classification task, whereas Spearman's correlation between the real and predicted values was used for the regression task[22].

## Statistical and meta-analyses

We performed a meta-analysis to determine the possible links between BMI (categorized into 'healthy weight', 'overweight' and 'obese') and our ranked SGBs across various publicly available studies comprising a total of 5,348 people who were not diagnosed with any specific disease. We first evaluated the ZOE MB health- and diet-ranks by assessing the

cumulative relative abundance and richness of the 50 most favourable and the 50 least favourable SGBs in each dataset in each BMI category: healthy weight, overweight and obese (see 'Public human microbiome datasets' for the specific cut-offs). Specifically, we assessed the number of intra-dataset, between-BMI groups pairwise comparisons in which the group median abundance or the group median count was higher in the lower BMI group (when considering most favourable SGBs from both ranks) or higher in the higher BMI group (when looking at least favourable SGBs). Next, we fit linear models for each dataset and pair of BMI categories: healthy weight versus overweight, healthy weight versus obese, and overweight versus obese. In the first model, we looked at the count of the 50 most favourable and unfavourable ZOE MB health- and diet-ranked SGBs. A second model was fitted on the cumulative relative abundance (arcsine square-root transformed) of the 50 most favourable and unfavourable SGBs in the two rankings. All models were adjusted by sex and age. Cohen's $d$ was used to estimate the effect size of the normalized difference between unfavourable and favourable ranked SGBs when considering cumulative abundances. This quantifies the difference between the means of two groups in terms of standard deviations. Specifically, as originally defined, a 'small' effect size corresponds to $d = 0.2$, a 'medium' effect size to $d = 0.5$ and a 'large' effect size to $d = 0.8$ (ref. 64). In these models, the lower BMI category of each comparison was used as the negative control, so negative coefficients reflect a higher count of SGBs in the lower BMI category, whereas positive coefficients reflect a higher count of SGBs in the higher BMI category. Effect sizes were summarized through meta-analysis, computed as a random-effect model using the Paule–Mandel heterogeneity on adjusted mean differences from the linear regression models (standardized for cumulative abundances). We assessed the presence of the 50 most favourable and most unfavourable SGBs from both the ZOE MB health- and diet-ranks among the countries considered in these analyses (18 in total) and when considering only people of healthy weight ($n = 2,837$). To link the ranked SGBs with the country, we fit a linear model on the count and cumulative relative abundance of the SGBs, and the models were adjusted by the sequencing depth of the study. We used ordinary least squares adjusted by sequencing depth when comparing two datasets from different countries, and linear mixed model blocked by dataset ID and adjusted by sequencing depth when comparing pairs of countries in which at least one country was represented by more than one dataset (country- and sequencing depth-adjusted $P$ values are presented in Supplementary Table 11).

A second meta-analysis tested the associations between our ZOE MB health- and diet-ranked SGBs and five gut-associated diseases (CVD, T2D, IBD, CRC and IGT) across studies, for a total of 4,816 samples ('Public human microbiome datasets'). Linear models were used to predict the binary disease outcome (healthy versus diseased) for each disease, using the cumulative abundances (arcsine square-root transformed) of the 50 most favourable or unfavourable SGBs, adjusting by sex, age and BMI. The betas of the linear models were converted into SMDs as described previously[65]. We also defined models to predict healthy versus diseased using the sum of the SGB ranks normalized between −1 and 1, considering all 661 SGBs for the ZOE MB health- and diet-ranks, once using the direct sum of the SGB ranks and once weighting ranks by the relative abundance of each SGB in each sample (transformed using the arcsine and square-root function to avoid overestimating the ranks of highly abundant species due to compositionality). SMDs were calculated similarly to those in the previous case. In all meta-analytical models, the set of cohorts considered comprised studies encompassing several diseases with a shared control group that we analysed separately. To account for the overlaps in the studies considered, we computed weights based on the inverse effect sizes variance-covariance matrix, as suggested previously[66,67]. Thus, five meta-analyses were performed, one for each disease: CVD (three datasets), T2D (six datasets), IBD (three datasets), CRC (ten datasets) and IGT (two datasets). Of note, in the comparisons of controls versus T2D, IGT and CVD, the MetaCardis French and German sub-cohorts were considered as different datasets, and their controls were meta-analysed as different cohorts. In particular, only French control samples were used in the CVD analysis, which included only French cases. Finally, meta-analysis summaries were computed using the same technique. Analyses we carried out with Python (v.3.12.0), using also the following libraries: numpy (v.1.26.2), scipy (v.1.11.4), statsmodels (v.0.14.0), and matplotlib (v.3.8.2) and seabron (v.0.11.2) for visualization.

## Ethical compliance

All study protocols are registered on clinicaltrials.gov and procedures are compliant with all relevant ethical regulations. Ethical approval for the PREDICT 1 study was obtained in the UK from the King's College London Research Ethics Committee (REC) and Integrated Research Application System (IRAS 236407), and in the USA from the institutional review board (Partners Healthcare Institutional Review Board (IRB) 2018P002078). Ethical approval for the PREDICT 2 study (Pro00033432) was obtained from Advarra IRB. Ethical approval for the PREDICT 3 study (Pro00044316, HR/DP-21/22-28300 and HR/DP-24/25-45829) was obtained from Advarra IRB and King's College London REC. Ethical approval for the METHOD study (Pro00044316; protocol no. 00044316) was obtained from Advarra IRB. Ethical approval for the BIOME study (HR/DP-23/24-39673) was obtained through King's College London REC. All participants provided written informed consent and all studies were carried out in accordance with the Declaration of Helsinki and Good Clinical Practice.

## Reporting summary

Further information on research design is available in the Nature Portfolio Reporting Summary linked to this article.

## Data availability

The human genome version used in the preprocessing of the microbiome samples is the GRCh37 genome assembly (hg19, GCA_000001405.1). Raw metagenomic samples, along with metadata information (sex, age, BMI and country) and microbiome profiles for all participants of the ZOE PREDICT Studies, are publicly available. Metagenomes from PREDICT 1 are publicly available as previously reported[9], whereas the PREDICT 2 and PREDICT 3 cohorts (US21, US22A and UK22A) are deposited in the European Nucleotide Archive (ENA) of the European Bioinformatics Institute (EBI) under accession numbers PRJEB75460, PRJEB75462, PRJEB75463 and PRJEB75464, and are publicly accessible. Sex, age, BMI, country and quantitative taxonomic profiles for each sample are publicly available within the curated MetagenomicData package[60] and at Zenodo (https://doi.org/10.5281/zenodo.15307999)[68]. The full list of species for the ZOE Microbiome Rankings are publicly available at https://zoe.com/our-science/microbiome-ranking, where future updates will also be made available. The version of the ZOE Microbiome Rankings discussed in the present work is reported in Supplementary Table 5. To protect participant privacy, individual participant clinical data are not publicly available and cannot be deposited in public repositories. Researchers may request access to the restricted data by submitting a research proposal via email to data.papers@joinzoe.com. All proposals will be reviewed by a sub-panel of the ZOE Scientific Advisory Board within 4 working weeks. Proposals, researchers or institutions requesting data will be approved if they meet the standard criteria related to ethics, privacy and data protection regulations. Approved researchers are required to enter into a data-sharing agreement with ZOE. The requested host parameters will be provided as ordered data points without loss of reproducibility, as the analysis of this work (including deriving the ranks) was performed using non-parametric statistics. These data are available at Zenodo (https://doi.org/10.5281/zenodo.17236382)[69] and are encrypted; access to the data will be granted to researchers whose

proposals are approved. All data from non-PREDICT external public cohorts used to validate the rankings are available in full at Zenodo (https://doi.org/10.5281/zenodo.17236261)[70].

## Code availability

The custom Python code developed for the meta-analyses performed on public data and included in this work is available at GitHub (https://github.com/SegataLab/inverse_var_weight) and at Zenodo (https://doi.org/10.5281/zenodo.17236261)[70]. The MetaPhlAn code for the taxonomic profiling is available at GitHub (https://github.com/biobakery/MetaPhlAn), Zenodo (https://doi.org/10.5281/zenodo.17236261)[70] and Bioconda (https://bioconda.github.io/recipes/metaphlan/README.html).

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

**Acknowledgements** We thank all the participants of the PREDICT programme. This work was supported by Zoe Ltd. and TwinsUK, which is funded by the Wellcome Trust, Medical Research Council, Versus Arthritis, European Union Horizon 2020, Chronic Disease Research Foundation (CDRF), the National Institute for Health Research (NIHR) Clinical Research Network (CRN) and Biomedical Research Centre based at Guy's and St Thomas' NHS Foundation Trust in partnership with King's College London. It was also supported by the European Research Council (ERC-CoG microTOUCH-101045015) to N.S., by the European Union NextGenerationEU (Interconnected Nord-Est Innovation programme, INEST) to N.S., by the National Cancer Institute of the National Institutes of Health (1U01CA230551) to N.S. and by the Premio Internazionale Lombardia e Ricerca 2019 to N.S.

**Author contributions** F. Asnicar, S.E.B., T.D.S. and N.S. conceived and supervized the study. F. Asnicar, P.M., E.P., K.M. and A.A. performed the analyses: F. Asnicar collected the data, performed microbiome analyses and species rankings; P.M. performed meta-analyses on public data; E.P. supported the analysis of longitudinal data; K.M. helped with microbiome profiling; A.A. supported the analysis of the two clinical trial, nutritional intervention studies. E.B., S.G., F.G. and R.D. set up the interface to retrieve clinical data information. G.F., G.B., L.R., G.P., F. Amati and K.M.B. supported with results interpretation. J.W. is co-founder of ZOE Ltd. and made the assembly of these large cohorts possible. F. Asnicar, P.M. and N.S. drafted the manuscript. All authors reviewed and edited the manuscript.

**Competing interests** J.W. and T.D.S. are co-founders of ZOE Ltd.—a commercial initiative active in the field of personalized nutrition—and owners of the ZOE PREDICT studies. E.B., F. Amati, A.A., S.G., F.G., R.D., J.W. and K.M.B. are, or have been, employees of Zoe Ltd. F. Asnicar, S.E.B., T.D.S. and N.S. are consultants to ZOE Ltd. F. Asnicar, R.D., J.W., S.E.B., T.D.S. and N.S. receive options with ZOE Ltd. All other authors declare no competing interests. Zoe Ltd. holds the following patent applications on the SGBs ranking: PCT (World) patent pending applications PCT/EP2024/058262, PCT/EP2024/058286 and PCT/EP2024/058290.

**Additional information**
**Correspondence and requests for materials** should be addressed to Francesco Asnicar or Nicola Segata.

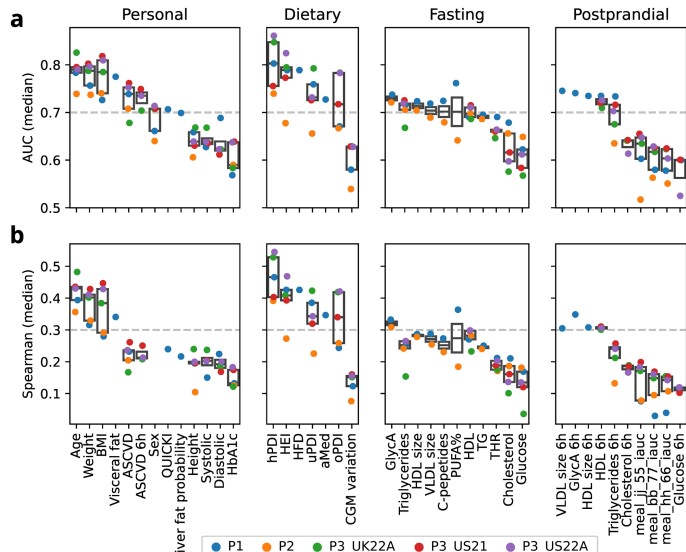

**Extended Data Fig. 1 | Microbiome predictive potential for personal information, dietary indices, fasting, and postprandial metabolic markers, via classification and regression random forest models.** Distributions of the random forest median AUCs (**a**) and median Spearman's correlation coefficients (**b**) (Methods) in the five cross-sectional PREDICT studies for the different clinical data divided into four categories: 'Personal', 'Dietary', 'Fasting', and 'Postprandial'. The AUC and Spearman's index thresholds of 0.7 and 0.3, respectively, are indicated with a dashed line. **a**) Each point represents the median AUC value obtained in cross-validation for each cohort when testing the first versus the fourth quartile of the corresponding clinical marker values on the x-axis. **b**) Each point represents the median Spearman's correlation coefficient for the predicted values by the regressor and the true values in the cross-validation setting for each cohort.

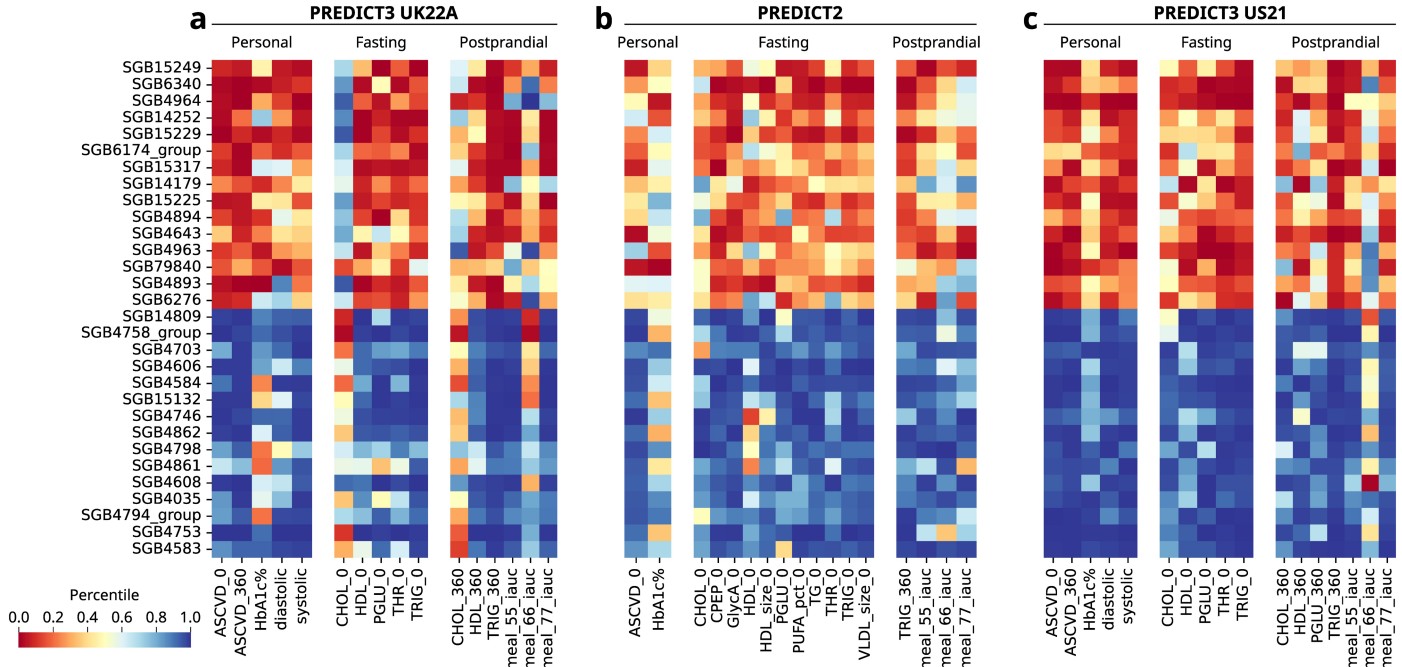

**Extended Data Fig. 2 | Detailed associations of the 15 top and bottom cardiometabolic-ranked SGBs in the PREDICT3 UK22A, PREDICT2, and PREDICT3 US21 cohorts.** The single-marker percentiles, divided into the three categories ('Personal', 'Fasting', and 'Postprandial') for the 15 most favorable and unfavorable ZOE MB Health-ranked SGBs the other three PREDICT cohorts not reported in Fig. 2 (**a**, PREDICT3 UK22A; **b**, PREDICT2, and **c**, PREDICT3 US21). Heatmaps with the single Spearman's partial correlations for all PREDICT cohorts are available in Supplementary Fig. 4.

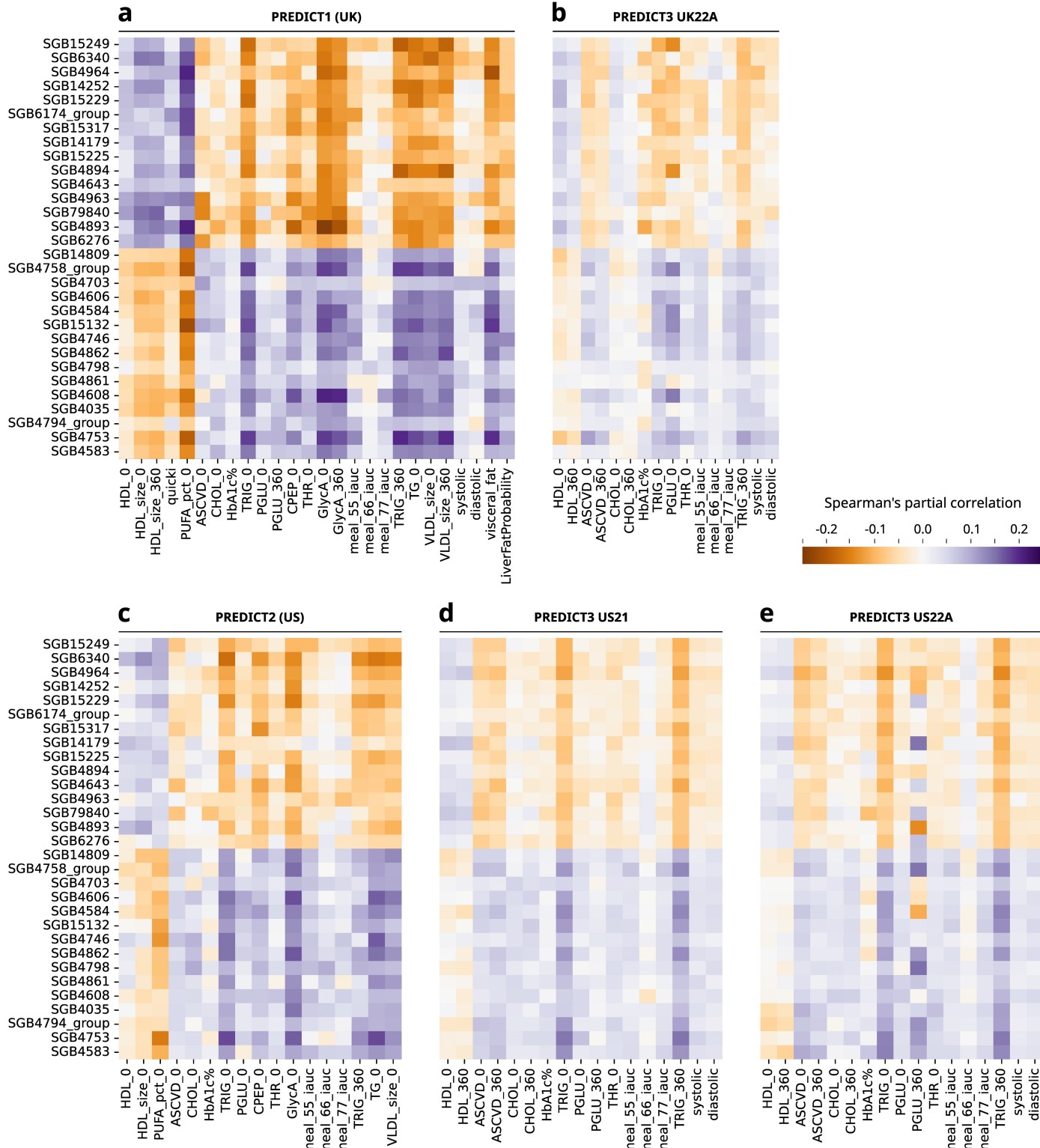

**Extended Data Fig. 3 | Spearman's partial correlations of the 15 top and bottom cardiometabolic-ranked SGBs. a-e)** Spearman's partial correlations (corrected for age, sex, and BMI) between SGB relative abundance and single marker values show consistency across the five PREDICT cohorts. These partial correlations were ranked and averaged first within and then across the three data categories ('Personal', 'Fasting', and 'Postprandial', reported in Supplementary Fig. 3) separately in each cohort. The cohorts' averages were then used to define the cardiometabolic rank (for those SGBs analyzed in at least two cohorts).

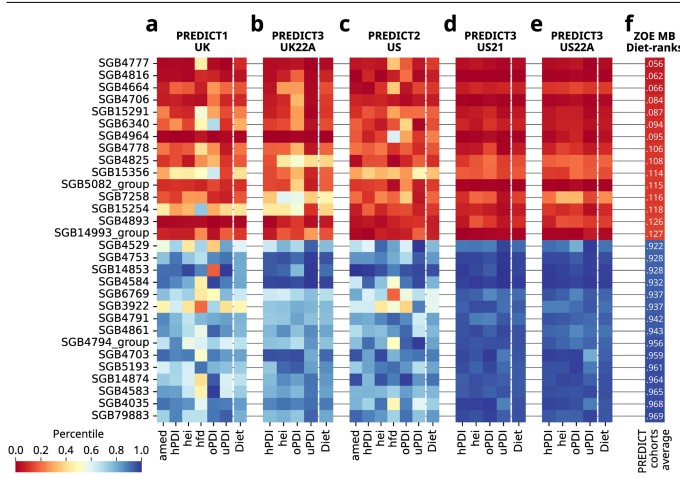

**Extended Data Fig. 4 | Diet associations of the 15 top and bottom diet-ranked SGBs. a-e**) For each PREDICT cohort, we computed Spearman's partial correlation between the SGBs' relative abundances and different diet indexes. Associations were ranked and averaged in each cohort separately. **f**) The ZOE MB Diet-ranking was computed for SGBs ranked in at least two PREDICT cohorts. The raw Spearman's partial correlations are available in Supplementary Fig. 7.

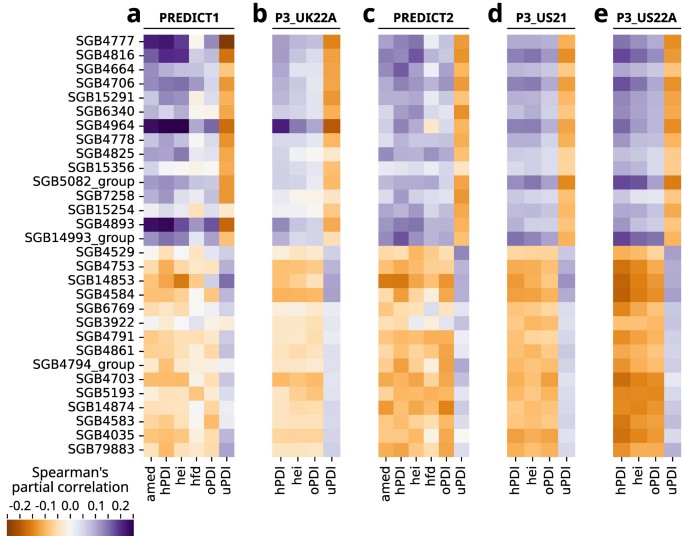

**Extended Data Fig. 5 | Spearman's partial correlations of the 15 top and bottom diet-ranked SGBs. a-e**) Study-wise Spearman's partial correlation coefficients (corrected for sex, age, and BMI) for the 15 most favorable and unfavorable ZOE MB Diet-ranked SGBs in different diet indexes. The associations appear consistent across cohorts.

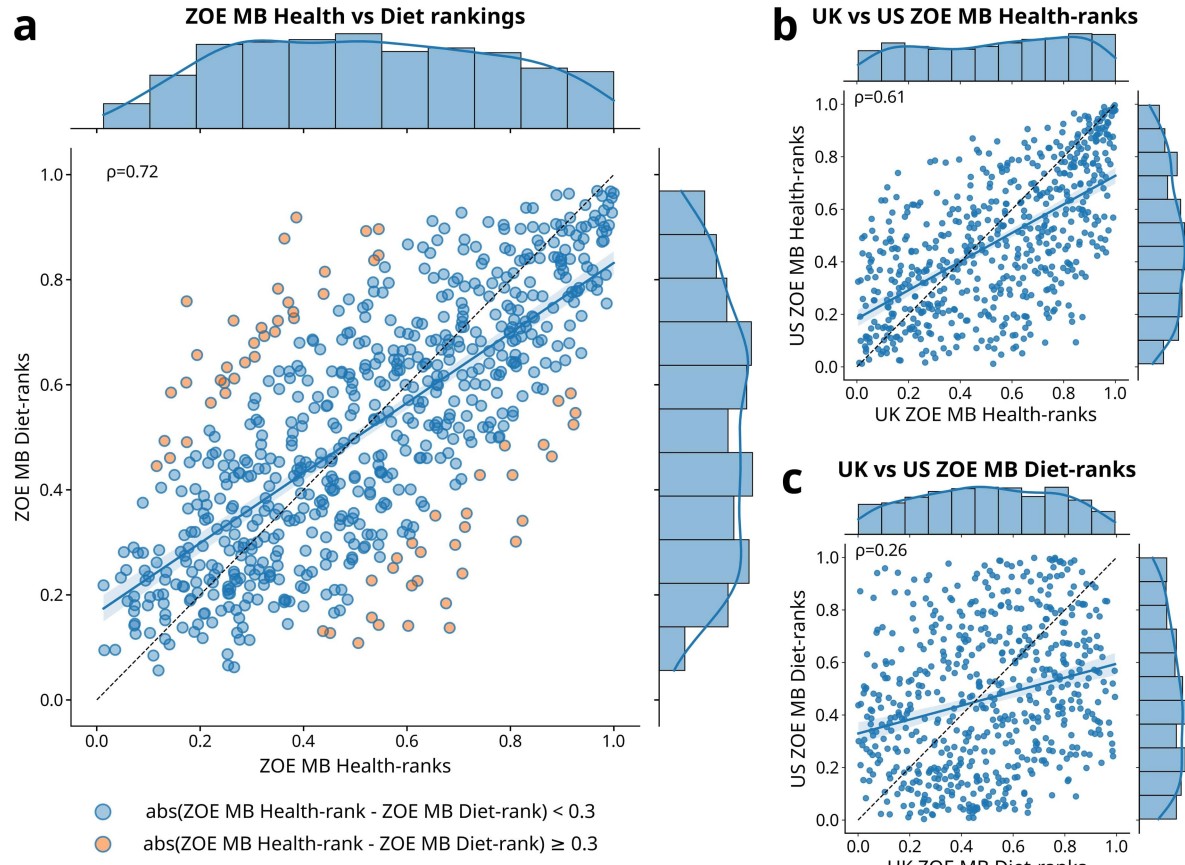

**a** ZOE MB Health vs Diet rankings

ρ=0.72

abs(ZOE MB Health-rank - ZOE MB Diet-rank) < 0.3
abs(ZOE MB Health-rank - ZOE MB Diet-rank) ≥ 0.3

**b** UK vs US ZOE MB Health-ranks

ρ=0.61

**c** UK vs US ZOE MB Diet-ranks

ρ=0.26

**Extended Data Fig. 6 | Comparison of the ZOE MB Health and Diet ranks and with geography. a**) The ZOE MB Health and Diet ranks are overall in agreement (Spearman's correlation = 0.72), albeit some SGBs show discordant rankings (absolute difference between the two ranks ≥ 0.3). These SGBs are highlighted in orange, and their ranks and taxonomy assignment are reported in Supplementary Table 6. **b,c**) Comparison of the ZOE MB Health (**b**) and Diet (**c**) ranks computed only on the PREDICT UK and US cohorts (Spearman's correlations of 0.61 and 0.26, respectively). The top and right-side histograms depict the x and y-axis marginal distributions in each plot.

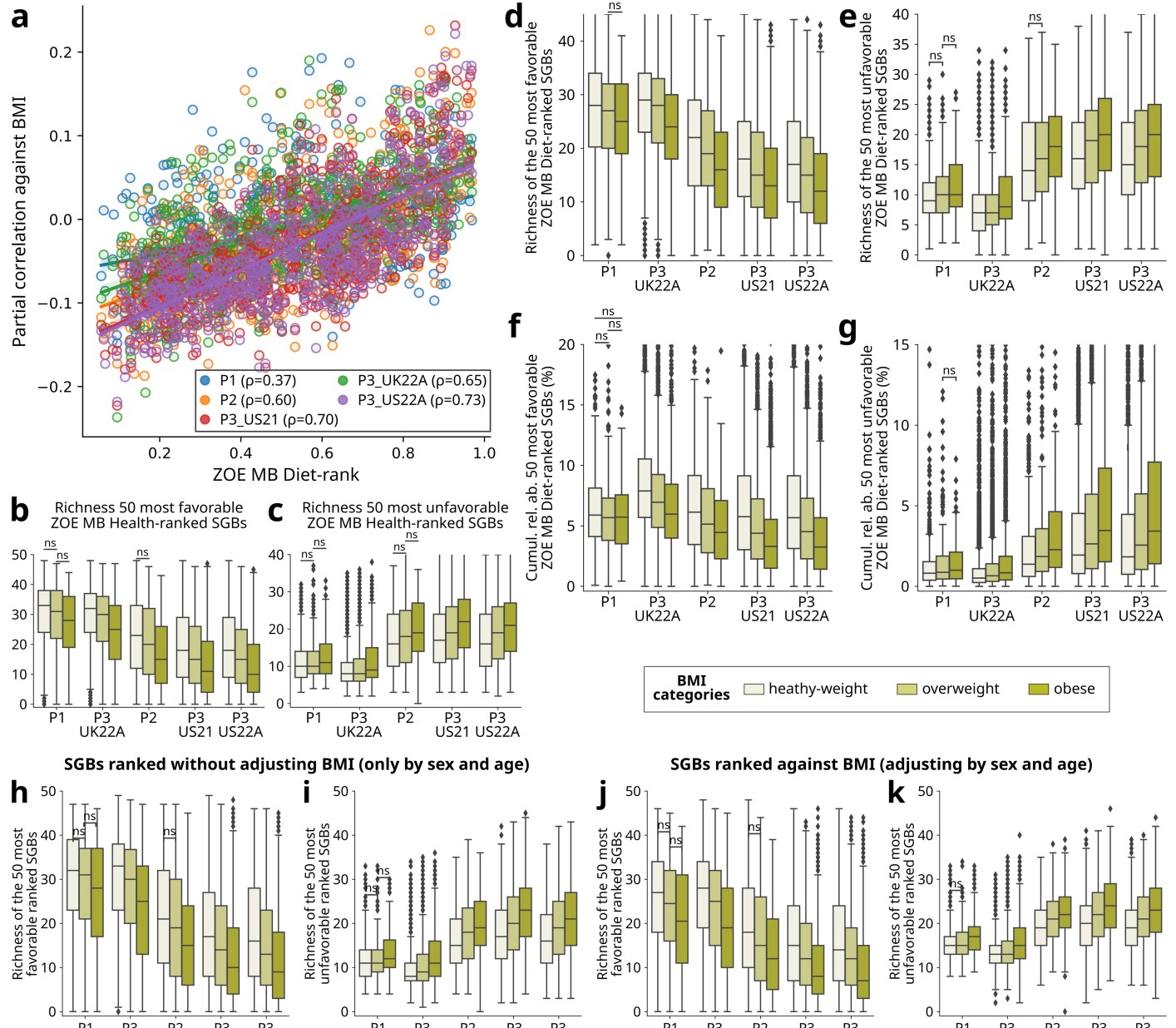

**Extended Data Fig. 7 | ZOE DIET ranks and their associations with BMI.**
**a**) Comparison of the ZOE MB Diet-ranks (x-axis) with the Spearman's partial
correlations (corrected for sex and age, y-axis) for the 661 ranked SGBs in the
five PREDICT cohorts. **b**) The number of the 50 most-favorably ranked SGBs
(ZOE MB Health-rank, Richness) detected in different BMI categories, showed
that increasing BMI, linked with increasing health risks, is reflected by a lower
presence of favorable SGBs. On the other hand, **c**) unfavorably-ranked SGBs
show an increasing count in higher-risk BMI categories. **d,e**) The box plots
report the number of the 50 most favorable and unfavorable ZOE MB Diet-ranked
SGBs of individuals stratified into three BMI categories (healthy-weight,
overweight, and obese) in each PREDICT cohort. **f,g**) Similarly, the box plots

represent the cumulative relative abundance of the 50 most favorable and
unfavorable ZOE MB Diet-ranked SGBs in individuals categorized into the three
BMI categories in each cohort. **h,i**) The box plots report the number of the
50 most favorably and most unfavorably ranked SGBs, ranked using the same
markers and categories as in the ZOE MB Health-ranks (Methods), but partial
correlations were corrected only for sex and age. **j,k**) Similarly, the box plots
report the count of the 50 most favorable and unfavorable SGBs in the three
BMI categories, with SGBs ranked according to their partial correlation with
BMI, adjusted by sex and age. Only non-significant FDR-corrected $P$ values
(ns, $P$ value > 0.01) from the Mann-Whitney U test are reported.

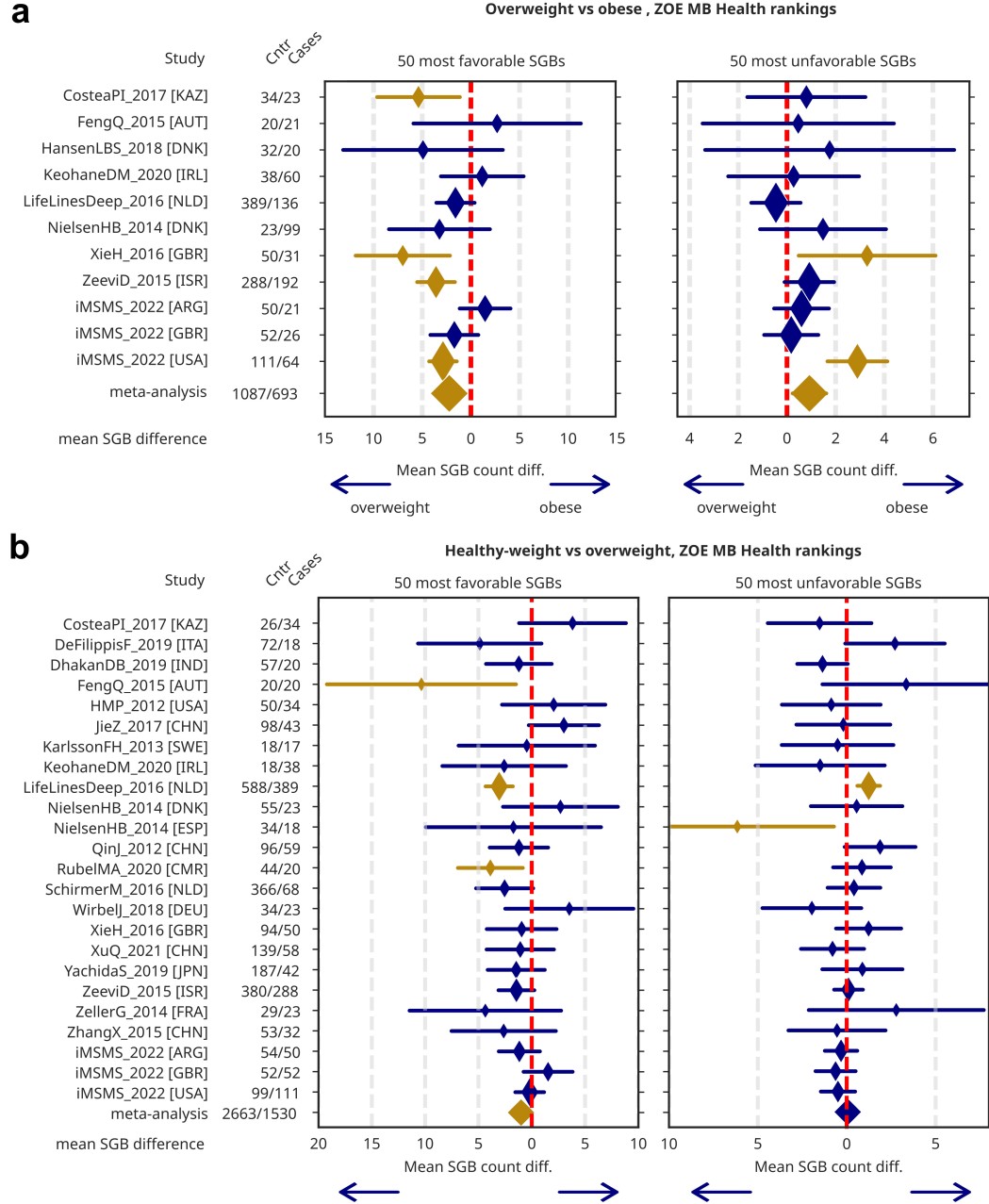

**Extended Data Fig. 8 | Meta-analysis of the 50 most favorably and unfavorably ZOE MB Health-ranked SGBs in overweight vs obese and healthy-weight vs overweight individuals. a)** Overweight individuals tend to carry a higher number of the 50 most favorably ZOE MB Health-ranked SGBs than obese individuals (left); the 50 most unfavorably ranked SGBs are increased in obese individuals vs overweight individuals (Methods). **b)** Healthy-weight individuals tend to carry a higher number of the 50 most favorably ZOE MB Health-ranked SGBs than overweight individuals (left); the 50 most unfavorably ranked SGBs are found in similar amounts in healthy-weight and overweight individuals (Methods). Error bars represent the 95% confidence interval.

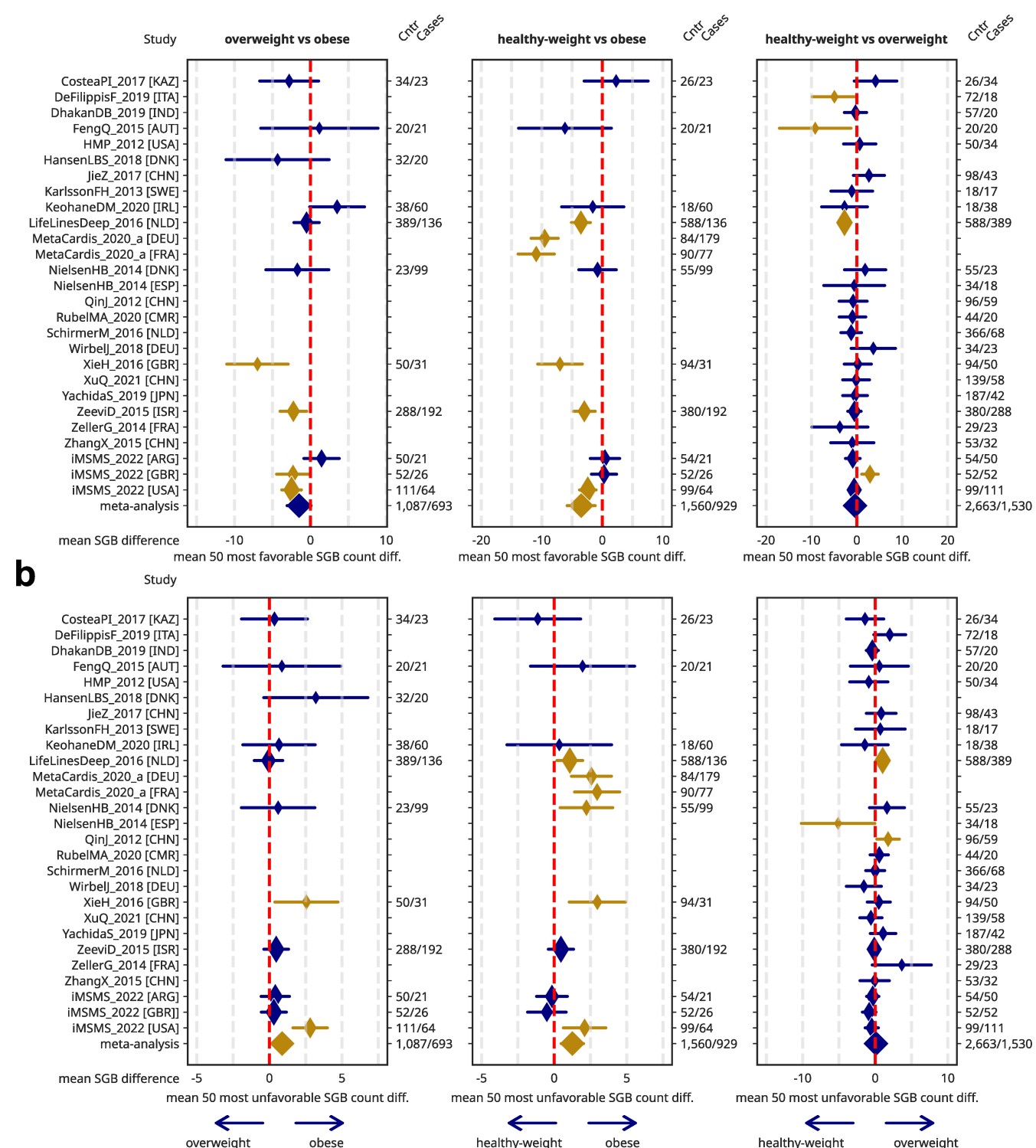

**Extended Data Fig. 9 | Meta-analysis of the 50 most favorably and unfavorably ZOE DIET-ranked SGBs comparing individuals from different BMI categories.** **a)** Comparison of the number of the 50 most favorable Diet-ranked SGBs in pairs of BMI categories. Healthy-weight and overweight individuals tend to have a higher number of favorably-ranked SGBs than obese individuals (Methods).

**b)** Comparison of the number of the 50 most unfavorably Diet-ranked SGBs in pairs of BMI categories. Obese individuals tend to have a higher number of unfavorably-ranked SGBs (Methods). Error bars represent the 95% confidence interval.

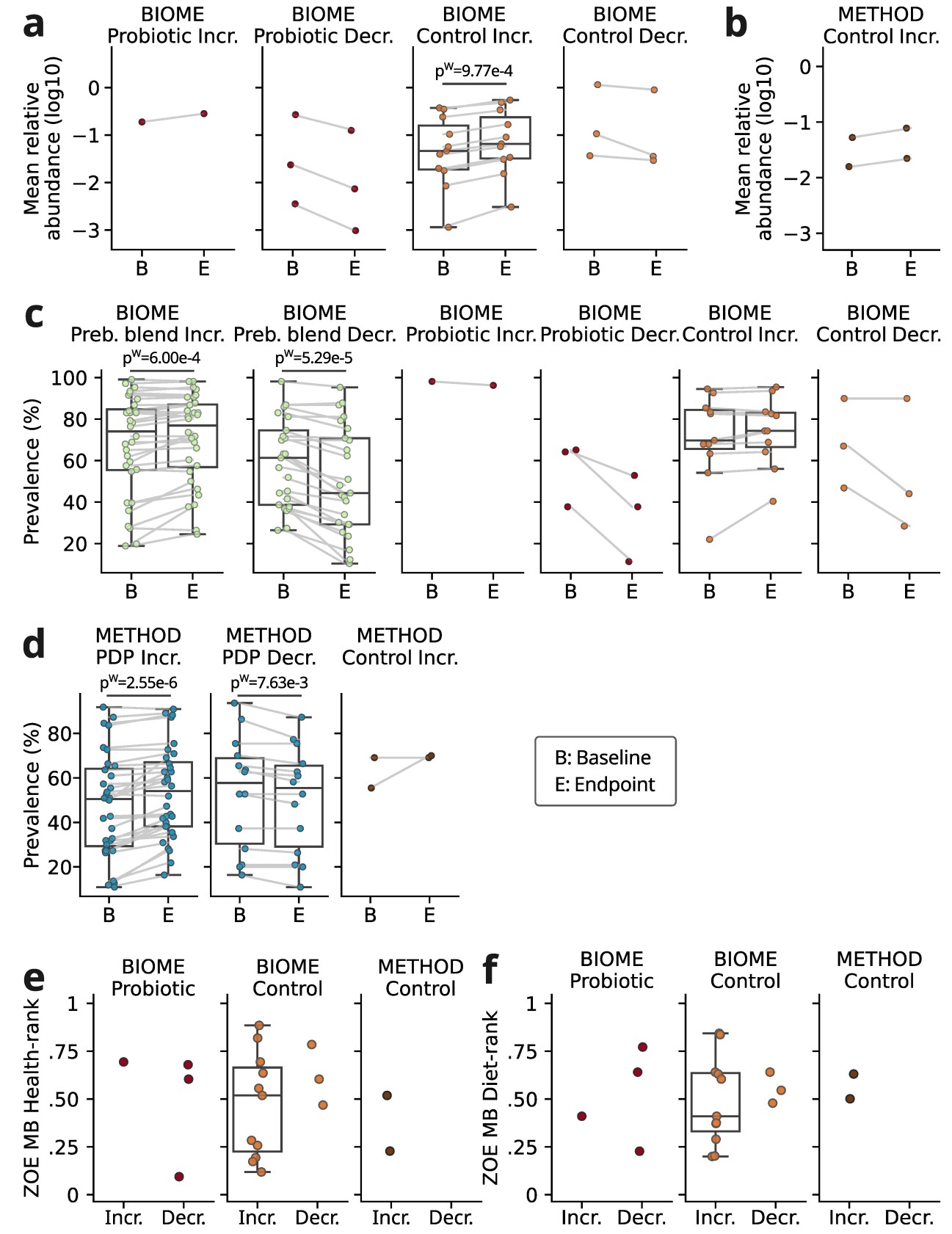

**Extended Data Fig. 10** | See next page for caption.

**Extended Data Fig. 10 | Significantly changing SGBs after dietary interventions show consistent patterns across cohorts in terms of relative abundance, prevalence, and ZOE MB Diet-ranks. a**) Distributions of the mean relative abundance of the significant SGBs for the probiotic and control arms of the BIOME cohort (relative to Fig. 4a). **b**) Distributions of the mean relative abundance of the significant SGBs for the control arm of the METHOD cohort (relative to Fig. 4b). **c**) Distributions of the prevalence of the significant SGBs of the BIOME cohort (relative to Fig. 4a) and **d**) of the METHOD cohort (relative to Fig. 4b). SGBs are separated into "increasing" and "decreasing", depending on their trend in relative abundance values, showing that SGBs found to be increased in relative abundance are also more prevalent, while the opposite is observed for SGBs decreasing in relative abundance. **e**) Distributions of the ZOE MB Health ranks for the significant SGBs in the Probiotic and Control arms of the BIOME cohort and METHOD cohorts. **f**) Distributions of the ZOE MB Diet ranks for the significant SGBs in the Probiotic and Control arms of the BIOME and METHOD cohorts.

# Reporting Summary

## Statistics

For all statistical analyses, confirm that the following items are present in the figure legend, table legend, main text, or Methods section.

| n/a | Confirmed | |
|---|---|---|
| ☐ | ☒ | The exact sample size (*n*) for each experimental group/condition, given as a discrete number and unit of measurement |
| ☐ | ☒ | A statement on whether measurements were taken from distinct samples or whether the same sample was measured repeatedly |
| ☐ | ☒ | The statistical test(s) used AND whether they are one- or two-sided<br>*Only common tests should be described solely by name; describe more complex techniques in the Methods section.* |
| ☐ | ☒ | A description of all covariates tested |
| ☐ | ☒ | A description of any assumptions or corrections, such as tests of normality and adjustment for multiple comparisons |
| ☐ | ☒ | A full description of the statistical parameters including central tendency (e.g. means) or other basic estimates (e.g. regression coefficient) AND variation (e.g. standard deviation) or associated estimates of uncertainty (e.g. confidence intervals) |
| ☐ | ☒ | For null hypothesis testing, the test statistic (e.g. *F*, *t*, *r*) with confidence intervals, effect sizes, degrees of freedom and *P* value noted<br>*Give P values as exact values whenever suitable.* |
| ☒ | ☐ | For Bayesian analysis, information on the choice of priors and Markov chain Monte Carlo settings |
| ☒ | ☐ | For hierarchical and complex designs, identification of the appropriate level for tests and full reporting of outcomes |
| ☐ | ☒ | Estimates of effect sizes (e.g. Cohen's *d*, Pearson's *r*), indicating how they were calculated |

*Our web collection on statistics for biologists contains articles on many of the points above.*

## Software and code

Policy information about availability of computer code

| Data collection | Sequenced metagenomic data were collected in FASTQ files, while individuals' metadata were self-reported using the ZOE mobile App. |
|---|---|
| Data analysis | MetaPhlAn (v4.beta.1 and v4.beta.2, both with database version Jan21_CHOCOPhlAnSGB_202103). All analysis were carried out in Python 3 (v3.12.0) using the following libraries: numpy (v1.26.2), scipy (v1.11.4), pandas (v2.1.3), pingouin (v0.5.4), sklearn (v1.3.2), and statsmodels (v0.14.0), and visualization was performed using the matplotlib (v3.8.2) and seaborn libraries (v0.11.2). Public metagenomic cohorts were retrieve from curatedMetagenomicData (v3). The custom Python code developed for the meta-analyses performed on public data and included in this work is available in the Github repository: https://github.com/SegataLab/inverse_var_weight and Zenodo (https://doi.org/10.5281/zenodo.17236262). The MetaPhlAn code for the taxonomic profiling is available in the Github repository: https://github.com/biobakery/MetaPhlAn, on Zenodo (https://doi.org/10.5281/zenodo.17236262), and via bioconda at https://bioconda.github.io/recipes/metaphlan/README.html. |

For manuscripts utilizing custom algorithms or software that are central to the research but not yet described in published literature, software must be made available to editors and reviewers. We strongly encourage code deposition in a community repository (e.g. GitHub). See the Nature Portfolio guidelines for submitting code & software for further information.

## Data

Policy information about availability of data

All manuscripts must include a data availability statement. This statement should provide the following information, where applicable:

- Accession codes, unique identifiers, or web links for publicly available datasets
- A description of any restrictions on data availability
- For clinical datasets or third party data, please ensure that the statement adheres to our policy

The human genome version used in the preprocessing of the microbiome samples is the GRCh37 genome assembly (hg19, GCA_000001405.1). Raw metagenomic samples, along with metadata information (sex, age, BMI, and country) and microbiome profiles for all participants of the ZOE PREDICT Studies, are publicly available. Metagenomes from PREDICT 1 are publicly available as previously reported 9, while the PREDICT 2 and PREDICT 3 cohorts (US21, US22A, and UK22A) are deposited in the European Nucleotide Archive (ENA) of the European Bioinformatics Institute (EBI) under accession numbers PRJEB75460, PRJEB75462, PRJEB75463, and PRJEB75464, and are publicly accessible. Sex, age, BMI, country, and quantitative taxonomic profiles for each sample are publicly available within the curatedMetagenomicData package 60 and at Zenodo (https://doi.org/10.5281/zenodo.15308000). The full list of species for the ZOE Microbiome Rankings are publicly available at https://zoe.com/our-science/microbiome-ranking, where future updates will also be made available. The version of the ZOE Microbiome Rankings discussed in the present work is reported in Supplementary Table 5. To protect participant privacy, individual participant clinical data are not publicly available and cannot be deposited in public repositories. ZOE is committed to supporting scientific reproducibility, and researchers requesting access to the data required for this purpose may submit a research proposal. Proposals, researchers, or institutions requesting data for reproducibility will be approved if they meet the standard criteria related to ethics, privacy, and data protection regulations. For reproducibility of the whole non-parametric analysis used to define the rankings, the required host parameters will be provided as ordered data points. This data is deposited in Zenodo (https://doi.org/10.5281/zenodo.17236383) and is encrypted; access to the data will be granted to researchers whose proposals will be approved. For proposals not intended for reproducibility, ZOE reserves all the rights to evaluate the scientific priority and relevance of the request. Researchers can request a proposal form by emailing data.papers@joinzoe.com. All proposals will be reviewed by a sub-panel of the ZOE Scientific Advisory Board within four working weeks. Approved researchers will be required to enter into a data-sharing agreement with ZOE. All data from non-PREDICT external public cohorts used to validate the rankings are available in full at https://doi.org/10.5281/zenodo.17236262.

## Research involving human participants, their data, or biological material

Policy information about studies with human participants or human data. See also policy information about sex, gender (identity/presentation), and sexual orientation and race, ethnicity and racism.

| | |
|---|---|
| Reporting on sex and gender | Sex information (not gender) were self-reported by individuals through the ZOE App with informed consent. Sex was used in the analysis as a covariate as detailed in the Methods. |
| Reporting on race, ethnicity, or other socially relevant groupings | Data on socially relevant variables such as race or ethnicity were not considered in the analyses. |
| Population characteristics | Overall, participants were aged around 50 years old, with about 80% females. |
| Recruitment | Information pertaining to the publicly available datasets used in this work are available from their respective publications: Asnicar F, Nat Med, 2021 (PREDICT 1, https://doi.org/10.1038/s41591-020-01183-8), Bermingham KM, Nat Med, 2024 (METHOD, https://doi.org/10.1038/s41591-024-02951-6), and Creedon AC, bioRxiv, 2024 (BIOME, https://doi.org/10.1101/2024.07.02.24309816). The other ZOE PREDICT studies have the following protocol IDs: METHOD (NCT05273268), BIOME (NCT06231706), PREDICT 2 (NCT03983733), PREDICT 3 US 21 and 22A (IRB Pro00044316), PREDICT3 UK 22A and 23ART (HR-23/24-28300). All PREDICT 3 cohorts refers to the Clinicaltrials.gov identifier: NCT04735835. Participants from the ZOE PREDICT studies paid a private company (ZOE Ldt.) to sequence their microbiomes and filled out informed consent forms that allows the usage of their data by ZOE for scientific purposes. |
| Ethics oversight | All study protocols are registered on clinicaltrials.gov and procedures are compliant with all relevant ethical regulations. Ethical approval for the PREDICT 1 study was obtained in the United Kingdom from the King's College London Research Ethics Committee (REC) and Integrated Research Application System (IRAS 236407), and in the United States from the institutional review board (Partners Healthcare Institutional Review Board (IRB) 2018P002078). Ethical approval for the PREDICT 2 study (Pro00033432) was obtained from Advarra IRB. Ethical approval for the PREDICT 3 study (Pro00044316, HR/DP-21/22-28300 and HR/DP-24/25-45829) was obtained from Advarra IRB and King's College London REC. Ethical approval for the METHOD study (Pro00044316; protocol no. 00044316) was obtained from Advarra IRB. Ethical approval for the BIOME study (HR/DP-23/24-39673) was obtained through King's College London REC. All participants provided written informed consent and all studies were carried out in accordance with the Declaration of Helsinki and Good Clinical Practice. |

Note that full information on the approval of the study protocol must also be provided in the manuscript.

# Field-specific reporting

Please select the one below that is the best fit for your research. If you are not sure, read the appropriate sections before making your selection.

☒ Life sciences ☐ Behavioural & social sciences ☐ Ecological, evolutionary & environmental sciences

For a reference copy of the document with all sections, see nature.com/documents/nr-reporting-summary-flat.pdf

# Life sciences study design

All studies must disclose on these points even when the disclosure is negative.

| | |
|---|---|
| Sample size | No a priori sample size calculation was performed, as we gather and analyzed 34,694 microbiome samples, leveraging the largest (to the best of our knowledge) and most comprehensive microbiome cohorts, providing robust statistical power sufficient for the primary exploratory and hypothesis-generating objectives of this study. |
| Data exclusions | Only one individual from the PREDICT 2 cohort was excluded from all the analysis as it had an empty (no species detected) microbiome profile. |
| Replication | The findings were assessed for reproducibility and validated across multiple independent cohorts, including publicly available external ones. This rigorous validation, achieved through the analysis of distinct datasets spanning different geographies and populations, substantiates the robustness of the defined microbial species ranks. No results/experiments replications were performed. |
| Randomization | Randomization was necessary and done with the machine learning analyses, in which a ten-times, ten-folds cross-validation was performed. |
| Blinding | All microbiome analyses were carried out blinded to group allocation and subject metadata by the primary analyst. Sample extraction and sequencing were performed independently by the processing laboratory, which was also separate from ZOE Ltd. employees. The use of distinct, independent parties for sample processing, sequencing, and final statistical analysis inherently minimized the potential for systematic bias in the analytical pipeline. |

# Reporting for specific materials, systems and methods

We require information from authors about some types of materials, experimental systems and methods used in many studies. Here, indicate whether each material, system or method listed is relevant to your study. If you are not sure if a list item applies to your research, read the appropriate section before selecting a response.

## Materials & experimental systems

| n/a | Involved in the study |
|---|---|
| ☒ | ☐ Antibodies |
| ☒ | ☐ Eukaryotic cell lines |
| ☒ | ☐ Palaeontology and archaeology |
| ☒ | ☐ Animals and other organisms |
| ☒ | ☐ Clinical data |
| ☒ | ☐ Dual use research of concern |
| ☒ | ☐ Plants |

## Methods

| n/a | Involved in the study |
|---|---|
| ☒ | ☐ ChIP-seq |
| ☒ | ☐ Flow cytometry |
| ☒ | ☐ MRI-based neuroimaging |

## Plants

| | |
|---|---|
| Seed stocks | n/a |
| Novel plant genotypes | n/a |
| Authentication | n/a |

