## [Peer Review File · Nature]

Gut microbes associated with health, nutrition and dietary interventions

Corresponding Author: Professor Nicola Segata

Version 1:

Reviewer comments:

Referee #1

(Remarks to the Author)

In this manuscript, Asnicar et. al. conduct an exceptionally large analysis of previously-collected metagenomic fecal samples. Their overarching primary objectives are to 1) identify microbial taxa that are associated and anti-associated with "health" (defined in various ways throughout the manuscript), and 2) use the prevalence and/or abundance of these taxa to derive something like a "health score" to be used in place of standard alpha diversity measures. This is a worthy thing to do, and in general, the authors succeed. Further, by using a massive dataset, the authors show that their results are generalizable. As detailed below, my primary critiques of this manuscript relate to 1) a lack of details in the methods section, and 2) a lack of clarity and rigor in fully demonstrating the generalizable nature of the identified taxa. Should these issues be corrected, I believe this manuscript is suitable for publication in Nature. Specific comments below.

1) At many points throughout the manuscript it is highlighted that health-associated taxa are associated with multiple different measures of human health (diet, personal markers, fasting markers, BMI, etc.). This is generally discussed as evidence that these health-associated taxa are robust to multiple different aspects of human health. However, the authors never discuss or grapple with the extent to which these markers of human health are correlated with each other. For example, if diet is highly correlated with BMI, it is unsurprising that taxa associated with diet are also associated with BMI. I'd like to see the manuscript handle / account for / discuss this in some way.

2) At many points throughout the manuscript it's mentioned that this is a "global microbiome ranking", but in reality it's heavily biased towards western, industrialized populations. Given how different the microbiomes of non-industrialized populations are, I highly doubt these rankings would work with non-industrialized populations. The authors briefly discuss this in the discussion, but I would prefer that this fact is more emphasized throughout the text. To me, it feels a bit demeaning to non-industrialized populations to say that these industrialized-specific rankings are global.

3) In the Figure 1 caption it's stated that "lower sequencing depths do not impact the number of species detected (Wilcoxon $p = 0.031$)". How was this test performed? Wilcoxon seems like the wrong way to do it. It is more appropriate to test if there is a significant relationship between the sequencing depth of a sample and the number of species detected in that sample. Please perform and report the results of that test. It seems fundamentally true that sequencing depth IS correlated with the number of species detected.

4) Substantially more details are needed on how the "species-level genome bins (SGB) approach" was used in this study. Specific questions include: 1) Is each taxa represented by a single genome? If so, are those genomes accessible somehow? If they're not accessible, does this mean that using the specific version and database of metaphlan that was used in this study is the only way to recapitulate the results of this study? 2) In the taxa descriptions in Figure 2a, there are some taxa with very high taxonomic levels (e.g. "p: Firmicutes (GGB9237 SGB14179)"). Does this represent a group, or does this represent an SGB that can only be classified at the phylum level? If it's a genome, why does it have such poor classification? 3) In Figure 2b, why do some SGBs have "_group" attached to their name?

5) The analysis performed and depicted in Figure 3f is really complex. This is a critical benchmark of the method, and I'd like to see a more straightforward benchmark in addition to / instead of this analysis. Specifically, please repeat the analysis presented in Figure 3b-e (which is intuitive and easy to understand) for the additional datasets shown in Figure 3f.

6) It's written that "Healthy-weight individuals carried on average 5.2 more of the 50 favorably ZOE MB Health-ranked SGBs than obese individuals ($p = 0.0003$, Figure 3f and Supplementary Table 8), which corresponded to a normalized difference in the cumulative abundances of unfavorably and favorably-ranked SGBs of -0.59 (Cohen's d , $p < 0.0001$, Supplementary Table 9)". What is the unit on this "-0.59"? I don't know how to interpret this. Additionally, please report the raw differences in cumulative abundance of these as well. That's the most straightforward way to understand this.

7) Please test whether there are differences in the prevalence and abundance of these taxa in healthy individuals between the countries / studies tested in Figure 3f. This is an important test of the generalizability of these taxa as health markers across countries.

8) For the "Microbiome taxonomic profiling" methods section; was any coverage or abundance threshold used to define "presence / absence"? Or was every taxa with at least 1 read mapped defined as present? Was metaphlan run with default settings? Please specify these things in the methods section.

9) The "Machine learning" section does not have nearly enough details to understand what was done. What are the hyperparameters of the random forest model? What code package was used to develop and evaluate the model?

10) I would really like to have the code used to perform the analyses described in the sections "Machine learning" and "Statistical & meta-analyses" made publicly available. These are really complex analyses, and having the code would make understanding the analysis and reproducing the analysis much easier.

Referee #2

(Remarks to the Author)

In this study, the authors investigated microbial associations with cardiometabolic parameters in 34,000 individuals from five PREDICT cohorts. They compiled a list and prioritized the top 50 favorable and unfavorable associated species based on average rankings. The study further demonstrated that these prioritized species were associated with BMI, various types of diseases, and showed responses to dietary intervention. The large sample size is noteworthy, and the dietary intervention provides additional insights beyond mere associations. The main conclusion was that this ranked list is more informative than microbial diversity. However, this information is not novel. Overall, the study lacks excitement in terms of contributing to fundamental knowledge or offering potential for clinical translation. Additionally, there are several serious concerns about the methodology.

The study is overall very descriptive. The entire study was based on a rank list which calculated from the averaged rank of partial Spearman correlation. Later, the authors also used linear regression, presence/absence analysis, and other methods. The choices of methods lack justification and consistency. It is questionable why partial Spearman correlation was used for the first step, while linear regression was used later for instance in meta-analysis of BMI associations.

Second, the authors computed the average rank score of associations across different traits and used these ranks for data visualization and analysis instead of the raw data. They divided their traits into three categories: personal, fasting, and postprandial, with each category containing 7-10 traits. The ranks of associations were averaged for each category per cohort and then further averaged to create the so-called ZOE health rank. What is the theory and assumption behind this average rank? The method would work if the assumption were met that all species show consistent association across different traits within the same category. There are a few concerns: a) the traits can be different, or b) some traits can be highly correlated (e.g., lipid traits). Such average ranking will dilute trait-specific signals and be overruled by the highly correlated traits. The choice of the top 50 favorable and unfavorable species was ambitious and lacked scientific and statistical justification. As part of the most important output of the study, the authors only made the full rank list available upon acceptance, which is unfortunately regrettable.

Many previous studies reported microbiome-based patient stratification model suggested that the combination of species abundance is more predictive than diversity itself. On one side, such a conclusion is not novel, and on the other hand, the conclusion presented in the current study lacks statistical support. The AUC were simply listed in the figure 4 by showing the difference in AUC values. However, it is not clear whether the difference is statistically difference. The author should compute the SD of AUC using bootstrap or other methods and perform statistical analysis on AUC differences. It is also noticed that cardiometabolic health was the main phenotype for association, while the descriptive power for CVD is rather limited (0.57-0.61), even lower than several previous studies (e.g., 0.86 in PMID: 29018189). Note, it is well acknowledged that microbial diversity is not a perfect predictor and has a certain degree of limitations. However, comparing the results to already known good but not best predictors does not necessarily mean that the presented models are best. There lacks a systematic comparison to all other previously predicted microbiome-based prediction models. For instance, the microbiome-based AUC for IBD prediction is often at the range of 0.8-0.9. However, the reported ZOE MB health predictive AUC for IBD showed a large variation in different studies, ranging from 52 to 82. Therefore, it leaves an impression that the robustness and advance of the established ZOE MB values were not thoroughly assessed. Additionally, the meaning of meta-analysis of AUC is unclear. The analysis lacks a clear biological reasoning.

Moreover, the methodology lacks sufficient details. For instance, whether the Spearman correlation was based on the relative abundance level when present only, or on all abundance levels including absence (meaning zeros). The terms "health weighted" and "diet weighted" showed in the Figure 4e lack an explanation. There are also no details on the "personalized dietary advice".

Another main conclusion from the study is the relationship between the gut microbiome and diet. Firstly, the FFQ data differed across the five different PREDICT cohorts. It is unclear how these FFQ differences affect the calculation of the diet index. Secondly, the gut microbiome was associated with both cardiometabolic health and diet. However, the study does not provide direct statistical evidence to support that diet modulates the gut microbiome index for cardiometabolic health, even though the ZOE-selected species showed differential abundance upon dietary intervention.

Minor comments:

- No page number and line number. It makes review report difficult
- Introduce the full name when the abbreviates were introduced at the first time, such as Atherosclerotic cardiovascular disease for ASCVD.
- Figure 2a plot for the relative abundance when present. Does Spearman correlation take absence (or zero) into account. If yes, the information of this plot is not very informative.

Referee #3

(Remarks to the Author)

Peer review of "Gut microbiome species indicative of cardiometabolic health are modulated by diet in large and interventional cohorts of over 34,000 individuals"

The article describes an ambitious effort where five large metagenomic ZOE PREDICT cross-sectional studies totaling over 34,000 individuals from the UK and the US were analyzed to identify microbiome species consistently associated with dietary quality or markers of cardiometabolic health. Within each of 5 cross-sectional studies, the authors computed partial Spearman correlations for each (species, marker) combination within a dietary or health category, ranked these correlations, normalized the ranks to fall between 0 and 1 (by dividing by the relevant number of correlations for a given cohort and health marker) and then averaged these normalized ranks across cohorts and categories for each species, with these species category averages then averaged together to create ZOE MB Diet Rank Scores and ZOE MB Health Rank Scores. They also identified the set of 50 most unfavorable species and of 50 most favorable species based on these rankings. They then assessed the correlation of these rank scores and measures based on the most unfavorable or favorable species with various health outcomes like BMI or disease status, using other publicly available metagenomic cohorts. Further, they demonstrated via meta-analyses that their derived measures compared favorably to widely used alpha-diversity measures in discriminating between diseased and nondiseased individuals. The authors also showed in a ZOE PREDICT longitudinal dietary intervention study that the selected species were appropriately correlated with changes in diet quality and BMI. The results are of interest to microbiome researchers because the species identified as having consistent associations with health measures and dietary quality may be of interest in their own right. Also, the development of derived variables based on these rankings may have appeal for those seeking additional useful and simple summary measures of the microbiome for use in clinical research. Major strengths of the study include the large sample size and the practical and generally sound methods used by the team in conducting and reporting their analysis, including the reporting of very detailed supplementary data.

There are some addressable shortcomings in the analysis and reporting. Also, the overall strength of the associations between the newly derived measures and the health and dietary quality measures is somewhat modest. Further, there is of course residual confounding inherent in observational data, particularly cross-sectional data. Also, the large number of species and markers considered required that the authors use an approach that makes some reasonable simplifications. Nevertheless, this is an important project and a suitably revised manuscript would be of interest to microbiome researchers. Moderate to major concerns:

1. The fourth paragraph of the introduction starts with a sentence describing shortcomings of nutritional intervention studies and then follows that with a second sentence that implies that these shortcomings can be overcome by large-scale comprehensive studies with multi-national populations. That is not correct. In particular, the confounding biases from local lifestyle and nutritional habits is not necessarily going to be removed by combining data from multiple locations and/or increasing the sample size. The authors should revise the second sentence to avoid the suggestion that internal validity threats can be overcome merely by large samples from multiple locations.

2. The second paragraph of the "Results & Discussion" section notes that there were good AUC measures associated with their random forest classification and regression algorithms that used microbiome features to discriminate among patients according to continuous markers. In reading the associated methods, though, one learns that the authors computed the AUC by considering only those patients in the top fourth and the bottom fourth of a continuous marker. Although the authors used that method in a previous publication, it is unacceptable, because it describes a discrimination task of little clinical or research relevance, particularly because it ignores data from the middle half of the analysis sample, and the individuals in this half that are ignored depends on the continuous marker. There are alternatives that can be used in this setting. If the authors considered the top fourth to be important, they can discriminate patients in the top fourth versus those in the bottom three fourths. (Similarly, they could discriminate between the bottom-fourth and those in the top three-fourths.) Another alternative, more clinically relevant, is to use marker specific thresholds that have clinical relevance. A third alternative, in case the authors want to use the ordinal classification into fourths or some other ordinal outcome with more than two levels is to use the asymmetric Somers D measure, the one that is commonly denoted Somers D (regression prediction | ordinal classification of biomarker). For a binary classification, this Somers D can be transformed directly into the AUC, using the formula $AUC = 0.5(\text{abs}(D) + 1)$ [See Newson, R. (2006). Confidence Intervals for Rank Statistics: Somers' D and Extensions. The Stata Journal, 6(3), 309- 334. <https://doi.org/10.1177/1536867X0600600302>.]

I was curious as to how inflated the reported AUCs here could be. Hence, I simulated in SAS a dataset with 20,000 observations, sampling each from a bivariate standard normal distribution with a correlation of 0.40, to make it relevant to reported correlation between predicted and actual values reported here for the better fitting models. I computed the relevant

Somers D for three scenarios and then converted these into C statistics by applying the transformation reported in the above paragraph. The first scenario was to not transform the marker variable; the second was to dichotomize the marker so that it contrasted the top fourth to the bottom three-fourths; the third was to do what the authors did here, contrasting the top fourth with the bottom fourth and ignoring the middle half. The AUCs (95% CI) were 0.63 (0.62, 0.64), 0.69 (0.68, 0.70), and 0.78 (0.77, 0.79), respectively. Hence, it is clear that contrasting the top fourth to the bottom fourth creates a misleading impression of how well a predictor can discriminate patients with respect to a marker. If feasible, the authors should revise their calculation of the AUCs used for this part of the manuscript, replacing it with something that involves the whole sample (e.g. something like scenario 1 or scenario 2) and then update their figures and text. Given that this particular section of the manuscript is mainly just trying to show proof of concept that microbiome data can be used to develop use machine learning and regression models and the authors are also reporting the very relevant Spearman correlation coefficient between real and predicted values of about 0.40, it would be reasonable for the authors to retain their current analysis but add to the second paragraph of the results section a note saying that the Spearman correlation coefficients are of more relevance in describing how well one can discriminate patients with respect to continuous markers using the microbiome, as the AUC was based on discriminating between the top and bottom fourths on the marker, an easier tasks which leads to higher AUC values. A word of computation advice: SAS and Stata appear to be to be much faster at being able to supply Somers D values than the R packages I tried on my simulated data. In SAS, you can use PROC FREQ with options to get Somers D. With Stata, you can download the package created by the author of the article cited in the previous paragraph.

3. Page 5, sentence describing results presented in Supplementary Table 4. To report the constituency of cohort-specific ranks for the 100 selected SGB, the authors say there is a “11.3% average variation”, but that is a misleading description of what should be called a “0.113 average range”, as the average is computed using the range statistics (max – min). The authors should also consider reporting the average coefficient of variation is 38.5%. That can be determined by dividing the standard deviation in each row by the mean in each row and then averaging across the rows.

4. Related to Concern 3, there should be some reporting of the internal consistency/ reliability of the cohort-specific (or country-specific ranking scores) that ultimately are averaged together to form ranking score. For example, the between-country Spearman correlation for the ranking scores for SRGs scored in both countries is only modest. Using the data in Supplemental Table 3, this correlation is 0.61 for the Cardiometabolic ranking score (Columns CMH_UK and CMH_US) and 0.26 for the Diet ranking score (Columns Diet_UK and Diet_US). It might be useful for the authors to note this, as it suggests that the relative ranking of a SRG, especially those SRG ranked in the middle, is likely to change several positions, on average, if the ranking scores were updated with data from new studies. Related to this, the authors should consider computing the within-SRG intracluster correlation coefficient associated with the cohort-specific ranking scores, the ratio of the between-SRG variance component to the sum of this component with the within-SRG between-cohort variance component. This can be estimated by fitting a mixed-effects model with all of the individual SRG-Cohort ranking scores for a given category (e.g. DIET or CMH), with random intercepts for SRG and then forming the ratio of the variance component for SRG with the sum of the between- and within-SRG variance components. (See <https://pubmed.ncbi.nlm.nih.gov/14969463/>)

5. After developing their rankings, the authors compare them for several SRGs. Notably, for most of them, at least one of the rankings being mentioned was in neither the top-50 nor the bottom-50. For example, to be in the top 50 for Health, the ranking score would have to be less than or equal to 0.170 for Health or to 0.188 for Diet. To be in the bottom 10, the ranking score would have to exceed 0.906 for Health or 0.866 for Diet. For SGB 6749, the Health ranking score is 0.267. Near the bottom of Page 7, SGB14838 is reported as having been found to be “associated with health cardiometabolic markers”, citing its ranking score of 0.363. That score does not necessarily demonstrate that SGB14838 was associated with the health markers, as it is not a particularly impressive score. I would suggest the authors stick to the informal descriptions “favorably ranked” or “unfavorably ranked” or else refer to reported results (perhaps in the supplement somewhere I missed) that demonstrates a statistically significant linkage of the SRG with the health or diet marker.

6. Page 14, last sentence in paragraph describing Figure 4e. Instead of “accurate proxy”, which was not assessed with the reported analysis, the authors should say “moderate and useful discriminator”.

Minor concerns

1. First and third paragraphs of “Results & Discussion” section and Figure 1. The present manuscript involves 6 primary studies, five of them cross-sectional and one of them longitudinal, but in some places in the manuscript, the authors refer to only five studies. When this is done, the authors are referring to the five cross-sectional studies. It would help avoid confusion if the authors add some adjectives here and there to make this more clear. Also, the first study, PREDICT 1, was overwhelmingly comprised of UK individuals (n=1,001), but it did have 97 US individuals. So, technically, the second sentence of the first paragraph is incorrect, because it counts these 97 US individuals as being in the UK. (Instead of there being only 21,243 US individuals, there were 97 more; whereas the UK had 97 fewer than 13,451 individuals.) The authors could correct this sentence in a couple of different ways, either by correcting the numbers or changing the description to say they are tallying across the studies belonging to a country and that they consider the PREDICT 1 study to be UK, despite the inclusion of some US individuals.

2. Page 18. Methods section. Paragraph “ZOE PREDICT cohorts definition”. There are only 50 states in the United States, so if the states North Dakota and Hawaii are not represented in the cohort, then that means 48 US states were represented, not 49. A suggested rephrasing would be “975 individuals from 48 US states and the federal District of Columbia (the states not included were North Dakota and Hawaii).”

3. Page 19, "Dietary data processing". There does not seem to be a description of how data were assessed in the intervention study (P3 UK 23A RT). Should there be?

4. Page 21. Commendably, the authors used "partial Spearman's correlation". Presumably, they used a relatively novel technique based on probability-scale residuals. They should provide a reference to either the package they used or a reference to the method (e.g. <https://www.rdocumentation.org/packages/PResiduals/versions/0.2-5/topics/partial.Spearman>)

5. Page 5 and throughout manuscript. The description of the normalized correlation rankings is not quite right. They are not percentiles, because percentages go from 0 to 100, while these go to 1. It might be more clear to say that rankings were normalized to range from 0 to 1 by dividing the relevant sample size.

Referee #4

(Remarks to the Author)

This is an unprecedented study setting a standard state for associations of the human gut microbiome with diet and health markers using metagenomic data from >30 thousand individuals from a commercial health platform. I believe this can be a significant contribution using adequate measures to assess reliable and reproducible associations. While overall, the literature is full with associative studies of the gut microbiome in humans, the sheer scale of this study makes it a unique reference work.

That being said, I have a couple of comments which need to be addressed, specifically regarding the protocol of the The PREDICT 3 UK 23A RT cohort and some clinical aspects.

1. Figure 1: females overrepresented thus this should be reflected as a limitation

2. Quite a drop in associations with the gut microbiome after visceral fat, i.e. Spearman correlations rather weak with ASCVD. This should be discussed with regard to the strong title "cardiometabolic" since it seems to be really driven by anthropometry

3. With regard to anthropometry, it would be helpful to have more clinically assessed data e.g. from PREDICT 1 and PDP to better understand the basic characteristics of the cohorts investigated. I understand that this is not available for all cohorts but could be provided where-ever possible.

4. The authors use the word metadata, while what they really mean are the clinical data derived from the clinical studies. Metadata rather describe data types. Please correct.

5. It seems like a control group is missing for the ZOE PREDICT longitudinal cohort (PDP). This should be clearly stated as a limitation.

6. While the authors highlight the strengths of their investigation, they did not mention any weaknesses. This is unusual and I urge the authors to insert a paragraph clearly stating the weaknesses of their study!

7. The authors should consider to provide a solution such as an online tool for the research community and individuals, where individual or batched datasets with metagenomic shotgun sequencing data can be uploaded (potentially restricted to specific strains to reduce data burden) to allow researchers to compute the rank scores for their purposes

8. Please name the NCT number for the PDP (The PREDICT 3 UK 23A RT cohort). Was this study registered? Is there an ethics vote on a protocol? The protocol should be added to the supplements.

9. How was the BMI in PDP assessed? Self-report?

10. For an overview it would be helpful to have a consort diagram combining the cohorts.

11. Rephrase the title – currently it indicates that the authors investigated how diet modulates the gut microbiome in more than 34,000 individuals. However, this is not the case for most measurements since associations were investigated and mainly not interventions (such as PREDICT 1 and PDP). Thus, that the microbiome was "modulated" by the diet in over 34,000 individuals does not seem correct here.

12. Since this work is strongly supported by ZOE, which is clearly stated, I wonder if this should be reflected in the title even. Such as "...data from a commercial diet intervention" or anything alike. But this is up to the editors to consider.

Version 3:

Reviewer comments:

Referee #1

(Remarks to the Author)

The authors have adequately addressed all my concerns and in my opinion this manuscript is now ready for publication. I also agree with the authors that Rev. Fig. 1 and Rev. Fig. 2 do not need to be added to the manuscript.

Referee #2

(Remarks to the Author)

Thank authors for having carefully clarified their methods. I am satisfied with their response and have no further comments.

Referee #3

(Remarks to the Author)

Peer review of revised manuscript, "Gut microbiome species indicative of cardiometabolic health are modulated by diet in large and interventional cohorts of over 34,000 individuals"

The article describes an ambitious effort where five large metagenomic ZOE PREDICT cross-sectional studies totaling over 34,000 individuals from the UK and the US were analyzed to identify microbiome species consistently associated with dietary quality or markers of cardiometabolic health. I reviewed the original submission and consider this a valuable study worthy of consideration for Nature. I feel the revised manuscript has been made even stronger. However, before it is suitable for publication, there are two deficiencies that I would like to see addressed. One deficiency has to do with a concern I raised in my original review. The second has to do with clarifying the description of the "Rankings definition" section.

1. Please clarify "Rankings Definition" description. From the text, we read that the process begins by computing partial correlations for each SRG and marker available in a cohort. The markers are assigned to categories. However, what is not clear is how the partial correlations are ranked together and normalized to address markers with opposite associations with health (e.g. high-density lipids vs. total triglycerides). I would guess that for each cohort and each marker, the partial correlations were ordered in the appropriate direction and then assigned normalized ranks from 0 to 1, with 0 being favorable to health and 1 being unfavorable to health. Then, for each cohort and SRG, all of the normalized ranks belonging to the same category were averaged together. Then, these cohort-, SRG-, and category averages were averaged together to yield cohort- and SRG-specific averages that were then averaged to yield SRG-specific averages that became the ZOE MB Health (or Dietary) Score. That's what Figure 2 seems to imply. However, the description in lines 739 to 746 is not clear. I believe it could be cleared up by removing some of the unnecessary mentions of "each cohort", such as on line "740" and in the sentence on lines 744 and 745, where this phrase appears twice and seems to raise the possibility that instead of ranking and normalizing the partial correlations by cohort and marker, the ranking and normalizing combined was done by cohort and category, combining together markers, which doesn't make sense because how then would the conflicting sort order for, say, HDL and triglycerides be accounted for? I trust the authors can develop a clearer description of their approach.

2. The second deficiency is more important. It has to do with the fact that the authors are retaining the AUC values that concern the pairwise comparison of the extreme quartiles, which is a concern because those AUC values are inflated and thus subject to misinterpretation because AUCs are effect sizes. That misinterpretation needs to be corrected, as I'll explain. The authors responded to my concerns about this AUC by reporting in the Supplemental Tables two additional AUC measures that are more clinically relevant, one contrasting the highest quartile to the bottom quartiles and the other contrasting the lowest quartile to the top three quartiles. I very much appreciate the authors providing this additional information. The AUCs for these latter tasks are more clinically relevant because in an actual prediction problem, one is unlikely to know that a patient whose status needs to be predicted belongs to the middle-half and should thus be excluded from the prediction task. By including these additional AUCs in the Supplemental Table, a reader is able to see that there is a high correlation among the three different AUC measures being reported. That supports the authors in their effort to prove that there ZOE index scores are correlated with health and dietary markers. However, AUCs are used widely enough that it is fair to assume that many readers are not just interested in the fact that the AUCs show correlation (by being above 0.50) but also by how much the amount the AUC are above 0.50. In particular, an AUC of 0.80 is regarded by many readers as qualitatively different from, say, 0.70. In their interpretation of Figure 1, the authors indeed feature the AUCs exceed 0.80. AUCs that contrast the highest and lowest quartiles are biased upward. Indeed, one can see from the author's supplemental table that they are about 0.07 to 0.09 higher on average, compared to the alternative AUCs reported there. It makes sense that the clinically irrelevant AUC contrasting the top and bottom quartiles would be inflated, because they concern an "easier" task, but I doubt most readers would realize that.

Fortunately, the authors combine their reporting of the AUCs with correlation measures. The correlation measures include everyone in the sample. Also, they do not involve coarsening a continuous marker into fourths, which also leads to an inflation in the AUC compared to what would be obtained by computing the Somers D(predictor|marker) for the uncoarsened marker and transforming that Somers D to the AUC scale, a measure known as the P-K (prediction probability) [see references below

So, here's what I believe would be a more appropriate rigorous and transparent way for the authors to report their results. In Figure 1b, replace the AUC value being reported there with a clinically relevant AUC, which could be the one of the two already available in the Supplemental Table (the authors could even choose either the top fourth vs. the lowest three-fourths or the opposite on a marker-specific basis) or it could be based on the P-K version of the AUC, the one that involves computing the Somers D (predictor|marker) and then transforming it to the 0 to 1 scale by adding one to its absolute value and then multiplying that sum by 0.5

Related to this, the authors should revise the Abstract and the text in the manuscript describing "strong" associations by instead describing them as "moderate to strong" or perhaps "moderately strong". In particular, the AUCs that are more clinically relevant will be of a magnitude more in line with the correlations being reported here and one can see by looking at the standard conversions among correlation, AUCs and Cohen's D values, that the effect sizes being featured in this valuable manuscript are generally between medium and strong. [see, for example, the effect size conversion website, <https://www.escale.site/>]

REFERENCES

Smith, WD, et al. "A MEASURE OF ASSOCIATION FOR ASSESSING PREDICTION ACCURACY THAT IS A GENERALIZATION OF NON-PARAMETRIC ROC AREA." *Statistics in Medicine.*, vol. 15, no. 11, 1996, pp. 1199–215, [https://doi.org/10.1002/\(SICI\)1097-0258\(19960615\)15:11<1199::AID-SIM218>3.0.CO;2-Y](https://doi.org/10.1002/(SICI)1097-0258(19960615)15:11<1199::AID-SIM218>3.0.CO;2-Y)

Van Calster, B., Van Belle, V., Vergouwe, Y. and Steyerberg, E.W. (2012), Discrimination ability of prediction models for ordinal outcomes: Relationships between existing measures and a new measure. *Biom. J.*, 54: 674-685. <https://doi.org/10.1002/bimj.201200026>

Referee #4

(Remarks to the Author)

I want to applaud the authors for the comprehensive revision according to the reviewers' comments. Here are my statements according to the answers to my previous comments:

Comment 1: no further questions

Comment 2: I find it not convincing to base the expression "metabolic health" or "cardiometabolic" solely on Spearman Correlations that are above the threshold of 0.3 of only age, weight, BMI and potentially visceral fat (which has not been cross-validated if I see this correctly), while all other parameters that have been established to assess cardiometabolic health are below the threshold (referring to Suppl. Fig. 2 and new Suppl. Table 1). One could also run a multivariate model and investigate individual parameter estimates. It is matter of current debate in the field and parameters of defining unhealthy obesity (i.e. cardiometabolic risk) have been identified (PMID: 39448862). This current debate is important to overcome rather traditional views that markers of obesity are generally markers of metabolic health and a panel of leading experts in the field has recently addressed this issue (PMID: 39824205). Thus, unless I oversee something here, I urge the authors to change the term cardiometabolic or metabolic health to e.g. anthropometry or anthropometry characteristics. Otherwise please convince me that the associations below the Spearman cut-off of 0.3 reflect clinically relevant associations to use the term cardiometabolic or metabolic health.

Comment 3: no further questions

Comment 4: E.g. in Suppl. Fig. 2 I would refer to the data on the x-axis as clinical data rather than metadata. But this is of course minor.

Comment 5: no further comment

Comment 6: no further comment

Comment 7: no further comment

Comment 8: Please provide the study protocols or at least a link to the source.

Comment 9: no further comment

Comment 10: no further comment

Comment 11: I find the title "Gut microbes linked with metabolic health, nutrition and diet interventions" much more adequate. However, as stated in my Comment 2 above, I am not convinced that the associations shown reflect associations with metabolic health. It rather reflects associations with adiposity or body mass index, since all other parameters reflecting the metabolic state such as blood pressure or HbA1c etc. seemed to exhibit only minor associations if at all. Thus I would exchange the term metabolic health with "body mass index".

Comment 12: I have no strong opinion about this but rather believe this to be an editorial decision.

Referee #5

(Remarks to the Author)

The manuscript is well-written and of large scientific interest. It presents a rich dataset and offers a clear and comprehensive overview of the cross-sectional associations between diet, the gut microbiome, and cardiometabolic health. From a scientific standpoint, it makes a valuable contribution to the growing body of literature exploring the role of the microbiome in human health.

Findings from this study could be relevant for public health strategies to lower the burden of cardiometabolic diseases. However, I do have some concerns regarding the cross-sectional design and the strength of the conclusions drawn from the findings. There is a risk that the public health implications of the study are overstated, particularly the suggestion that targeting the microbiome may be an effective strategy to prevent or reduce cardiometabolic disease. These points are outlined in more detail below.

Public health relevance

The observed associations, if causal, are potentially important from both a nutritional science and public health perspective. By identifying specific gut microbial species linked to dietary patterns and health outcomes, the authors highlight the microbiome as a key mediator in the diet-disease relationship. The "ZOE Microbiome Ranking 2025" offers a novel framework for ranking microbes by cardiometabolic risk and may inform personalized nutritional interventions to improve metabolic health. The study also shows that dietary changes can beneficially alter gut microbiota composition, underscoring its modifiability through nutrition. These findings reinforce the role of diet in preventing cardiometabolic diseases such as cardiovascular disease and type 2 diabetes. The large sample size (>34,000 individuals) adds robustness, and the study supports the potential of precision nutrition based on individual microbiome profiles for effective disease prevention and health promotion. However, the cross-sectional design limits causal inference, reducing the applicability of the findings to dietary guideline development. The latter is usually based on stronger designs (prospective cohort studies and RCTs), because causality is key for effective public health strategies.

Role of gut microbiome

A key concern is that the authors position the gut microbiome as the primary mediator through which diet influences cardiometabolic health, while insufficiently acknowledging the independent effects of diet itself. Although they recognize the cross-sectional design and the potential for reverse causation, they pay limited attention to the possibility that the microbiome largely reflects dietary patterns rather than causally driving health outcomes. Evidence from intervention studies demonstrating that direct, longterm modification of the microbiome improves cardiometabolic health remains scarce. Moreover, global variation in microbiome composition contrasts with the widespread and consistent rise in obesity and cardiometabolic diseases across diverse populations, suggesting that Westernized diets and lifestyles, rather than the

microbiome per se, are the main drivers. Further investigation into factors that disrupt the microbiome, such as antibiotic use, and their long-term cardiometabolic impacts could be valuable (although such studies are inherently challenged by confounding by indication). Overall, the article risks overstating the potential of microbiome-targeted interventions at the individual level, which may divert focus from broader public health strategies, including food reformulation and modifications to the food environment.

In my view, the authors give insufficient attention to these aspects, and I recommend they address this more thoroughly in the Discussion. They may elaborate on this statement in lines 578-580: "One key limitation of our study design is that it does not allow disentangling directly the effect that diet exerts on the microbiome to improve cardiometabolic health from the impact of diet only". They would do well to properly frame this in the Introduction, setting appropriate expectations for the reader.

Version 4:

Reviewer comments:

Referee #3

(Remarks to the Author)

The authors have adequately addressed all my analysis and reporting concerns. In my opinion this manuscript is now ready for publication.

Referee #4

(Remarks to the Author)

I have no further comments and recommend publication.

Referee #5

(Remarks to the Author)

I am satisfied with the given answers and the adjustments in the text. I have no further comments.

Response to the Comments of the Editor and Reviewers

EDITOR

First, you shall see that all 4 reviewers, besides reviewer #2, find the work of general interest, but all of them raise several technical concerns that would need to be addressed in full, in addition to the inclusion of a full description of all the methods used, and full access to the raw and processed sequencing data, the full results underpinning the main claims (such as the ranking list of species) and the code used for all analyses - such transparency is required for the purposes of peer review, and it would be necessary to provide unfettered access to the reviewers in any further rounds of review.

Dear editor, we would like to thank you and the reviewers for the very constructive comments. We carefully revised our original manuscript to address all raised points and we believe that the current version of the manuscript greatly improves the robustness of the results we are showing. In particular, we included two new longitudinal cohorts that are registered clinical trials, METHOD (NCT05273268, <https://doi.org/10.1038/s41591-024-02951-6>) and BIOME (NCT06231706, <https://doi.org/10.1101/2024.07.02.24309816>). In the first cohort, individuals are separated into two arms, a personalized dietary program (PDP, n=177) versus a general dietary advice (control, n=170) to test the efficacy of the personalized dietary advice. In BIOME, individuals were divided into three arms, a prebiotic blend group (n=116), a probiotic group (n=113), and controls (n=120).

Regarding the data availability, we strongly believe it is important that the data is publicly available. It took longer than expected, but now all metagenomic sequences are deposited in ENA at EBI (PREDICT 1: PRJEB39223, PREDICT 2: PRJEB75460, PREDICT 3 US21: PRJEB75462, PREDICT 3 US22A: PRJEB75463, PREDICT 3 UK22A: PRJEB75464, and PREDICT 3 UK23ART: PRJEB75465). In addition, we are providing in Zenodo the taxonomic profile we used in our analysis with the metadata information we were allowed to make publicly available (namely, sex, age, and BMI; <https://doi.org/10.5281/zenodo.15308000>).

We would like to point out that the full lists of ranked species for both the Health and Diet ranks were already available to reviewers, as originally submitted as **Supplementary Table 3** (now renamed **Supplementary Table 5**). In addition, we have now made sure that these ranks are also publicly available on the webpage hosted at ZOE, as per the link provided in the **Data Availability** paragraph (<https://zoe.com/our-science/microbiome-ranking>).

Second, referee #4 mentions the lack of a control arm for the diet interventional trial, and as noted when we discussed the manuscript initially, we had concerns that this trial was not registered and did not adhere to our guidelines for interventional trials (<https://www.nature.com/nm/editorial-policies/clinicalresearch>). Without a control arm and proper registration, we cannot publish the results of this trial, so any revision must exclude these data in their entirety. It seems that the data collected from the other ZOE cohorts were obtained from trials that were registered and adhered to the protocols and reporting requirements required for studies of this type, so the cross-sectional analyses of these data are fine to include, but the unregistered interventional trial needs to be removed. For trials that make claims on the outcome of an intervention (in this case that "following a personalized diet program, favorably ranked species improved and unfavorably ranked species decreased."), an appropriate control cohort is necessary. I should mention too that any claims relating to the impact of diet on the rankings obtained from the cross-sectional analysis should be very cautiously presented, and without unsubstantiated implications for the role of the precise dietary intervention in associations between health markers and the microbiome.

Dear editor, we took seriously the comments by the reviewers about the missing control arm in our original longitudinal cohort. With the revised manuscript, we completely revised the last part of our analysis. We have now included two dietary intervention cohorts that are registered clinical trials, namely METHOD (NCT05273268, <https://doi.org/10.1038/s41591-024-02951-6>) and BIOME (NCT06231706, <https://doi.org/10.1101/2024.07.02.24309816>). We are now able to show that the change in the microbiome we previously observed is reproducible across cohorts and countries (UK and US) and is not present in control arms or even in the arm with the *L. rhamonosus* probiotic supplementation. Finally, to make our final consideration more robust, we used meta-analysis to identify the microbial species increasing and decreasing the most across the three cohorts to show that their ZOE MB Health and Diet ranks are on the more favorable and more unfavorable sides, respectively.

Notice that we did not make any direct claim on the success of the diet interventions. What we say is that dietary interventions (via advice or use of prebiotics) are resulting in an increase of the favourably scored microbes and a decrease of the unfavourably scored ones. As such, and after refining the working, we are suggesting to keep the data on the individuals in the original intervention study. We do believe these additional over 1,000 individuals are strengthening the statistics and the generalizability of the results. We ask the editor to consider our suggestion as reported in the revised manuscript. Nonetheless, these over 1,000 individuals are not strictly necessary to prove our points, as the two clinical trials are providing enough support, so if the editor still believes that these data cannot be presented, we are fine in removing them from the manuscript.

We would appreciate your careful attention to the following:

STATISTICS: When revising your manuscript, you should ensure that any statistical analysis used is sound and that it conforms to Nature's guidelines). A collection of articles explaining the basics of statistical analysis and advice on how to best present it can be found here.

REPRODUCIBILITY: All of the checklists provided with the current submission (Reporting summary, Editorial policy checklist, and Code and software checklist (if applicable)) should be updated to reflect the revisions made and submitted with the revised manuscript.

DATA AND CODE AVAILABILITY STATEMENTS: All original research manuscripts published in Nature Portfolio journals must include a Data availability statement. This statement must make the conditions of access to the “minimum dataset” that is necessary to interpret, verify and extend the research in the article, transparent to readers. This minimum dataset may be provided through deposition in public community/discipline-specific repositories, custom proprietary repositories for certain types of datasets, or general repositories like Figshare, Zenodo and Dryad. Providing large datasets in Supplementary Information is strongly discouraged; the preferred approach is to make data available in repositories. More information on Nature Portfolio’s reporting standards and preparing your Data availability statement can be found here.

For all studies using custom code or mathematical algorithms that are deemed central to the conclusions, a Code availability statement must be included, indicating whether and how the code or algorithm can be accessed, including any restrictions to access. The Code availability statement should be provided as a separate section after the Data availability statement but before the references. Code should be deposited in a DOI-minting repository such as Zenodo, Gigantum or Code Ocean and cited in the reference list. We encourage you to manage subsequent code versions and to use a license approved by the open source initiative. Additional details can be found here.

EXTENDED DATA: Extended Data do not appear in the print version of the paper but are included online within the full-text HTML and at the end of the online PDF. Extended Data are an integral part of the paper, and only data that directly contribute to the main message should be included. All Extended Data must be referred to in the main text, and their legends should be listed sequentially at the end of the main text, not in the Extended Data files. Extended Data should be assembled into a maximum of 10 A4 size, multi-panelled display items, submitted as individual files in .jpg, .tif or .eps format only. They should be of the same quality as print figures, but there are important differences in their formatting. More specific instructions are provided here. If you need to describe complex processes, we encourage you to include a schematic of the main finding as part of the Extended Data to aid readers unfamiliar with the immediate discipline.

SUPPLEMENTARY INFORMATION: Supplementary Information (SI) is online-only, peer-reviewed material that is essential background to the study (e.g., large data sets, more complex methods, and calculations), but which is too large or impractical, or of interest only to a few specialists, to justify inclusion in the print version of the paper (see here for further details).

While SI should not typically contain data figures (any figures additional to those appearing in the main text should be formatted as Extended Data), we require that the raw, uncropped data for gels be presented as an SI figure (see below). Tables may be included in SI, but only if they are unsuitable for formatting as Extended Data (e.g., tables containing large data sets or raw data tables that are best suited to Excel files). If a manuscript has SI, each discrete item of the SI (e.g., videos, tables) must be referred to at an appropriate point in the main manuscript.

SOURCE DATA (GRAPHS): To increase transparency, we strongly encourage you to provide, in spreadsheet form, the data underlying the graphical representations used in figures. In the case of all experiments presenting data from animal models, this is a requirement and is not optional. This is in addition to our well-established data-deposition policy for specific types of experiments and large datasets. Online readers of the manuscript will be able to access the graphical source data directly from the figure legend. Spreadsheets must be submitted in .xls, .xlsx or .csv formats. One file per figure is permitted. If there is a multi-panelled figure, the source data for each panel should be clearly labeled in the file; alternatively the source data for a figure can be included in multiple, clearly labeled sheets within an Excel file. File sizes of up to 30 MB are permitted, but it is expected that the vast majority of graphical source data files will be considerably smaller than this. When submitting these files with your manuscript, you should select the "Source Data" file type and use the title field in the file description tab to indicate the figure(s) to which the source data pertains.

Dear Editor, we carefully check our **Methods** section to ensure that all tools and statistical analyses are clearly explained and detailed, as well as data and code are available. We have now included a **Code Availability** section in which we provide the Python code we used for the meta-analysis. Similarly, Extended Data and Supplementary Information are all referenced in the main manuscript text and come with captions clearly explaining their information.

Referee #1 (Remarks to the Author):

Referee #1 expertise: microbiome, bioinformatics, statistics

In this manuscript, Asnicar et. al. conduct an exceptionally large analysis of previously-collected metagenomic fecal samples. Their overarching primary objectives are to 1) identify microbial taxa that are associated and anti-associated with “health” (defined in various ways throughout the manuscript), and 2) use the prevalence and/or abundance of these taxa to derive something like a “health score” to be used in place of standard alpha diversity measures. This is a worthy thing to do, and in general, the authors succeed. Further, by using a massive dataset, the authors show that their results are generalizable. As detailed below, my primary critiques of this manuscript relate to 1) a lack of details in the methods section, and 2) a lack of clarity and rigor in fully demonstrating the generalizable nature of the identified taxa. Should these issues be corrected, I believe this manuscript is suitable for publication in Nature. Specific comments below.

1) At many points throughout the manuscript it is highlighted that health-associated taxa are associated with multiple different measures of human health (diet, personal markers, fasting markers, BMI, etc.). This is generally discussed as evidence that these health-associated taxa are robust to multiple different aspects of human health. However, the authors never discuss or grapple with the extent to which these markers of human health are correlated with each other. For example, if diet is highly correlated with BMI, it is unsurprising that taxa associated with diet are also associated with BMI. I'd like to see the manuscript handle / account for / discuss this in some way.

We thank the reviewer for bringing this to our attention and agree that several biomarkers of favourable (or unfavourable) cardiometabolic health and of healthy (or unhealthy) diets are correlated and are expected to be correlated. And our selection of the most representative markers to be included in our analysis is also based on this. We, however, missed to provide an assessment of how much the different markers considered in the different categories, defined to rank the species, were correlated with each other. We have now included **Supplementary Table 4**, which reports the pairwise Spearman's correlations of the markers considered in the different PREDICT cohorts, with their assigned category. These data confirm that, for example, fasting and postprandial versions of the same marker tend to strongly correlate with each other. However, since these markers represent distinct biological and metabolic functions and host characteristics and their uncoupling is at the base of clinical consideration (otherwise measuring postprandial markers would have little relevance) we argue that it is important to represent both in an index that is supposed to capture as many host-associated health-related read outs as possible. Similarly, there are also strong negative correlations between markers that intentionally represent opposite types of biomolecular pathways. For example, C-peptides and QUICKI score have a Spearman's $\rho = -0.809$, where the former is a proxy to measure the level of insulin and the latter aims to represent insulin sensitivity.

Overall, we agree with the reviewer that it is important to add the table of correlations to the manuscript, and we also highlight the concept in the following new text added in the **Ranking gut microbiome species linked to cardiometabolic health** of the **Results & Discussion**:

*“To balance the contribution of different markers within cohorts, and to capture distinct biological and metabolic functions and host characteristics, we grouped them into three overarching categories: (i) anthropometric-derived and accessible health-related measures (hereafter called “personal” and including, e.g., ASCVD and blood pressure), (ii) fasting (e.g., GlycA, triglycerides, HDL, cholesterol, and glucose), and (iii) postprandial cardiometabolic markers. Some of these markers represent the same metabolic function over time and showed positive correlations between their fasting and postprandial measurements, while others represent opposite types of the same biomolecular pathways and showed negative correlations among them (**Supplementary Table 4**).”*

And we also updated the last paragraph of the **Conclusions**:

*“Importantly, our microbiome species ranking system proved accurate in reflecting the changes induced by multiple large-scale dietary intervention trials with associated CMH marker improvements (**Figures 4 and 5**). Indeed, the cross-sectional associations were reflected in a significant and substantial increase of health-associated microbiome species and a reduction or depletion of unfavorably ranked species. Many health-associated host markers are correlated as they are the nutritional indicators, and disentangling their direct interactions from those mediated by the microbiome will remain elusive until large-scale microbiome interventions will be possible in humans. In this respect, one key limitation of our study design is that it does not allow disentangling directly the effect that diet exerts on the microbiome to improve cardiometabolic health from the impact of diet only. This is particularly important as diet-based ranks were more dependent on country-related differences compared to cardiometabolic ranks, and further studies should explore food-specific links with gut microbial species and cardiometabolic outcomes in greater detail ⁹⁷, although this would entail designing large scale studies in which both introduction of single foods and alterations of specific microbiome characteristics (e.g. by administration of specific microbiome members) are tested which is currently highly problematic. Nonetheless, the confirmation of the cross-sectional patterns along the diet-microbiome-health axis in both longitudinal intervention trials and a large-scale longitudinal study conducted outside the framework of registered clinical trials, not only increases the intrinsic value of the rankings but confirms that the human gut microbiome can be successfully modulated by dietary intervention and that the effects on the microbiome of such interventions are both predictable and reproducible. By providing the full list of ranked microbial species, this work can be exploited and built upon in future research on microbiome-powered precision nutrition and can be expanded in the future to*

more diverse populations and lifestyles that are currently underrepresented in microbiome, nutritional, and health studies.”

2) At many points throughout the manuscript it's mentioned that this is a “global microbiome ranking”, but in reality it's heavily biased towards western, industrialized populations. Given how different the microbiomes of non-industrialized populations are, I highly doubt these rankings would work with non-industrialized populations. The authors briefly discuss this in the discussion, but I would prefer that this fact is more emphasized throughout the text. To me, it feels a bit demeaning to non-industrialized populations to say that these industrialized-specific rankings are global.

We thank the reviewer for bringing this to our attention, as we completely agree with them and it is a very important point that we overlooked when writing the original text. Indeed, non-Westernized microbiomes are so different (as we and others saw in several papers) that the rankings in our manuscript are not going to be meaningful for non-Westernized populations. One of the main points of our work is that consistent and extensive nutritional, health, and microbiome data in a large population are all necessary to reproducibly enable diet-microbiome-health studies, and unfortunately, at the moment, this is not available for non-Westernized populations.

We have thus removed or changed the wording throughout the text to reflect these considerations.

“These species were organized into two ~~global~~–microbiome rankings, one representative of CMH and another of diet quality, respectively.”

*“Each cohort-level category comprises several different markers, nonetheless, the 15 best and worst ~~globally~~–ranked species showed ~~have~~–consistent associations when considering every marker in each category, with ~~an 11.3% average~~–a 38.5% coefficient of variation, 0.036 index of dispersion, and 0.008 mean absolute deviation, in the cohort-specific ranks (**Figure 2b** and **Supplementary Fig. 5**, and **Supplementary Table 6**).”*

“Altogether, these results suggest that the ~~global~~–cardiometabolic microbial rankings can stratify individuals based on their obesity status regardless of geography.”

“These category averages were then averaged within each cohort and the cohort averages were ~~finally~~–averaged to represent the overall ~~global~~–ranking. SGBs were retained in the overall ranking if they were ranked in at least two different cohorts and this led to a final ranking for ~~a total of 661 SGBs with a global ranking~~.”

“Multiple factors were crucial in the robust definition of the proposed microbiome species ranking systems. First of all, the scale of our combined cohorts with

consistent experimental protocols for metagenomic sequencing and analysis is unprecedented. Second, the geographic diversity spanning all US states and UK regions, while confined to typical Westernized lifestyles and diets, allowed us to overcome local lifestyle-associated microbiome configurations.”

“By providing the full list of ranked microbial species, this work can be exploited and built upon in future research on microbiome-powered precision nutrition and can be expanded in the future to more diverse populations and lifestyles that are currently underrepresented in microbiome, nutritional, and health studies.”

3) In the Figure 1 caption it's stated that "lower sequencing depths do not impact the number of species detected (Wilcoxon $p = 0.031$)". How was this test performed? Wilcoxon seems like the wrong way to do it. It is more appropriate to test if there is a significant relationship between the sequencing depth of a sample and the number of species detected in that sample. Please perform and report the results of that test. It seems fundamentally true that sequencing depth IS correlated with the number of species detected.

The reviewer is correct that, in general, sequencing depth and species richness within samples are correlated and we agree with this. We apologize for the lack of clarity in our original text about this. Our point, however, was a different one; we wanted to emphasize that, despite some differences in sequencing depth *between* cohorts, the total number of detected species between cohorts was similar. For example, the average sequencing depth in P1 was 8.8 Gbp, which yielded a unique number of detected species (i.e., species detected at least once in the cohort) of 2,390. The average sequencing depth of the P3 UK23ART cohort was lower (5.1 Gbps) and with many more individuals, but yielded a similar number of species, namely 2,508. So, our point was about the fact that the integration of different cohorts, despite their differences in sequencing depths, was not affecting our ability to identify a common set of prevalent species over which we then define our rankings. What we compared using the Wilcoxon signed-rank test was the distribution of sequencing depths (**Figure 1a**, middle barplot, left y-axis) against the distribution of the total number of species detected (**Figure 1a**, middle barplot, right y-axis) in each PREDICT cohort. Thus, what we compared was the total number of detected species against the average sequencing depth of each cohort. To avoid confusion, we have removed this, and we have changed the wording in the legend of **Figure 1** as follows to better reflect our intention.

“For each cohort, sample size and the percentage of female participants (% F) are reported in the upper barplots, while sequencing depth (left-most, darker color, average Gbases) and the total number of detected species (right-most, lighter color) are reported in the middle barplots, showing that cohorts with lower sequencing depths do not have fewer total number of detected species.”

In addition, it is important to underline that samples in one cohort are never directly contrasted, correlated, or compared with those in other cohorts, as we always perform cohort-specific analyses that we then generalize via meta-analysis or machine learning, so differences in sequencing depths between cohorts are not relevant from this viewpoint.

For completeness and as requested by the reviewer, we included below the analysis to show how sequencing depth affects the number of species detected in each sample.

Rev. Fig. 1. Relationship between sequencing depth and number of detected species within samples. The scatterplot shows to what extent sequencing depth (depicted here as the number of reads per sample on the x-axis, 'n. reads', log₁₀ scale) correlates with the number of detected species (y-axis, 'n. species') in each microbiome sample for all of the six ZOE PREDICT cohorts.

While sequencing depth affects richness to some extent, it was uncorrelated with specific host factors. Thus, we considered that the benefit of occasionally detecting a few additional low-abundance microbes in samples with higher read counts outweighs the added noise from uneven sequencing depth.

Additionally, as proof of concept that subsampling would not heavily impact our defined ranks, we subsampled all metagenomes of the PREDICT 1 cohort to 25 million reads, re-did the taxonomic profiling, the partial correlations, and the ranks definition.

Rev. Fig. 2. Sequencing depth does not impact the calculation of partial correlation and ranks' definition. The left-most scatterplot shows the relationship between the original (x-axis) partial Spearman's correlations and those calculated on the subsampled microbiome profiles (y-axis). The right-most scatterplot shows the relationship between the ranks defined on the taxonomic profiles from the full set of reads (x-axis) and those computed from the subsampled profiles (y-axis). Overall, there is a high-level of agreement between the full and subsampled profiles. So, we think this justifies our ranks on being defined over the full set of reads, without the need for subsampling.

We modified the **Microbiome taxonomic profiling** section of the **Methods** to report the above considerations, and while we feel the additional figures are not worth adding to the manuscript, but we are open to do so if the reviewers think differently.

“All microbiome samples from the PREDICT cohorts were profiled using MetaPhlAn 4 (version 4.beta.2, database vJan21_CHOCOPhlanSGB_202103), without performing reads subsampling, as the benefit of occasionally detecting a few additional low-abundance species in samples with higher number of read outweigh the potential noise from uneven sequencing depth.”

4) Substantially more details are needed on how the “species-level genome bins (SGB) approach” was used in this study. Specific questions include: 1) Is each taxa represented by a single genome? If so, are those genomes accessible somehow? If they’re not accessible, does this mean that using the specific version and database of metaphlan that was used in this study is the only way to recapitulate the results of this study? 2) In the taxa descriptions in Figure 2a, there are some taxa with very high taxonomic levels (e.g. “p: Firmicutes (GGB9237 SGB14179)”). Does this represent a group, or does this represent an SGB that can only be classified at the phylum level? If it’s a genome, why does it have such poor classification? 3) In Figure 2b, why do some SGBs have “_group” attached to their name?

We thank the reviewer for their thoughtful suggestions, we would like to provide some direct answers to the questions raised.

(1) Is each taxa represented by a single genome?

No, the underlying taxa definition relies on the SGBs (i.e., species-level genome bins) as defined in this publication: <https://doi.org/10.1016/j.cell.2019.01.001> that are comprising from a minimum of 1 reference genome or 5 MAGs and reaches over thousands of genomes and MAGs for a total of 26,970 SGBs.

(2) In the taxa descriptions in Figure 2a, there are some taxa with very high taxonomic levels (e.g. “p: Firmicutes (GGB9237 SGB14179)”). Does this represent a group, or does this represent an SGB that can only be classified at the phylum level?

Many SGBs represent unknown species, which are not described by any reference genome available in public databases. Some of these unknown SGBs are genomically not close to any known genus or family taxonomic label and hence are assigned to their closest phylum label. So, the phylum assignment is not reflecting a poor classification, like could be with 16S analysis, but is, in this case, highlighting the level of “unknownness” of the SGBs.

(3) In Figure 2b, why do some SGBs have “_group” attached to their name?

The “_group” label is a consequence of the process that builds the MetaPhlAn database. Briefly, the “_group” label is used when two or more SGBs are genomically too similar (because of taxonomic issues or particular biological characteristics) and it is not possible to identify SGB-specific marker genes for the taxonomic classification. When this happens, the SGB with the highest

number of genomes and MAGs is selected, and the close and smaller SGB(s) are not considered when assessing the uniqueness of the marker genes. Additionally, these and more in-depth details about the MetaPhlAn 4 algorithm and database can be found in its original publication, which, we believe, can answer all reviewers' doubts: <https://doi.org/10.1038/s41587-023-01688-w>. We have now updated the **Microbiome taxonomic profiling** section of the **Methods**, which is now expanded to incorporate more details about the MetaPhlAn method as per the reviewer's suggestions.

“All microbiome samples from the PREDICT cohorts were profiled using MetaPhlAn 4 (version 4.beta.2, database vJan21_CHOCOPHiAnSGB_202103), without performing reads subsampling, as the benefit of occasionally detecting a few additional low-abundance species in samples with higher number of read outweigh the potential noise from uneven sequencing depth. Samples retrieved from cMD3 described in the ‘Public human microbiome datasets’ section, were profiled with MetaPhlAn 4 (version 4.beta.1, database vJan21_CHOCOPHiAnSGB_202103) using default parameters in both cases (among default parameters, the stat_q is set to 0.2 by default, which defines the quantiles for the robust average coverage calculation), which precludes the necessity for additional prevalence filters considering its default parameters are tailored for the taxonomic profiling of human microbiome samples¹⁰⁶. MetaPhlAn 4 is a publicly available taxonomic profiler for metagenomic samples (Github repository: <https://github.com/biobakery/MetaPhlAn>) that leverages medium and high-quality genomes from isolates and metagenome-assembled genomes (MAGs). Isolate genomes and MAGs are clustered at 95% average nucleotide identity (ANI) to define species-level genome bins (SGBs), as previously described³¹. If an SGB cluster contains a genome isolate, then it is referred to by that isolate taxonomic label. If an SGB only contains MAGs, then it represents an unknown species cluster and is assigned the taxonomic label of a genus, family, or phylum, according to which is the genomically closest with a taxonomic label from isolate genomes. Since the taxonomic classification of MetaPhlAn depends on species-specific marker genes, sometimes there are several SGBs of very closely related genomes for which the identification of SGB-specific markers is not feasible. In this case, more than one SGB can be considered together and the label “_group” is appended to the representative SGB ID. In this way, MetaPhlAn 4 improves the resolution of the taxonomic profiling task¹⁰⁷.”

5) The analysis performed and depicted in Figure 3f is really complex. This is a critical benchmark of the method, and I'd like to see a more straightforward benchmark in addition to / instead of this analysis. Specifically, please repeat the analysis presented in Figure 3b-e (which is intuitive and easy to understand) for the additional datasets shown in Figure 3f.

We thank the reviewer and we agree on the criticality of public data investigation, as well as on the necessity of further assessing the ZOE MB Health and Diet ranks on public data. We also agree on the substantial degree of synthesis of **Figure 3f** (**Figure 3d** in the latest manuscript version), which we believe could have been complemented by a more straightforward investigation of the ranked SGBs distribution in public datasets. We thus replicated the analysis previously presented in **Figure 3b-e** (now **Figure 3b,c** and **Supplementary Figure 9b,c**), in the datasets that were previously only used for meta-analyses. In particular, the new **Supplementary Figure 10a** shows, for 27 public datasets (5,348 samples), the richness and the cumulative abundance distributions of the 50 most favorably ranking and the 50 least favorably ranking SGBs in the ZOE MB Health list by BMI category (healthy-weighted, overweight, obese). **Supplementary Figure 10b** shows, correspondingly, the richness and the cumulative abundance distributions of the 50 most favorably ranking and the 50 least favorably ranking SGBs in the ZOE MB Diet list, divided by BMI category. To validate our findings, we focused on the 50 most favorable and unfavorable microbes from both the ZOE MB Health and Diet ranks on public data. For each public dataset, we compared the SGBs richness and median cumulative abundance between lower and higher BMI categories. Then, we counted the concordant instances between richness and cumulative abundance with respect to BMI categories (e.g., lower BMI had more favorable microbes than higher BMI, or lower BMI had less unfavorable microbes than higher BMI). We then used a binomial test to determine if these patterns were statistically significant against a random expectation.

The amended text parts in the **The ZOE MB Health and Diet rankings stratify individuals based on body mass index** section of the **Results & Discussion** now read:

*“To generalize these associations, we leveraged a total of 5,348 healthy individuals from 27 cohorts and 20 countries divided into three BMI categories, healthy-weight ($n = 2,837$), overweight ($n = 1,562$), and obese ($n = 949$, **Methods, Supplementary Table 9**). In 47 pairwise comparisons, 34 had a higher median richness for the 50 most favorably ranked ZOE MB Health SGBs in lower BMI groups vs higher BMI groups (binomial P value = 0.003, **Supplementary Table 10, Supplementary Fig. 10a**). We assessed both SGB prevalence and cumulative abundance in healthy-weight only individuals (**Supplementary Fig. 11a**), and country had more significant differences than sequencing depth (**Supplementary Table 11**), highlighting the generalization of the identified ranks. Next, we used linear regression, adjusted by sex and age, to predict BMI categories based on the per-sample count of the 50 most favorably and unfavorably ZOE MB Health-ranked SGBs, and then meta-analyzed across cohorts (**Methods**).”*

“Similarly, we tested the association of the 50 most favorably and unfavorably ZOE MB Diet-ranked SGBs with BMI, and found similar but milder signals

compared to the ZOE MB Health ranks (average Spearman's correlations between the two ranks and BMI of 0.61 and 0.72, respectively, **Figure 3a** and **Supplementary Fig. 9a**). Using public datasets, 36 intra-dataset comparisons out of 47 showed a higher median cumulative abundance and a higher median richness of the 50 most favorable SGBs in lower BMI classes compared to higher BMIs (binomial *P* value = 0.0003, **Supplementary Fig. 10b**). Conversely, 36 comparisons showed a higher median count of the least favorable 50 SGBs for the higher BMI classes compared to the lower BMI groups (binomial *P* value = 0.0003, **Supplementary Table 10**). Also for the ZOE MB Diet ranks, among healthy-weight only individuals, we assessed the SGBs prevalence and cumulative relative abundance (**Supplementary Fig. 11b**), and we found that country showed a greater impact than sequencing depth in determining cross-dataset differences (**Supplementary Table 11**). The contribution of diet-ranked SGBs (number of species detected and their cumulative relative abundance) in different BMI categories similarly showed a decreasing number and cumulative relative abundance of favorably-ranked SGBs and increasing of unfavorably-ranked SGBs (**Supplementary Fig. 9d-g**)."

The **Methods** were also expanded to provide all the details about these analyses:

"We performed a meta-analysis to determine the possible links between BMI (categorized into 'healthy-weight', 'overweight', and 'obese') and our ranked SGBs across various publicly-available studies comprising a total of 5,348 individuals, which were not diagnosed for any specific disease. We first evaluated the ZOE MB Health and Diet ranks by assessing the cumulative relative abundance and richness of the 50 most favorable and the 50 least favorable SGBs in each dataset in each BMI category: 'healthy-weight', 'overweight', and 'obese' (see **Public human microbiome datasets** paragraph for the specific cut-offs). Specifically, we assessed the number of intra-dataset, between-BMI groups pairwise comparisons in which the group median abundance or the group median count was higher in the lower BMI group (when considering most favorable SGBs from both ranks) or higher in the higher BMI group (when looking at least favorable SGBs). Next, we fit linear models for each dataset and pair of BMI categories: 'healthy-weight' vs. 'overweight', 'healthy-weight' vs. 'obese', and 'overweight' vs. 'obese'."

Supplementary Fig. 10. The ZOE MB Health and Diet-ranked SGBs stratify individuals according to their BMI in 27 public cohorts. a,b) The number and cumulative relative abundance of the 50 most favorably ranked SGBs from the ZOE MB Health rank (top four panels) and from the ZOE MB Diet rank (bottom four panels) detected in individuals from 27 public datasets, showed that increasing BMI is reflected by a lower presence of favorable SGBs

(top two rows). On the other hand, the 50 most unfavorably-ranked SGBs show an increasing count and cumulative abundance in higher BMI categories (bottom two rows).

6) It's written that "Healthy-weight individuals carried on average 5.2 more of the 50 favorably ZOE MB Health-ranked SGBs than obese individuals ($p = 0.0003$, Figure 3f and Supplementary Table 8), which corresponded to a normalized difference in the cumulative abundances of unfavorably and favorably-ranked SGBs of -0.59 (Cohen's d , $p < 0.0001$, Supplementary Table 9)". What is the unit on this " -0.59 "? I don't know how to interpret this. Additionally, please report the raw differences in cumulative abundance of these as well. That's the most straightforward way to understand this.

We apologize for the lack of clarity in the way we reported our meta-analysis results on BMI. In the revised manuscript, we have clarified the interpretation of Cohen's d values and included the raw differences in SGBs' counts and cumulative abundances.

In particular, Cohen's d is a measure of effect size and has no unit, since it is defined as the standardised difference between two means, i.e., the mean difference normalized by the standard deviation. A value of Cohen's d of 1 (or -1) would mean that the difference between the two means is as big as the pooled standard deviation across the two groups. As originally defined by Cohen (1988), "small" effect sizes correspond to $d = 0.2$, "medium" effect sizes to $d = 0.5$, and "large" ones to $d = 0.8$. We hope that with the clearer order of words, this statistic now better shows that the effect of the cumulative relative abundance of ranked SGBs is moderate to large. We updated the **Statistical & meta-analyses** section of the **Methods** to clarify what Cohen's d represents and how to interpret it:

"Cohen's d was used to estimate the effect size of the normalized difference between unfavorable and favorable ranked SGBs when considering cumulative abundances. This quantifies the difference between the means of two groups in terms of standard deviations. Specifically, as originally defined, a "small" effect size corresponds to $d = 0.2$, a "medium" effect size to $d = 0.5$, and a "large" effect size to $d = 0.8$ ¹⁰⁹."

Regarding the raw differences in cumulative abundance, the new **Supplementary Table 10** reports for both ranks, the median SGBs' richness and cumulative abundance in each BMI category for the public datasets considered. In addition, we made explicit the median differences for both. Of note, we calculated median differences of both richness and cumulative abundance group-wise, as the non-paired study designs couldn't allow us to calculate sample-wise differences.

We amended the text, which now reads as follows:

*"Healthy-weight individuals carried on average 5.2 more of the 50 favorably ZOE MB Health-ranked SGBs than obese individuals (P value = 0.0003 , **Figure 3d** and **Supplementary Table 12**), which corresponded to a normalized difference*

*in the cumulative abundances of unfavorably and favorably ranked SGBs of Cohen's $d = -0.59$ (P value < 0.0001 , **Supplementary Table 10 and 13, Methods).**"*

7) Please test whether there are differences in the prevalence and abundance of these taxa in healthy individuals between the countries / studies tested in Figure 3f. This is an important test of the generalizability of these taxa as health markers across countries.

We thank the reviewer and we agree on the importance of evaluating the distribution of the ZOE MB Health and Diet SGBs' ranks in different countries, focusing on healthy individuals. To do this, we considered cohorts with healthy individuals with a BMI ≥ 18.5 and < 25 . We identified in total 2,837 samples from 26 cohorts and 18 countries. For both ZOE MB Health and Diet ranks, we counted the presence of the 50 most and least favorable SGBs in each cohort, and then organized cohorts by country. These new results are now reported in **Supplementary Figure 10**. Additionally, we computed the cumulative abundance of the 50 most and least favorable ranked SGBs, results reported in **Supplementary Figure 11**.

We next used linear models adjusted by samples' sequencing depth to assess the country's relationship with the SGBs' count or cumulative abundance. We then compared all pairs of countries. Some countries were represented by more than one cohort, and in this case, we used linear mixed models, blocking by dataset. Countries represented by a single dataset were evaluated using ordinary least squares models. These results are presented in **Supplementary Table 11**.

In the main text, these results are now mentioned in the paragraph discussing the relationships with BMI with respect to the ZOE MB Health ranks:

*"In 47 pairwise comparisons, 34 had a higher median richness for the 50 most favorably ranked ZOE MB Health SGBs in lower BMI groups vs higher BMI groups (binomial P value = 0.003, **Supplementary Table 10, Supplementary Fig. 10a**). We assessed both SGB prevalence and cumulative abundance in healthy-weight only individuals (**Supplementary Fig. 11a**), and country had more significant differences than sequencing depth (**Supplementary Table 11**), highlighting the generalization of the identified ranks."*

These results, based on the ZOE MB Diet ranks, are now mentioned in the following sentence:

*"Also for the ZOE MB Diet ranks, among healthy-weight only individuals, we assessed the SGBs prevalence and cumulative relative abundance (**Supplementary Fig. 11b**), and we found that country showed a greater impact than sequencing depth in determining cross-dataset differences (**Supplementary Table 11**)."*

These analyses are now described in the Methods as follows:

*“Further, we assessed the presence of the 50 most favorable and most unfavorable SGBs from both the ZOE MB Health and Diet ranks among the countries considered in these analyses (18 in total) and when considering only the healthy-weight individuals ($n = 2,837$). To link the ranked SGBs with the country, we fit a linear model on the SGBs count and on the SGBs’ cumulative relative abundance, and the models were adjusted by the sequencing depth of the study. We used ordinary least squares adjusted by sequencing depth when comparing two datasets by different countries, and linear mixed model blocked by dataset ID and adjusted by sequencing depth when comparing pairs of countries in which at least one country was represented by more than one dataset (country and the sequencing depth adjusted P values are presented in **Supplementary Table 11**).”*

a

ZOE MB Health Ranking in 2,837 healthy-weight individuals from 26 cohorts

b

ZOE MB Diet Ranking in 2,837 healthy-weight individuals from 26 cohorts

iMSMS_2022
 FengQ_2015
 JieZ_2017
 ZhangX_2015
 XuQ_2021
 QinJ_2012
 RubelMA_2020
 WirbelJ_2018
 MetaCardis_2020
 NielsenHB_2014
 NielsenHB_2014
 MetaCardis_2020
 ZellerG_2014
 XieH_2016
 iMSMS_2022
 DhakanDB_2019
 KeohaneDM_2020
 Zeevid_2015
 DeFilippisF_2019
 Yachidas_2019
 CosteaPI_2017
 LifeLinesDeep_2016
 SchirmerM_2016
 KarlssonFH_2013
 HMP_2012
 iMSMS_2022

ARG
 AUT
 CHN
 CMR
 DEU
 DNK
 ESP
 FRA
 UK
 IND
 IRL
 ISR
 ITA
 JPN
 KAZ
 NLD
 SWE
 USA

Supplementary Fig. 11. The distribution of the ZOE MB Health and Diet-ranked SGBs in 2,837 healthy-weight individuals from 18 countries and 26 public cohorts. a,b) The distributions of the cumulative relative abundance and prevalence of the 50 most favorable (upper) and unfavorable (lower) SGBs in the ZOE MB Health rankings (top panels), showing variable distributions across datasets and countries, and the cumulative relative abundance and prevalence of the 50 most favorable (upper) and unfavorable (lower) SGBs in the ZOE MB Diet rankings (bottom panels). The effect of sequencing depth, which can be associated with datasets and countries for both analyses is assessed in **Supplementary Table 10**.

8) For the “Microbiome taxonomic profiling” methods section; was any coverage or abundance threshold used to define “presence / absence”? Or was every taxa with at least 1 read mapped defined as present? Was metaphlan run with default settings? Please specify these things in the methods section.

We thank the reviewer for pointing out the lack of clarity about the taxonomic profiling analysis of the microbiomes. We expanded the **Microbiome taxonomic profiling** section of the **Methods**, also in response to an earlier comment. In doing so, we also encourage the reader to refer to the MetaPhlAn 4 publication, which covers in great detail any additional aspects that are beyond the scope of this manuscript’s **Methods** section. In the text, we have now specified that default parameters were used, which means that applying any presence/abundance threshold is not necessary since MetaPhlAn 4 is robust enough to not call a taxon on assignment of a single read (or in general when not enough “evidence” for the presence is available). Worth noting that among the default parameters, MetaPhlAn uses the *stat_q*, set to 0.2 by default, for the robust average coverage calculation, which basically means that at least 20% of SGB-specific markers should attract at least one read for the SGB to be detected as “present”. The details in the **Methods** now read as follows:

*“All microbiome samples from the PREDICT cohorts were profiled using MetaPhlAn 4 (version 4.beta.2, database vJan21_CHOCOPhIAnSGB_202103), without performing reads subsampling, as the benefit of occasionally detecting a few additional low-abundance species in samples with higher number of read outweigh the potential noise from uneven sequencing depth. Samples retrieved from cMD3 described in the ‘Public human microbiome datasets’ section, were profiled with MetaPhlAn 4 (version 4.beta.1, database vJan21_CHOCOPhIAnSGB_202103) using default parameters in both cases (among default parameters, the *stat_q* is set to 0.2 by default, which defines the quantiles for the robust average coverage calculation), which precludes the necessity for additional prevalence filters considering its default parameters are tailored for the taxonomic profiling of human microbiome samples¹⁰⁴. MetaPhlAn 4 is a publicly available taxonomic profiler for metagenomic samples (Github repository: <https://github.com/biobakery/MetaPhlAn>) that leverages medium and high-quality genomes from isolates and metagenome-assembled genomes (MAGs). Isolate genomes and MAGs are clustered at 95% average nucleotide*

identity (ANI) to define species-level genome bins (SGBs), as previously described³¹. If an SGB cluster contains a genome isolate, then it is referred to by that isolate taxonomic label. If an SGB only contains MAGs, then it represents an unknown species cluster and is assigned the taxonomic label of a genus, family, or phylum, according to which is the genomically closest with a taxonomic label from isolate genomes. Since the taxonomic classification of MetaPhlAn depends on species-specific marker genes, sometimes there are several SGBs of very closely related genomes for which the identification of SGB-specific markers is not feasible. In this case, more than one SGB can be considered together and the label “_group” is appended to the representative SGB ID. In this way, MetaPhlAn 4 improves the resolution of the taxonomic profiling task¹⁰⁵.”

9) The “Machine learning” section does not have nearly enough details to understand what was done. What are the hyperparameters of the random forest model? What code package was used to develop and evaluate the model?

We thank the reviewer for highlighting the lack of clarity. We expanded the **Machine learning** section of the **Methods**.

“To assess the link between the human gut microbiome composition, we developed and used a machine learning (ML) framework based on random forest (RF) classification and regression algorithms from the scikit-learn Python package (as implemented in the RandomForestClassifier and RandomForestRegressor functions, respectively), both with ‘n_estimators=1000’ and ‘max_features=sqrt’ parameters¹⁰⁸. We trained RF classifiers and regressors on MetaPhlAn 4 estimated SGB-level relative abundances (arcsine square-root transformed) to assess the extent to which the outcome variable was predictable from the microbiome as a proxy of the strength of the microbiome-variable association. This framework was originally used and described in¹⁴ and accounts for the presence of twin pairs in the data, which avoids biases due to identical values in twins. Briefly, the framework employs a cross-validation (CV) approach, randomly splitting the dataset into training and testing folds with an 80/20 ratio, respectively, and repeated 100 times (as implemented in the StratifiedShuffleSplit function). Folds are also constructed to maintain a similar ratio of the two classes to predict as they appear in the full data. For target variables with continuous values, classification was performed contrasting the first against the fourth quartile. Performances were evaluated using the area under the receiver operating characteristic curve (AUC) for the classification task, while Spearman’s correlation between the real and predicted values was used for the regression task⁴⁶.”

10) I would really like to have the code used to perform the analyses described in the sections “Machine learning” and “Statistical & meta-analyses” made publicly available. These are really

complex analyses, and having the code would make understanding the analysis and reproducing the analysis much easier.

We thank the reviewer for this comment. We have now included the **Code Availability** section, in which we specify and make the code we used for the meta-analyses available (https://github.com/SegataLab/inverse_var_weight).

“Code Availability

The custom Python code developed for the meta-analyses performed on public data and included in this work is available in the Github repository: https://github.com/SegataLab/inverse_var_weight. The MetaPhlAn code for the taxonomic profiling is available in the Github repository: <https://github.com/biobakery/MetaPhlAn> and via bioconda at <https://bioconda.github.io/recipes/metaphlan/README.html>.”

Moreover, we also expand the **Methods** section to provide more details about libraries versions and parameters of the different tools employed. The updated text for the **Machine learning** section is provided in our previous answer and not repeated here. Below is reported the text of the **Statistical and meta-analyses** section:

“Statistical and meta-analyses

*We performed a meta-analysis to determine the possible links between BMI (categorized into ‘healthy-weight’, ‘overweight’, and ‘obese’) and our ranked SGBs across various publicly-available studies comprising a total of 5,348 individuals, who were not diagnosed with any specific disease. We first evaluated the ZOE MB Health and Diet ranks by assessing the cumulative relative abundance and richness of the 50 most favorable and the 50 least favorable SGBs in each dataset in each BMI category: ‘healthy-weight’, ‘overweight’, and ‘obese’ (see **Public human microbiome datasets** paragraph for the specific cut-offs). Specifically, we assessed the number of intra-dataset, between-BMI groups pairwise comparisons in which the group median abundance or the group median count was higher in the lower BMI group (when considering most favorable SGBs from both ranks) or higher in the higher BMI group (when looking at least favorable SGBs). Next, we fit linear models for each dataset and pair of BMI categories: ‘healthy-weight’ vs. ‘overweight’, ‘healthy-weight’ vs. ‘obese’, and ‘overweight’ vs. ‘obese’. In the first model, we looked at the count of the 50 most favorable and unfavorable ZOE MB Health and Diet-ranked SGBs. A second model was fitted on the cumulative relative abundance (arcsine square-root transformed) of the 50 most favorable and unfavorable SGBs in the two rankings. All models were adjusted by sex and age. Cohen’s *d* was used to estimate the effect size of the normalized difference between unfavorable and favorable ranked SGBs when considering cumulative abundances. This quantifies the difference between the means of two groups in terms of standard deviations. Specifically, as originally defined, a “small” effect size corresponds to $d = 0.2$, a “medium” effect size to $d = 0.5$, and a “large” effect size to $d = 0.8$ ¹⁰⁹. In these models, the lower BMI category of each comparison was used as the negative*

control, so negative coefficients reflect a higher count of SGBs in the lower BMI category, while positive coefficients reflect a higher count of SGBs in the higher BMI category. Effect sizes were summarized via meta-analysis, computed as a random effect model using the Paule-Mandel heterogeneity on adjusted mean differences from the linear regression models (standardized for cumulative abundances). Further, we assessed the presence of the 50 most favorable and most unfavorable SGBs from both the ZOE MB Health and Diet ranks among the countries considered in these analyses (18 in total) and when considering only the healthy-weight individuals ($n = 2,837$). To link the ranked SGBs with the country, we fit a linear model on the SGBs count and on the SGBs' cumulative relative abundance, and the models were adjusted by the sequencing depth of the study. We used ordinary least squares adjusted by sequencing depth when comparing two datasets by different countries, and linear mixed model blocked by dataset ID and adjusted by sequencing depth when comparing pairs of countries in which at least one country was represented by more than one dataset (country and the sequencing depth adjusted P values are presented in **Supplementary Table 11**).

A second meta-analysis tested the associations between our ZOE MB Health and Diet-ranked SGBs and five gut-associated diseases (CVD, T2D, IBD, CRC, and IGT) across studies, for a total of 4,816 samples (see "Public human microbiome datasets" section). Linear models were used to predict the binary disease outcome (healthy vs diseased) for each disease, using the cumulative abundances (arcsine square-root transformed) of the 50 most favorable or unfavorable SGBs, adjusting by sex, age, and BMI. The betas of the linear models were converted into standardized mean differences as previously described¹¹⁰. We also defined models to predict healthy vs diseased using the sum of the SGBs' ranks normalized between -1 and 1, considering all 661 SGBs for the ZOE MB Health and Diet ranks, once using the direct sum of the SGBs' ranks and once weighting ranks by the relative abundance of each SGB in each sample (transformed using the arcsine and square-root function to avoid overestimating the ranks of highly abundant species due to compositionality). Standardized mean differences were calculated similarly as in the previous case. In all meta-analytical models, the set of cohorts considered comprised studies encompassing multiple diseases with a shared control group that we analyzed separately. To account for the overlaps in the studies considered, we computed weights based on the inverse effect-sizes variance-covariance matrix, as previously suggested^{111,112}. Thus, five meta-analyses were performed, one for each disease: CVD (three datasets), T2D (six datasets), IBD (three datasets), CRC (10 datasets), and IGT (two datasets). Of note, in the comparisons of controls vs T2D, IGT, and CVD, the MetaCardis French and German sub-cohorts were considered as different datasets, and so they were their controls that were meta-analyzed as different cohorts. In particular, only French control samples were used in the CVD analysis, which included only French cases. Finally, meta-analysis summaries were computed using the same technique.

*The meta-analysis presented in **Figure 5a** was performed using a fixed effect with inverse variance testing the 300 SGBs that showed a statistically significant change in at least one of the three longitudinal cohorts (BIOME Q value < 0.01, METHOD Q value < 0.1, and P3-UK23ART P value < 1.255e-05, **Figure 4a-c**). The individual cohort values included in this meta-analysis are reported in **Supplementary Table 25.***

Referee #2 (Remarks to the Author):

Referee #2 expertise: microbiome meta-analyses, machine-learning, biostatistics

We thank the reviewer for taking the time to assess our work and engage in scientific discussion with us. We have addressed all points in a sentence-by-sentence manner.

In this study, the authors investigated microbial associations with cardiometabolic parameters in 34,000 individuals from five PREDICT cohorts. They compiled a list and prioritized the top 50 favorable and unfavorable associated species based on average rankings. The study further demonstrated that these prioritized species were associated with BMI, various types of diseases, and showed responses to dietary intervention. The large sample size is noteworthy, and the dietary intervention provides additional insights beyond mere associations. The main conclusion was that this ranked list is more informative than microbial diversity. However, this information is not novel. Overall, the study lacks excitement in terms of contributing to fundamental knowledge or offering potential for clinical translation. Additionally, there are several serious concerns about the methodology.

We understand the reviewer's point of view and acknowledge that we used alpha diversity to assess part of our work. However, we did not want to convey that the main conclusion was that our microbiome rankings are better than alpha diversity. We worked on this revision to make it clearer.

The main finding was to uncover a set of microbes associated with better or worse diets and cardiometabolic health across populations and countries. The focus was thus on uncovering single microbial species of relevance, and it was not our intention to find a way to "score" a microbiome in a way that is more informative than alpha diversity. The fact that we derived a sample-specific score from our rankings and compared it with alpha diversity is just one of the ways we employed to show that the overall framework was consistent and accurate.

Throughout the manuscript, we have also expanded the discussion about why the favorably-ranked SGBs could indeed be considered favorable species and vice versa for the unfavorable ones. Below are reported the pieces of text where this is discussed in the **Ranking gut microbes associated with healthier diets** section:

"For example, R. torques (SGB 4608) and F. plautii (SGB 15132) were linked to multiple cardiometabolic conditions and were concordantly associated with both unfavorable ZOE MB Health and Diet ranks (0.991-0.904 and 0.981-0.901, respectively) in the current study. On the other hand, the favorably-ranked Blautia glucerasea (SGB 4816) was described elsewhere to hinder visceral fat accumulation and blood glucose and triglycerides in mice⁷⁴ (ZOE MB health and Diet-ranks of 0.267 and 0.062, respectively). Clostridium saccharogumia (SGB 6749, recently renamed Thomasclavelia saccharogumia^{75,76}) is another species displaying specific health-promoting phenotypes that we linked in this study to

beneficial dietary and cardiometabolic markers (0.192 ZOE MB Diet-rank and 0.182 ZOE MB Health-rank). The C. saccharogumia species can catalyze lignans derived from plant foods, otherwise recalcitrant to host and bacterial metabolism, into phytoestrogens with strong antioxidant activities⁷⁷⁻⁷⁹. Furthermore, the favorable ZOE MB Health and Diet ranks of F. prausnitzii (SGB 15317) and Lachnospira eligens (SGB 5082; 0.059-0.188 and 0.276-0.115, respectively) mirror their established beneficial effects on the human host. Indeed, both species display strong anti-inflammatory properties via the release of short-chain fatty acids (SCFAs) and other bioactive metabolites^{50,80} and were previously correlated to healthy dietary habits^{14,81,82}. Additionally, in a dietary fiber supplementation trial involving T2D patients, L. eligens was selectively increased and negatively associated with postprandial glucose and insulin, body weight, and waist circumference⁸³, indicating that precise dietary interventions aimed at stimulating beneficial bacterial growth have already been proven as effective strategies in preventing and alleviating metabolic disorders symptoms.”

In the **Dietary interventions improve favorably ranked species** section:

*“Among the many species with a significant change found in the prebiotic blend arm, there are the beneficial fibre-degrading Bifidobacterium adolescentis (SGB 17244), Bifidobacterium longum (SGB 17248), and Blautia obeum (SGB 4811)⁸⁹⁻⁹¹, as well as the butyrate-producing Agathobaculum butyriciproducens (SGB 14993), Anaerobutyricum hallii (SGB 4532), and Coprococcus catus (SGB 4670)^{92,93}. In contrast, the species Dysosmobacter welbionis (SGB 15078), among the top unfavourably associated SGBs in our study, was significantly decreased by the same dietary intervention (**Supplementary Table 25**).”*

“Of note, the prominent butyrate producers Roseburia hominis (SGB 4936) and A. butyriciproducens (SGB 14993) were also found increased in the PDP intervention.”

The clinical translation thus lies on the hand, as it offers future research a starting point to assess how favorable or unfavorable an individual’s microbiome is with respect to cardiometabolic health or diet. Moreover, these microbes could be targeted in a clinical setting to further elucidate their contribution to health via, e.g., differential gene expression or solo or within-community dynamics. Future research and clinical/nutritional translation will be plentiful and diverse.

Nonetheless, to the point of alpha diversity, the main reason for comparing with alpha diversity is that it is one of the few popular metrics that can summarize one’s entire microbiome as a single number. There are no other current metrics to evaluate the score of single microbes. Considering that the message that the reviewer received from our original submission was the comparison with alpha diversity, which was not our intention to be the main point, we have now revised the full text to lower the weight we put on the comparison against alpha diversity. In the same direction, we also included more

analyses on the public cohorts we gathered to show how our ranked SGBs are distributed, and based on the reviewers' comments, we have now put more value on the longitudinal analysis by including two registered clinical trials, showing how our newly defined ranks perform.

The study is overall very descriptive. The entire study was based on a rank list which calculated from the averaged rank of partial Spearman correlation. Later, the authors also used linear regression, presence/absence analysis, and other methods. The choices of methods lack justification and consistency. It is questionable why partial Spearman correlation was used for the first step, while linear regression was used later for instance in meta-analysis of BMI associations.

We thank the reviewer for their comment and we believe we understand where the confusion comes from and hope to clear up any misunderstandings by improving the clarity of the text.

Partial Spearman correlations were used only to generate the SGBs rankings, and we think this relatively simple, easy-to-interpret, and direct approach is a strength of our study (as also commended by Reviewer #3). To further motivate the choice of partial correlation for deriving the SGBs' rankings, we wanted to use the simplest way to link the prevalent species in each cohort to the different data representing either cardiometabolic health or diet. Considering the known links between the microbiome with age, sex, and BMI, the choice of using partial correlations enabled us to correct for these covariates, instead of using simple correlation coefficients. Moreover, we decided to use Spearman's correlation coefficient, instead of Pearson's, because it is based on the ranks (instead of the absolute values) of the variables, making the Spearman's coefficient less sensitive to outlier values.

The meta-analysis, which does not generate the rankings and whose purpose was instead to integrate the findings and rankings from different studies, relies, by virtue of it being a meta-analysis, on linear regression (correlation cannot be used to summarize correlations from different datasets in a meta-analysis setting). We would like to specify further that linear regression in the meta-analysis was only used to derive the summary trend, which in our manuscript appears in **Figures 3d-f** (last line in each panel) and **5a**, and was labeled as "Meta-analysis", "Fixed effect", or "Random effect".

Thus, we used what can be considered the simplest and most standard statistical methods to derive the ranking and to summarize them in a meta-analysis. This is now also better explained in the **Results & Discussion**:

*"To generalize these associations, we leveraged a total of 5,348 healthy individuals from 27 cohorts and 20 countries divided into three BMI categories, healthy-weight ($n = 2,837$), overweight ($n = 1,562$), and obese ($n = 949$, **Methods, Supplementary Table 9**). In 47 pairwise comparisons, 34 had a*

*higher median richness for the 50 most favorably ranked ZOE MB Health SGBs in lower BMI groups vs higher BMI groups (binomial P value = 0.003, **Supplementary Table 10, Supplementary Fig. 10a**). We assessed both SGB prevalence and cumulative abundance in healthy-weight only individuals (**Supplementary Fig. 11a**), and country had more significant differences than sequencing depth (**Supplementary Table 11**), highlighting the generalization of the identified ranks. Next, we used linear regression, adjusted by sex and age, to predict BMI categories based on the per-sample count of the 50 most favorably and unfavorably ZOE MB Health-ranked SGBs, and then meta-analyzed across cohorts (**Methods**)."*

And in the **Methods**:

"We performed a meta-analysis to determine the possible links between BMI (categorized into 'healthy-weight', 'overweight', and 'obese') and our ranked SGBs across various publicly-available studies comprising a total of 5,348 individuals, which were not diagnosed for any specific disease. We first evaluated the ZOE MB Health and Diet ranks by assessing the cumulative relative abundance and richness of the 50 most favorable and the 50 least favorable SGBs in each dataset in each BMI category: 'healthy-weight', 'overweight', and 'obese' (see **Public human microbiome datasets** paragraph for the specific cut-offs). Specifically, we assessed the number of intra-dataset, between-BMI groups pairwise comparisons in which the group median abundance or the group median count was higher in the lower BMI group (when considering most favorable SGBs from both ranks) or higher in the higher BMI group (when looking at least favorable SGBs). Next, we fit linear models for each dataset and pair of BMI categories: 'healthy-weight' vs. 'overweight', 'healthy-weight' vs. 'obese', and 'overweight' vs. 'obese'. In the first model, we looked at the count of the 50 most favorable and unfavorable ZOE MB Health and Diet-ranked SGBs. A second model was fitted on the cumulative relative abundance (arcsine square-root transformed) of the 50 most favorable and unfavorable SGBs in the two rankings. All models were adjusted by sex and age. When considering cumulative abundances, Cohen's d was used to estimate the effect size of the normalized difference between unfavorable and favorable ranked SGBs. This quantifies the difference between the means of two groups in terms of standard deviations. Specifically, as originally defined, a "small" effect size corresponds to $d = 0.2$, a "medium" effect size to $d = 0.5$, and a "large" effect size to $d = 0.8$ ¹⁰⁷. In these models, the lower BMI category of each comparison was used as the negative control, so negative coefficients reflect a higher count of SGBs in the lower BMI category, while positive coefficients reflect a higher count of SGBs in the higher BMI category. Effect sizes were summarized via meta-analysis, computed as a random effect model using the Paule-Mandel heterogeneity on adjusted mean differences from the linear regression models (standardized in the case of cumulative abundances)."

Second, the authors computed the average rank score of associations across different traits and used these ranks for data visualization and analysis instead of the raw data. They divided their traits into three categories: personal, fasting, and postprandial, with each category containing 7-10 traits. The ranks of associations were averaged for each category per cohort and then further averaged to create the so-called ZOE health rank. What is the theory and assumption behind this average rank? The method would work if the assumption were met that all species show consistent association across different traits within the same category. There are a few concerns: a) the traits can be different, or b) some traits can be highly correlated (e.g., lipid traits). Such average ranking will dilute trait-specific signals and be overruled by the highly correlated traits. The choice of the top 50 favorable and unfavorable species was ambitious and lacked scientific and statistical justification. As part of the most important output of the study, the authors only made the full rank list available upon acceptance, which is unfortunately regrettable.

We thank the reviewer for their comments. We created the categories of personal, fasting, and postprandial, on purpose, since each category incorporates a different number of individual traits. In this way, we avoided the average ranks at the cohort level being driven by one category (the one with the largest number of traits). Instead, each category had the same weight when computing the averages at the cohort level. While there might be differences between the correlations with traits, our aim was not to highlight differences but to find similarities among similar traits and across different categories. That way, we wanted to ensure as much as possible that our correlations are robust in the face of various phenotypic characteristics and do not rely heavily on one trait or another. The assumption is that individuals are diverse and no single trait can accurately define a person, thus, we wanted to incorporate various traits.

As also pointed out by the previous reviewer, it is true that some traits considered were highly correlated, and we have now included these results in **Supplementary Table 4**, which reports the within metadata correlations. For most of the highly correlated traits (either below -0.5 or above 0.5 Spearman's correlation), they will belong to the same category: 27 for personal, 50 for fasting, 14 for postprandial, and 12 for habitual diet. Fewer were found between categories, 7 for the personal-fasting and 27 for the fasting-postprandial. This further supports our definition of the categories to avoid the ranks to reflect only some of the traits, while instead capturing distinct biological and metabolic functions and host's characteristics.

We would highlight that in several analyses (i.e., **Figs. 3a, 4g-i, 5, Supplementary Figs. 8, 9a, and 15f**) we always considered all ranked SGBs, while in only some other analyses we decided to focus on the top 50 most favorable and unfavorable ones. Nonetheless, the choice of partially focusing on these top 50 species was based on several considerations. First, mid-ranked SGBs may not exhibit strong associations, making them less interesting, whereas the extremes, by the definition of the ranks themselves, should capture most biologically relevant signals. Second, we needed a sufficient number of species to ensure good representations in public microbiome

cohorts. Finally, we needed a manageable number of SGBs to focus on and to avoid promoting the ranks as a single score to represent the microbiome (like alpha diversity), as we believe that the complexity of the microbiome is difficult to summarize into one single value. Hence, we decided to focus on the top 50 from both ends of our ranks, as they should represent the most favorable and unfavorable species. While we indeed think that focusing on the most correlated SGBs is important for the mentioned prioritization reasons, we of course acknowledge that the choice of the number "50" is to some extent arbitrary and we now report it in the manuscript (see below). To reflect these rationales for the selection of the 50 most favorably and unfavorably ranked SGBs, we amended the text as follows:

“To focus on the SGBs most strongly linked to CMH, we selected the 50 most favorably and unfavorably ranked, ensuring broad representation across individuals while avoiding mid-ranked SGBs, which may show weaker associations.”

Finally, we would like to underline that the full lists of ranked SGBs, with their specific cohort and country ranks, for both the cardiometabolic and dietary metadata, were provided as **Supplementary Table 3** in the original version of the manuscript. What we provide upon acceptance is all the metagenomic data (all metagenomes now uploaded to ENA/EBI, accessions listed in the **Data Availability** paragraph) and the web page hosted by ZOE (<https://zoe.com/our-science/microbiome-ranking>), which will provide the current version, as well as updated versions of the rankings, as in the future they might be defined over a larger set of individuals, considering different health markers and countries. We apologize that this was unclear, but the full list of ranked SGBs was available and included in the supplementary material (previously **Supplementary Table 3**, now **Supplementary Table 5**), which will also be open access for readers.

Many previous studies reported microbiome-based patient stratification model suggested that the combination of species abundance is more predictive than diversity itself. On one side, such a conclusion is not novel, and on the other hand, the conclusion presented in the current study lacks statistical support.

We completely agree that species abundance information (and indirectly species presence) has more predictive power than alpha diversity, as was shown in numerous studies. This is why, for example, machine learning classifiers trained on species composition to link microbiome composition with host phenotypes (e.g., disease) outperform models based only on alpha diversity. We have previously shown and exploited this in multiple studies, including the one of PREDICT1 (<https://doi.org/10.1038/s41591-020-01183-8>) as well as in contexts distinct from diet or cardiometabolic health, such as colorectal cancer (<https://doi.org/10.1038/s41591-019-0405-7>), response to immunotherapy (<https://doi.org/10.1038/s41591-022-01695-5>), and fecal microbiome transplantation (<https://doi.org/10.1038/s41591-022-01964-3>). We also recently wrote an in-depth review

on this topic (<https://doi.org/10.1038/s41579-023-00984-1>). However, in our study, the focus was on the individual species associated with diet and cardiometabolic health, it was not about a new scoring system for single microbiomes. The comparison with alpha diversity wanted to show that when summarizing all ranks into a single index for a specific sample, the rank-based index was more informative than alpha diversity. Machine learning could not be directly compared here as an ML model is usually trained on the whole set of species and not just on single ranks or values (as for the number of species for alpha diversity). In other words, to support the validity of our newly derived species-level rankings, one of the multiple comparisons we adopted was to show that our derived 1-dimensional index was more predictive of host health in external datasets than alpha diversity, the most commonly used 1-dimensional index, but not that our score should be used in clinical settings instead of more refined models. We apologize if this intent was not clear in the manuscript and we have revised the text accordingly to clarify our intent.

“While the ranking-based scoring of single samples cannot have the same predictive power for host phenotypes compared to condition-specific supervised learning approaches relying directly on labeled training data, our results showed how embedding the ranking system into a simple one-dimensional microbiome index provides a meaningful evaluation of microbiome health conditions.”

The AUC were simply listed in the figure 4 by showing the difference in AUC values. However, it is not clear whether the difference is statistically difference. The author should compute the SD of AUC using bootstrap or other methods and perform statistical analysis on AUC differences. It is also noticed that cardiometabolic health was the main phenotype for association, while the descriptive power for CVD is rather limited (0.57-0.61), even lower than several previous studies (e.g., 0.86 in PMID: 29018189). Note, it is well acknowledged that microbial diversity is not a perfect predictor and has a certain degree of limitations. However, comparing the results to already known good but not best predictors does not necessarily mean that the presented models are best. There lacks a systematic comparison to all other previously predicted microbiome-based prediction models. For instance, the microbiome-based AUC for IBD prediction is often at the range of 0.8-0.9. However, the reported ZOE MB health predictive AUC for IBD showed a large variation in different studies, ranging from 52 to 82. Therefore, it leaves an impression that the robustness and advance of the established ZOE MB values were not thoroughly assessed.

We thank the reviewer for pointing out these limitations in the AUCs panel of the previous **Figure 4**. We apologize that this analysis was not very well defined in our original version. The AUC values were obtained by applying the standard conversion formula to the Cohen's *d* values reported in the previous **Figure 4a-d**, i.e., the estimated standard normal cumulative distribution of the Cohen's *d* divided by the square root of two. Our original intention was not to present these as optimized predictive models, but rather to provide a more intuitive interpretation of the Cohen's *d* effect sizes, particularly

compared to a single summary metric like alpha diversity. However, we agree that this presentation could be misleading.

In this revised version of our manuscript, based also on previous comments from this reviewer, we decided to downplay our discussion and comparison with alpha diversity. Since we have included two clinical trials with longitudinal samplings, we decided to dedicate more space to discuss the longitudinal changes in the microbiome in these trials and their links with our defined ZOE MB Health and Diet SGBs ranks. So, we removed the panel with the AUC values, as it was redundant and not providing more information than the Cohens' *d* from the meta-analyses, and it was also detracting from our main findings. Currently, the comparison with alpha diversity is available in **Supplementary Fig. 14b** (also reported below for convenience).

To clarify the scope of our work, in this study, we focus on defining and using the ZOE MB scores to investigate associations with cardiometabolic health and dietary markers and track longitudinal microbiome shifts. While we presented the initial AUCs to help compare our results with those obtained by using alpha diversity. We concur with the reviewer that building and validating robust predictive models for specific diseases like CVD or IBD was beyond the aims of this particular study.

Supplementary Fig. 14. Meta-analyses of the 50 most favorable and unfavorable SGBs according to the ZOE MB Health and Diet-rankings across disease categories. a) Meta-analyses of the mean difference of the number of the 50 most favorable (left) and unfavorable (right) SGBs found in each sample from 25 public cohorts from five diseases (age ≥ 16 and BMI ≥ 18.5 , retrieved from curatedMetagenomicData 3⁸⁵). Dark-red and light-red markers refer to ZOE MB Health and Diet ranks, respectively. Circles represent the mean count difference from a linear model adjusted by sex, age, and BMI. Diamonds indicate the coefficient of a random-effect meta-analysis (**Methods**). **b)** Meta-analysis on alpha-diversity (richness and Shannon's entropy) in discriminating between cases and controls.

Additionally, the meaning of meta-analysis of AUC is unclear. The analysis lacks a clear biological reasoning.

Moreover, the methodology lacks sufficient details. For instance, whether the Spearman correlation was based on the relative abundance level when present only, or on all abundance levels including absence (meaning zeros). The terms “health weighted” and “diet weighted” showed in the Figure 4e lack an explanation. There are also no details on the “personalized dietary advice”.

We thank the reviewer for highlighting some lack of clarity about the partial correlation calculation and terminology. As originally reported in our manuscript, a prevalence filter was applied to focus on non-rare species and thus ensure a minimum number of non-zero relative abundance values (second paragraph of the **Ranking gut microbiome species linked to cardiometabolic health** section of the **Results**). To better clarify this point, we have now expanded the section of the Methods, which now reads as follows:

*“For each pair of SGB and cardiometabolic marker, we calculated the partial Spearman’s correlation between that SGB’s relative abundance values (including zeros) and that marker’s values, adjusting for sex, age, and BMI, using the “pingouin” Python package (v0.5.4, <https://github.com/raphaelvallat/pingouin>) (**Supplementary Figs. 4 and 7**).”*

We apologize for the lack of clarity in the original **Figure 4e**. As we answered to the previous reviewer’s comments, the former **Figure 4** was removed and some of the results are now kept in **Figure 3e,f**, while we decided to remove the panel with the AUCs as it was based on the direct conversion of Cohen’s *d* values, so redundant with results from the meta-analysis. The original **Figure 4e** panel is now removed, but we wanted to apologize and comment about the lack of definition of the “health weighted” and “diet weighted” terms. Those two terms were referring to the AUC values (converted from Cohen’s *d*) of the “Weighted score sum” (original **Figure 4c**) for the ZOE MB Health (“health weighted”) and Diet ranks (“diet weighted”). For convenience, we report the new **Figure 3** below.

Figure 3. Microbial species in the ZOE MB Health and Diet ranks show significant and reproducible associations with BMI categories and diseases. [...] e) Meta-analysis on the cumulative relative abundance (arcsin square-root transformed) of the 50 most-favorable (left) and unfavorable (right) SGBs from the two rankings. The model is fitted to the disease group and adjusted by sex, age, and BMI. Standardized mean differences (SMD) are retrieved and meta-analyzed (Methods, meta-analysis on SGB richness reported in Supplementary Fig. 14a). f) Meta-analysis of the sum of the normalized ZOE MB Health and Diet-ranks, weighted by arcsin square-root of relative abundance values (right, “Weighted score sum”) or unweighted (left, “Score sum”). Disease acronyms appearing on the figure, IBD: inflammatory bowel disease; CRC: colorectal cancer; IGT: impaired glucose tolerance; T2D: type-2 diabetes; CVD: cardiovascular disease.

We apologize for the lack of details on the personalized dietary advice. Some details were available in the original version of our manuscript in the section **ZOE PREDICT cohorts definition** of the **Methods**, which we updated and now reads as follows:

“The PREDICT 3 UK 23A RT cohort comprises 1,124 healthy individuals who followed a personalized dietary program (PDP) intervention study and from whom pre and post-intervention stool samples were collected ($n = 2,248$). The duration

of the PDP intervention was on average 5 months and 3 weeks (min 42 days and max 295 days), during which participants received personalized food-level dietary advice through the ZOE mobile phone application. Recommendations were based on person-specific food quality scores computed using the ZOE 2022 Algorithm that incorporates food characteristics (e.g., macronutrients, glycemic load, fat quality, level of processing), individual's glucose control and postprandial triglyceride concentrations, gut microbial composition, baseline CVD risk, and prior health history⁸⁷. These person-specific food scores ranged from 0 to 100, with higher values indicating more healthful food items. The PDP did not specifically restrict energy intake (i.e., calorie counting). Participants completed one food frequency questionnaire (FFQ) at baseline and one post-intervention to track changes in dietary habits. Other than enrolling adults with no chronic health issues, participants were also required to self-report that they had not taken antibiotics in the 3 months before the study enrollment, nor during the two sampling times.”

Another main conclusion from the study is the relationship between the gut microbiome and diet. Firstly, the FFQ data differed across the five different PREDICT cohorts. It is unclear how these FFQ differences affect the calculation of the diet index.

We thank the reviewer for raising this point. We included summary statistics of the PREDICT cohorts in the new **Supplementary Table 1**, in which we also report the different dietary indexes. The dietary indexes were calculated per cohort, so we do not expect biases due to slightly different FFQs or because the countries do not have the exact same list of foods. In addition, our FFQs were processed by leveraging a hierarchical tree of foods to automate the FFQ processing for all individuals, including the diet indexes. The newly developed 264-item FFQ used in the PREDICT 3 cohorts, i.e., PREDICT 3 UK22A and US22A, and P3 UK 23A RT, was adapted from the European Investigation into Cancer and Nutrition (EPIC)-Norfolk Study FFQ, and the Diet History Questionnaire (DHQ III). Consequently, there is a large overlap between the food items collected across the FFQs; for example, 90% of questions in the EPIC FFQ are included in the PREDICT 3 FFQ. The 264-item FFQ includes several additional food items to accurately measure modern eating habits and food supply/environment, addressing a limitation of older FFQs [1], and provides the rationale for developing (and validating) and utility in the newer PREDICT cohorts. Within each cohort, we see consistent associations between the a priori dietary patterns and gut microbiome, which may suggest the relationship remains independent of variance introduced by different dietary collection methods.

Dietary record tools such as the FFQ are the most commonly used for assessing dietary intakes in most large-scale epidemiological studies, especially for investigating the relationship between dietary and health outcomes. The PDIs and HEI have become several of the primary diet indices used for studying plant-based dietary patterns and the level of adherence to a specified pattern or a set of recommendations in epidemiologic

studies. Consistent associations of adherence to diet indices with health have been demonstrated across multiple dietary collection methods in systematic reviews of prospective cohort studies [2] and multi-cohort studies. Further, variance in diet indices computed across different dietary collection methods has previously been demonstrated. For example, the validity of PDI scores derived from FFQs was assessed relative to the mean of 24 hr dietary recalls and blood and urine biomarkers, with moderate to good validity in all racial/ethnic subgroups [3].

We updated the description in the **Dietary data processing** section of the **Methods**, which now reads as follows:

“In the PREDICT cohorts, we assessed long-term food intakes using FFQs, which were largely consistent across cohorts. Specifically, for PREDICT 1 participants (UK), we employed a modified 131-item European Prospective Investigation into Cancer and Nutrition (EPIC) FFQ⁹⁸. Participants in PREDICT 2 (US) were surveyed using a similarly validated DHQ-III FFQ including 135 items about food and beverage as well as 26 questions about dietary supplements⁹⁹. In PREDICT 3 UK22A and US22A, and P3 UK 23A RT (baseline and endpoint) we developed and employed a 264-item FFQ adapted from the European Investigation into Cancer and Nutrition (EPIC)-Norfolk Study FFQ and the Diet History Questionnaire (DHQ III). Consequently, there is a large overlap between the food items collected across the FFQs, for example, 90% of questions in the EPIC FFQ are included in the PREDICT 3 FFQ. This FFQ also includes additional food items to accurately capture modern eating habits, a limitation of older FFQ versions¹⁰⁰. In the PREDICT 3 US21 cohort, FFQs were not collected, and only short-term logged dietary data were collected via the ZOE mobile phone application and used instead.”

[1] Wennberg M, Kastenbom L, Eriksson L, Winkvist A, Johansson I. Validation of a digital food frequency questionnaire for the Northern Sweden Diet Database. *Nutrition Journal*. 2024 Jul 24;23(1):83.

[2] Jarvis SE, Nguyen M, Malik VS. Association between adherence to plant-based dietary patterns and obesity risk: a systematic review of prospective cohort studies. *Applied Physiology, Nutrition, and Metabolism*. 2022 Aug 19;47(12):1115-33.

[3] Cousineau BA, Mitchell EL, Hodge RA, Vaccarino V, Alvarez JA, Flanders WD, Stein AD, Mitchell DC, McCullough ML, Hartman TJ. Reproducibility and Validity of Plant-Based Dietary Indices in the Cancer Prevention Study-3 Diet Assessment Substudy. *The Journal of Nutrition*. 2024 Nov 26.

Secondly, the gut microbiome was associated with both cardiometabolic health and diet. However, the study does not provide direct statistical evidence to support that diet modulates the gut microbiome index for cardiometabolic health, even though the ZOE-selected species showed differential abundance upon dietary intervention.

If we interpret the comment from the reviewer correctly, the reviewer is pointing at a lack of direct evidence for a causality link between variations in diet, variations in microbiome, and variations in cardiometabolic health. We agree that this is one of the main limitations of our study design, and we have now highlighted it better (see below). However, this is currently a limitation that cannot really be overcome without direct non-nutritional intervention on the gut microbiome: if we see longitudinally that a food A is associated to the increase of the gut bacterium B, to show that B is acting on cardiometabolic health independently (or in addition) to food A, we would need to administer bacterium B to individuals and follow them in time. However, this is currently not possible as there is no protocol nor ethical support for doing that, with the only exception of "traditional" probiotics (that are, however, usually not members of the gut microbiome) or a very few approved next-generation biotherapeutics. We would like to further clarify that the aim of our study was not to create a new 1-dimensional microbiome index, but rather to show that we can rank microbial species according to their consistent associations with diet and cardiometabolic markers. Nevertheless, in our study with an unprecedented sample size, we provide multiple lines of evidence in three longitudinal studies (two newly added clinical trials and a large interventional study) that changes in the microbiome are associated with subsequent changes in the microbiome composition and cardiometabolic markers consistently with what the cross-sectional associations pointed out. In this direction, in the revised version of our manuscript, we have now included two dietary intervention studies that are registered clinical trials (namely, METHOD and BIOME) in which we show that individuals in the intervention arms have a greater changes in the microbiome composition compared to the controls, and these changes are linked to the favorable and unfavorable ranked species. Results are provided in the **Dietary interventions improve favorably ranked species** section and in **Figure 4**. In particular, it was observed that individuals assigned to the PDP group of the METHOD paper (<https://doi.org/10.1038/s41591-024-02951-6>), who followed a personalized dietary intervention, after 18 weeks they showed in general a lower energy intake and a decrease in triglycerides, HbA1c, weight, and waist circumference. While this is not resolute in disentangling the relative contribution of diet to health and of the diet-induced microbiome changes on health, it still describes the collective effect of diet and microbiome on health in a statistically reproducible and consistent way.

Despite these new analyses, we agree with the reviewer's concern that our study design fails in proving one key point, i.e., whether the impact of diet on cardiometabolic health is necessarily mediated by the gut microbiome. In the revised manuscript, we now acknowledge this in the **Conclusions** as a key limitation of our study:

"Many health-associated host markers are correlated as they are the nutritional indicators, and disentangling their direct interactions from those mediated by the microbiome will remain elusive until large-scale microbiome interventions will be possible in humans. In this respect, one key limitation of our study design is that it does not allow disentangling directly the effect that diet exerts on the microbiome to improve cardiometabolic health from the impact of diet only. This is particularly important as diet-based ranks were more dependent on

country-related differences compared to cardiometabolic ranks, and further studies should explore food-specific links with gut microbial species and cardiometabolic outcomes in greater detail ⁹⁷, although this would entail designing large scale studies in which both introduction of single foods and alterations of specific microbiome characteristics (e.g. by administration of specific microbiome members) are tested which is currently highly problematic.”

Minor comments:

- No page number and line number. It makes review report difficult

We apologize for this and thank the reviewer for pointing it out, this is now fixed.

- Introduce the full name when the abbreviates were introduced at the first time, such as Atherosclerotic cardiovascular disease for ASCVD.

We thank the reviewer for highlighting this. However, ASCVD was correctly introduced the first time it appeared (on page 3), and the acronym was used two times on page 4. Then, the ASCVD acronym was reintroduced in the **Methods** (section **Cardiometabolic markers collection**), page 19, as we considered the **Methods** as a separate part of the manuscript, and reintroducing the acronym could help the readers We carefully checked all other acronyms to ensure they were correctly introduced the first time they were mentioned.

- Figure 2a plot for the relative abundance when present. Does Spearman correlation take absence (or zero) into account. If yes, the information of this plot is not very informative.

The panel “Relative abundance when present” of **Fig. 2a** shows the range of relative abundance values of the top 15 most favorably and unfavorably-ranked SGBs, not showing the zeros which are (i) not displayable on a log scale (although could be forced to be visualized); and (ii) being at least 20% by definition of the prevalence filter, would force the boxplot towards the zero making the interpretation of the visualization more difficult. On the other hand, the calculation of the partial Spearman’s correlations considers the zeros, but the purpose of the plot was to show the general trend of the relative abundance of these species, showing if there were some patterns between the most favorable vs. the most unfavorable ones in terms of relative abundance.

Referee #3 (Remarks to the Author):

Referee #3 expertise: clinical trials, biostatistics

Peer review of “Gut microbiome species indicative of cardiometabolic health are modulated by diet in large and interventional cohorts of over 34,000 individuals”

The article describes an ambitious effort where five large metagenomic ZOE PREDICT cross-sectional studies totaling over 34,000 individuals from the UK and the US were analyzed to identify microbiome species consistently associated with dietary quality or markers of cardiometabolic health. Within each of 5 cross-sectional studies, the authors computed partial Spearman correlations for each (species, marker) combination within a dietary or health category, ranked these correlations, normalized the ranks to fall between 0 and 1 (by dividing by the relevant number of correlations for a given cohort and health marker) and then averaged these normalized ranks across cohorts and categories for each species, with these species category averages then averaged together to create ZOE MB Diet Rank Scores and ZOE MB Health Rank Scores. They also identified the set of 50 most unfavorable species and of 50 most favorable species based on these rankings. They then assessed the correlation of these rank scores and measures based on the most unfavorable or favorable species with various health outcomes like BMI or disease status, using other publicly available metagenomic cohorts. Further, they demonstrated via meta-analyses that their derived measures compared favorably to widely used alpha-diversity measures in discriminating between diseased and nondiseased individuals. The authors also showed in a ZOE PREDICT longitudinal dietary intervention study that the selected species were appropriately correlated with changes in diet quality and BMI. The results are of interest to microbiome researchers because the species identified as having consistent associations with health measures and dietary quality may be of interest in their own right. Also, the development of derived variables based on these rankings may have appeal for those seeking additional useful and simple summary measures of the microbiome for use in clinical research. Major strengths of the study include the large sample size and the practical and generally sound methods used by the team in conducting and reporting their analysis, including the reporting of very detailed supplementary data.

There are some addressable shortcomings in the analysis and reporting. Also, the overall strength of the associations between the newly derived measures and the health and dietary quality measures is somewhat modest. Further, there is of course residual confounding inherent in observational data, particularly cross-sectional data. Also, the large number of species and markers considered required that the authors use an approach that makes some reasonable simplifications. Nevertheless, this is an important project and a suitably revised manuscript would be of interest to microbiome researchers.

We thank the reviewer for their comment, highlighting the strengths of our study, and appreciate the comments and suggestions raised, which we addressed point-by-point below.

Moderate to major concerns:

1. The fourth paragraph of the introduction starts with a sentence describing shortcomings of nutritional intervention studies and then follows that with a second sentence that implies that these shortcomings can be overcome by large-scale comprehensive studies with multi-national populations. That is not correct. In particular, the confounding biases from local lifestyle and nutritional habits is not necessarily going to be removed by combining data from multiple locations and/or increasing the sample size. The authors should revise the second sentence to avoid the suggestion that internal validity threats can be overcome merely by large samples from multiple locations.

We thank the reviewer for pointing this out and agree with the point raised. We tried to keep the original text short and highlight the need for large-scale data in nutritional studies; however, we agree that large-scale studies alone are not the solution to overcome all existing limitations. We updated the last paragraph of the **Introduction**, which now reads as follows:

“Nutritional intervention studies usually involve low sample-size cohorts at the population level and are often limited by their statistical power and specificity to local lifestyle and dietary habits. Large-scale comprehensive studies with multi-national populations can help disentangle some of the complex interplays between dietary patterns and the gut microbiome to develop personalized interventions to prevent and treat CMDs. Accordingly, we put together, sampled, and analyzed five of the largest metagenomic cohorts available to date, comprising more than 34,000 individuals and spanning two continents, paired with diet data and detailed markers of cardiometabolic health (CMH). We identified microbiome species consistently associated with more favorable and (inversely) unfavorable cardiometabolic markers across continents. These species were organized into two microbiome rankings, representing CMH and diet quality, respectively. These rankings were associated with improved BMI and diet quality in three longitudinal dietary cohorts.”

2. The second paragraph of the “Results & Discussion” section notes that there were good AUC measures associated with their random forest classification and regression algorithms that used microbiome features to discriminate among patients according to continuous markers. In reading the associated methods, though, one learns that the authors computed the AUC by considering only those patients in the top fourth and the bottom fourth of a continuous marker. Although the authors used that method in a previous publication, it is unacceptable, because it describes a discrimination task of little clinical or research relevance, particularly because it ignores data from the middle half of the analysis sample, and the individuals in this half that are ignored depends on the continuous marker. There are alternatives that can be used in this setting. If the authors considered the top fourth to be important, they can discriminate patients in the top fourth versus those in the bottom three fourths. (Similarly, they could discriminate between the bottom-fourth and those in the top three-fourths.) Another alternative, more clinically relevant, is to use marker specific thresholds that have clinical relevance. A third alternative, in case the authors want to use the ordinal classification into fourths or some other

ordinal outcome with more than two levels is to use the asymmetric Somers D measure, the one that is commonly denoted Somers D(regression prediction | ordinal classification of biomarker). For a binary classification, this Somers D can be transformed directly into the AUC, using the formula $AUC = 0.5(\text{abs}(D) + 1)$ [See Newson, R. (2006). Confidence Intervals for Rank Statistics: Somers' D and Extensions. The Stata Journal, 6(3), 309- 334.

<https://doi.org/10.1177/1536867X0600600302.>]

I was curious as to how inflated the reported AUCs here could be. Hence, I simulated in SAS a dataset with 20,000 observations, sampling each from a bivariate standard normal distribution with a correlation of 0.40, to make it relevant to reported correlation between predicted and actual values reported here for the better fitting models. I computed the relevant Somers D for three scenarios and then converted these into C statistics by applying the transformation reported in the above paragraph. The first scenario was to not transform the marker variable; the second was to dichotomize the marker so that it contrasted the top fourth to the bottom three-fourths; the third was to do what the authors did here, contrasting the top fourth with the bottom fourth and ignoring the middle half. The AUCs (95% CI) were 0.63 (0.62, 0.64), 0.69 (0.68, 0.70), and 0.78 (0.77, 0.79), respectively. Hence, it is clear that contrasting the top fourth to the bottom fourth creates a misleading impression of how well a predictor can discriminate patients with respect to a marker. If feasible, the authors should revise their calculation of the AUCs used for this part of the manuscript, replacing it with something that involves the whole sample (e.g. something like scenario 1 or scenario 2) and then update their figures and text. Given that this particular section of the manuscript is mainly just trying to show proof of concept that microbiome data can be used to develop use machine learning and regression models and the authors are also reporting the very relevant Spearman correlation coefficient between real and predicted values of about 0.40, it would be reasonable for the authors to retain their current analysis but add to the second paragraph of the results section a note saying that the Spearman correlation coefficients are of more relevance in describing how well one can discriminate patients with respect to continuous markers using the microbiome, as the AUC was based on discriminating between the top and bottom fourths on the marker, an easier tasks which leads to higher AUC values. A word of computation advice: SAS and Stata appear to be to be much faster at being able to supply Somers D values than the R packages I tried on my simulated data. In SAS, you can use PROC FREQ with options to get Somers D. With Stata, you can download the package created by the author of the article cited in the previous paragraph.

We thank the reviewer for pointing out this aspect and greatly appreciate the reviewer's effort in testing the different scenarios. We want to highlight that the ML results were not used to define the rankings, but were used and presented in **Fig. 1b** and **Supplementary Table 1** (original submission, now **Supplementary Table 2**) to show the strength of the markers considered for the partial Spearman's correlation, which was used for defining the rankings. As suggested by the reviewer, we performed the ML classification analyses considering (i) the 1st quartile as one class and the 2nd, 3rd, and 4th quartiles as the other class, and (ii) the 1st, 2nd, and 3rd quartiles as one class and the 4th quartile as the other class. These results are now provided in **Supplementary Table 2**. Additionally, we calculated the Spearman's correlations between the three AUCs to assess how much the three ML tests were in agreement. Overall, despite some

differences in the AUC absolute values, we found a high agreement when comparing ML classification performed using the (a) 1st vs 4th quartiles, (b) 1st vs 2nd, 3rd, and 4th quartiles, and (c) 1st, 2nd, and 3rd vs 4th quartiles. The Spearman's correlations for (a) vs (b), (a) vs (c), and (b) vs (c) were 0.932, 0.927, and 0.830, respectively. In case the reviewer strongly believes we should change the information visualized in **Fig. 1b** or that we should add this new information to the panel, we are happy to do so. Moreover, we added further details to the **Results** section, which now reads as follows:

*“This analysis revealed strong associations consistent across the five ZOE PREDICT cohorts between the microbiome and markers of personal characteristics, nutrition, and CMH (**Figure 1b**), which were categorized for ML classification by predicting the 1st quartile versus the remaining quartiles as well as by predicting the 4th quartile versus the remaining quartiles.”*

3. Page 5, sentence describing results presented in Supplementary Table 4. To report the constituency of cohort-specific ranks for the 100 selected SGB, the authors say there is a “11.3% average variation”, but that is a misleading description of what should be called a “0.113 average range”, as the average is computed using the range statistics (max – min). The authors should also consider reporting the average coefficient of variation is 38.5%. That can be determined by dividing the standard deviation in each row by the mean in each row and then averaging across the rows.

We thank the reviewer for catching this. Indeed, as suggested, we have now computed and added the coefficient of variations to the main text and within the **Supplementary Table 6**. Additionally, we also included the index of dispersion and the mean absolute deviation (using the median as the central value). As indicated by the reviewer, the coefficient of variation is 38.5%, but we would also highlight that both the index of dispersion and the mean absolute variation report low number values (0.036 and 0.008, respectively), suggesting that ranks of the same SGB across cohorts are not overdispersed or randomly distributed, but rather clustered. The text now reads as follows:

*“Each cohort-level category comprises several different markers, nonetheless, the 15 best and worst-ranked species showed consistent associations when considering every marker in each category, with a 38.5% coefficient of variation, 0.036 index of dispersion, and 0.008 mean absolute deviation, in the cohort-specific ranks (**Figure 2b** and **Supplementary Fig. 5**, and **Supplementary Table 6**).”*

4. Related to Concern 3, there should be some reporting of the internal consistency/ reliability of the cohort-specific (or country-specific ranking scores) that ultimately are averaged together to form ranking score. For example, the between-country Spearman correlation for the ranking scores for SRGs scored in both countries is only modest. Using the data in Supplemental Table 3, this correlation is 0.61 for the Cardiometabolic ranking score (Columns CMH_UK and

CMH_US) and 0.26 for the Diet ranking score (Columns Diet_UK and Diet_US). It might be useful for the authors to note this, as it suggests that the relative ranking of a SRG, especially those SRG ranked in the middle, is likely to change several positions, on average, if the ranking scores were updated with data from new studies. Related to this, the authors should consider computing the within-SRG intracluster correlation coefficient associated with the cohort-specific ranking scores, the ratio of the between-SRG variance component to the sum of this component with the within-SRG between-cohort variance component. This can be estimated by fitting a mixed-effects model with all of the individual SRG-Cohort ranking scores for a given category (e.g. DIET or CMH), with random intercepts for SRG and then forming the ratio of the variance component for SRG with the sum of the between- and within-SRG variance components. (See <https://pubmed.ncbi.nlm.nih.gov/14969463/>)

We agree with the reviewer that assessing the consistency of the rankings is an important aspect. Indeed, in the original version of our manuscript, in the section **Ranking gut microbes associated with healthier diets** we referred to **Supplementary Fig. 7a** (now **Supplementary Fig. 8a**) and **Table 3** (now **Supplementary Table 5**), for a comparison of the Health vs. the Diet rankings. As pointed out by the reviewer, consistency and discordance between the rankings are important aspects to discuss. Later in the section, we identified and discussed those SGBs ($n = 65$) showing an absolute rank difference ≥ 0.3 , highlighted in **Supplementary Fig. 8a** and listed in **Supplementary Table 8**, as we wanted to provide some insights about these differentially ranked species. As the reviewer highlighted in their comment, country is another important aspect characterizing our cohorts. That's why at the end of the paragraph, we focused on discordant rankings if computed considering only cohorts from the UK or the US. Country-specific differences were then reported in **Supplementary Figs. 8b** and **8c**. One of the reasons we decided to focus on the arbitrarily picked for ease of use, 50 SGBs ranked at both extremes, was because the 50 most favorable and unfavorable SGBs showed more consistent rankings across cohorts, despite the different country of origin. In addition, when we discussed the variability of the ranks (**Supplementary Figure 5** depicts the top/bottom 15), we have now included in **Supplementary Table 6** the coefficient of variation, index of dispersion, and mean absolute deviation for ranks of the 50 most favorable and unfavorable SGBs.

As suggested by the reviewer, we fit a linear model to estimate the between and within SGB variance for the cohort and the UK vs US comparisons, for both the Health and Diet ranks, to compute the Intraclass Correlation Coefficient (ICC). For the cohort comparison, we obtained a between variance of 0.0533 and a within variance of 0.0316 from the model on the Health ranks, which yielded an ICC = 0.6275. Instead, the model on the Diet ranks gave a between variance of 0.0388 and a within variance of 0.0463, which yielded an ICC = 0.4563. For the UK vs US comparison of the Health ranks, we obtained a between variance of 0.0459 and a within variance of 0.0315, which yielded an ICC = 0.5929. Similarly, for the Diet ranks, we obtained a between variance of 0.0195 and a within variance of 0.0547, which yielded an ICC = 0.2623. These are now also

reported at the end of the **Ranking gut microbes associated with healthier diets** section of the **Results**:

*“We then investigated how the ZOE MB Health and Diet rankings vary across the US and UK populations. Overall, UK and US ZOE MB Health rankings showed high consistency (Spearman’s $p = 0.61$, **Supplementary Fig. 8b**), while country-specific ZOE MB Diet rankings were more heterogeneous (Spearman’s $p = 0.26$, **Supplementary Fig. 8c**). To further quantify the proportion of the total variation attributable to the country and the variation within an SGB to reflect the disagreement between the UK and US ranks, we calculated for both the ZOE MB Health and Diet ranks the intraclass correlation coefficient (ICC) ⁸⁶. We got an ICC = 0.5929 for the ZOE MB Health ranks, and ICC = 0.2623 for the ZOE MB Diet ranks, suggesting that the ZOE MB Health ranks are more consistent across countries than the ZOE MB Diet ranks (**Supplementary Fig. 8b,c**). Comparing the variability of the ranks across cohorts, we obtained an ICC = 0.6275 and an ICC = 0.4563 for the ZOE MB Health and Diet ranks. This could indicate that diet rankings were able to capture dietary differences between countries, reflected by country-specific microbial compositions; however, generally, the most favorably-ranked species appeared to match across populations with similar levels of industrialization and lifestyles.”*

We would like to highlight that, while we show that these rankings are able to stratify people according to their health status, and in longitudinal cohorts are able to identify people who significantly improved their diet, they are likely not universal. These SGBs are potentially much more specific to populations following a Westernized lifestyle. So, the main message of this work is that by the exploitation of very large metagenomic cohorts, we are now able to rank microbial species in a meaningful way with respect to diet and cardiometabolic health. But as databases expand and computational tools improve, we foresee that future studies will be able to improve these rankings further, for example, by considering more heterogeneous cohorts or developing country-specific rankings, and extend them to a larger number of species. Finally, microbial species is only one aspect of the microbiome, and future works could extend this type of ranking to other microbiome features, such as functional profiles.

5. After developing their rankings, the authors compare them for several SRGs. Notably, for most of them, at least one of the rankings being mentioned was in neither the top-50 nor the bottom-50. For example, to be in the top 50 for Health, the ranking score would have to be less than or equal to 0.170 for Health or to 0.188 for Diet. To be in the bottom 10, the ranking score would have to exceed 0.906 for Health or 0.866 for Diet. For SGB 6749, the Health ranking score is 0.267. Near the bottom of Page 7, SGB14838 is reported as having been found to be “associated with health cardiometabolic markers”, citing its ranking score of 0.363. That score does not necessarily demonstrate that SGB14838 was associated with the health markers, as it is not a particularly impressive score. I would suggest the authors stick to the informal descriptions “favorably ranked” or “unfavorably ranked” or else refer to reported results (perhaps

in the supplement somewhere I missed) that demonstrates a statistically significant linkage of the SRG with the health or diet marker.

We thank the reviewer for pointing this out. Indeed, what we originally meant was that those SGBs were found in the favorable range, despite being present with very different cardiometabolic and diet ranks (absolute rank difference ≥ 0.3). We have now corrected the text as suggested, which reads as follows:

*“Among these, for example, *Harryflintia acetispora* SGB14838 was found associated with favorable cardiometabolic markers and unfavorable diets (ZOE MB Health-rank = 0.363 and ZOE MB Diet-rank = 0.879) in this study.”*

6. Page 14, last sentence in paragraph describing Figure 4e. Instead of “accurate proxy”, which was not assessed with the reported analysis, the authors should say “moderate and useful discriminator”.

We thank the reviewer for their suggestion, which we have implemented in the text as follows:

“Overall, the ZOE MB Health and Diet-rankings were a moderate and useful discriminator for the health status and have the potential to be used as a baseline assessment of microbiome’s status along the health-disease axis.”

Minor concerns

1. First and third paragraphs of “Results & Discussion” section and Figure 1. The present manuscript involves 6 primary studies, five of them cross-sectional and one of them longitudinal, but in some places in the manuscript, the authors refer to only five studies. When this is done, the authors are referring to the five cross-sectional studies. It would help avoid confusion if the authors add some adjectives here and there to make this more clear. Also, the first study, PREDICT 1, was overwhelmingly comprised of UK individuals (n=1,001), but it did have 97 US individuals. So, technically, the second sentence of the first paragraph is incorrect, because it counts these 97 US individuals as being in the UK. (Instead of there being only 21,243 US individuals, there were 97 more; whereas the UK had 97 fewer than 13,451 individuals.) The authors could correct this sentence in a couple of different ways, either by correcting the numbers or changing the description to say they are tallying across the studies belonging to a country and that they consider the PREDICT 1 study to be UK, despite the inclusion of some US individuals.

We thank the reviewer for pointing this out. We have updated **Fig. 1**, correcting the PREDICT 1 numbers to reflect the presence of samples from individuals from both the UK and the US.

2. Page 18. Methods section. Paragraph “ZOE PREDICT cohorts definition”. There are only 50 states in the United States, so if the states North Dakota and Hawaii are not represented in the cohort, then that means 48 US states were represented, not 49. A suggested rephrasing would be “975 individuals from 48 US states and the federal District of Columbia (the states not included were North Dakota and Hawaii).

We thank the reviewer for picking up on our mistake. All of the US states were represented in this cohort, so we have corrected the text to reflect this fact.

3. Page 19, “Dietary data processing”. There does not seem to be a description of how data were assessed in the intervention study (P3 UK 23A RT). Should there be?

We thank the reviewer for identifying this. We have added details of how data was assessed in the P3 UK 23A RT intervention study in the ‘**Dietary data processing**’ section of the **Methods**, which now reads as follows:

“In PREDICT 3 UK22A and US22A, and P3 UK 23A RT (baseline and post intervention) we employed a newly developed 264-item FFQ”.

4. Page 21. Commendably, the authors used “partial Spearman’s correlation”. Presumably, they used a relatively novel technique based on probability-scale residuals. They should provide a reference to either the package they used or a reference to the method (e.g. <https://www.rdocumentation.org/packages/PResiduals/versions/0.2-5/topics/partial.Spearman>)

We thank the reviewer for acknowledging this. Partial correlations were calculated using the “pingouin” Python package (v0.5.4). The approach relies on the variables’ variance-covariance matrix as in the R package “ppcor” (PMID [26688802](https://pubmed.ncbi.nlm.nih.gov/26688802/)). We have updated the **Rankings definition** section of the **Methods**, which now reads as follows:

“For each pair of SGB and marker, the partial Spearman’s correlation was computed, adjusting for sex, age, and BMI, using the “pingouin” Python package (v0.5.4, <https://github.com/raphaelvallat/pingouin>) (Supplementary Figs. 3 and 6).”

5. Page 5 and throughout manuscript. The description of the normalized correlation rankings is not quite right. They are not percentiles, because percentages go from 0 to 100, while these go to 1. It might be more clear to say that rankings were normalized to range from 0 to 1 by dividing the relevant sample size.

We thank the reviewer for highlighting this. Rank percentiles were derived using the pandas.DataFrame.rank function, by setting the param pct=True. The behavior of this function is to return the ranks as percentiles in the range between 0 and 1, so rank percentiles are normalized by the sample size (see

<https://github.com/pandas-dev/pandas/blob/0691c5cf90477d3503834d983f69350f250a6ff7/pandas/ libs/algos.pyx#L1254>). We have edited the Methods text to be clearer on this aspect, which now reads as follows:

*“Then, ranked partial correlations were represented as rank percentiles in the range from 0 to 1 (pandas.DataFrame.rank function with param pct=True), by normalizing them by the respective sample size (**Figure 2b,c** and **Supplementary Figs. 2 and 5**) and for each category, the average percentiles of the markers in that category were computed (**Figure 2a** and **Supplementary Fig. 5**).”*

Referee #4 (Remarks to the Author):

Referee #4: gut microbiome and metabolism, clinical trials, metabolic disease

This is an unprecedented study setting a standard state for associations of the human gut microbiome with diet and health markers using metagenomic data from >30 thousand individuals from a commercial health platform. I believe this can be a significant contribution using adequate measures to assess reliable and reproducible associations. While overall, the literature is full with associative studies of the gut microbiome in humans, the sheer scale of this study makes it a unique reference work.

We thank the reviewer for the positive assessment of our manuscript.

That being said, I have a couple of comments which need to be addressed, specifically regarding the protocol of the The PREDICT 3 UK 23A RT cohort and some clinical aspects.

1. Figure 1: females overrepresented thus this should be reflected as a limitation

The reviewer raises an important detail that we have now highlighted in the conclusions as follows:

“We acknowledge that the demographic composition of the cohorts may influence some associations and we are continuing to expand in both population scale and precision of each host-associated readout.”

2. Quite a drop in associations with the gut microbiome after visceral fat, i.e. Spearman correlations rather weak with ASCVD. This should be discussed with regard to the strong title “cardiometabolic” since it seems to be really driven by anthropometry

We thank the reviewer for pointing this out. We believe the comment arose from **Supplementary Fig. 1**. We agree that markers such as visceral fat or BMI are markers that represent body fat and seem to be strongly linked with the microbiome composition and abundance. In contrast, ASCVD is a multi-factorial risk estimator based on age, sex, race, blood pressure, cholesterol levels, and other risk factors, such as smoking. This is a wide range of markers summarized into one single index, and the fact that it shows milder associations with the microbiome may not be that unexpected. We would like to highlight that the figure reports the performance of a Random Forest model trained to predict the target variable using only the SGB-level relative abundance, not other host factors. Moreover, cardiometabolic health is a term that does not exclude anthropometric measurements. For example, as we stated in the **Introduction**, known risk factors for type-2 diabetes are the consumption of calorie-dense, ultra-processed, and animal-based diets, but they also encompass anthropometric measurements such as BMI (also associated with increased blood pressure), cholesterol and insulin resistance levels, which are also known risk factors for cardiometabolic disorders. Thus,

anthropometric information, such as BMI (despite being an imperfect marker) and blood pressure (which we consider among the many other markers like fasting and postprandial cholesterol, glucose, and triglycerides levels), should be considered risk factors for cardiometabolic disorders. To make this clearer, we amended the **Introduction** (reported below) in which we better highlight the fact that BMI can be considered a marker of cardiometabolic disease, in particular because it is widely available in cohorts that also sampled the microbiome:

“Cardiometabolic diseases (CMDs) are the leading causes of morbidity and mortality in Western countries and constitute a heavy burden on global healthcare systems ¹⁻⁵. Among these, the most predominant diseases are cardiovascular disease and type-2 diabetes (T2D) ⁶, which are connected with the increased consumption of calorie-dense, high-risk processed foods, observed over the past few decades ^{7,8}. Habitual diet is among the known risk factors associated with CMDs and is the primary modifiable target for prevention and treatment ^{9,10}. Many markers of CMDs, ranging from clinical measurements (e.g., blood pressure) to lipid profiles (e.g., triglycerides, cholesterol, and lipoproteins), glucose (e.g., fasting glucose levels and hemoglobin A1c) and inflammatory (e.g., cytokines and high-sensitivity C-reactive protein) markers, are usually not available in large population studies that sample the microbiome. A known risk factor of CMDs is body mass index (BMI), an anthropometric marker that has been negatively associated with overall health and, in particular, with increased blood pressure, cholesterol and insulin resistance levels, and inflammation, despite limitations in its ability to account for differences in sex, age, ethnicity and the distinction between muscle and fat, and fat distribution ¹¹⁻¹³.”

3. With regard to anthropometry, it would be helpful to have more clinically assessed data e.g. from PREDICT 1 and PDP to better understand the basic characteristics of the cohorts investigated. I understand that this is not available for all cohorts but could be provided where-ever possible.

We thank the reviewer for their comment. We added a new **Supplementary Table 1** describing the main anthropometric information about the PREDICT cohorts analyzed in this work. We also updated the text in the **Linking the gut microbiome with dietary and cardiometabolic markers in large populations** section of **Results & Discussion**, which now reads as follows:

*“We used the previously published ZOE PREDICT 1 cohort 14 (n = 1,098) together with four newly defined and very large-scale microbiome cohorts from the ZOE PREDICT Studies (n = 33,596, **Figure 1a, Supplementary Table 1, Methods**), to assemble an extensive microbiome dataset of individuals with detailed dietary records along with anthropometric and cardiometabolic measures.”*

4. The authors use the word metadata, while what they really mean are the clinical data derived from the clinical studies. Metadata rather describe data types. Please correct.

We thank the reviewer for their suggestion, which we have implemented throughout the manuscript.

5. It seems like a control group is missing for the ZOE PREDICT longitudinal cohort (PDP). This should be clearly stated as a limitation.

We thank the reviewer for highlighting this. The longitudinal cohort included in the original version of the manuscript (named P3_UK23ART) comprised 1,124 individuals who followed a volunteer and personalized dietary intervention, receiving user-specific dietary recommendations via a smartphone app. As noted, this cohort was not a multi-arm intervention study. However, for these individuals we had dietary and BMI information both at baseline and after the diet intervention period, which allowed us to assess that in overall, this general population cohort, improved their diet, as shown by the significant increase in the healthy plant-based diet index (hPDI) and the significant decrease of the unhealthy plant-based diet index (uPDI). Additionally, we observed a significant decrease of about 1 point of BMI (**Supplementary Fig. 16**). To make our results more robust, we reproduced the same analysis on two independent cohorts, one from the US (named ZOE METHOD) and from the UK (named ZOE BIOME), which are randomized controlled interventional studies with clinical trial registration (ClinicalTrials.gov registrations: NCT05273268 and NCT06231706, respectively). Both these studies comprise a control and an intervention arm, with the addition of a prebiotic arm present only in the ZOE BIOME cohort.

Below, we report the new paragraphs presenting and discussing these results and the new Figures 4 and 5.

“Dietary interventions improve favorably ranked species

To validate the effect of diet changes on the presence and abundance of gut microbial species according to their ZOE MB Health rankings, we re-analyzed two dietary intervention studies, namely ZOE METHOD⁸⁷ and BIOME⁸⁸ (ClinicalTrials.gov registrations: NCT05273268 and NCT06231706, respectively). Briefly, the ZOE METHOD cohort comprised in total n = 347 individuals from the US who were split into a personalized dietary intervention program (PDP, n = 177) arm versus a general diet advice following the United States Department of Agriculture recommendations (control, n = 170) arm. Individuals assigned to the PDP group followed a personalized dietary intervention, following suggestions provided via a mobile app, and overall, they showed lower energy intake and a significant decrease in triglycerides, HbA1c, weight, and waist circumference after 18 weeks⁸⁷. The ZOE BIOME cohort, comprised in total n = 399 healthy adults from the UK, who were randomized into three groups, primary intervention (prebiotic blend, n = 116), functional control (control, n = 120), and daily probiotic

(probiotic, $n = 113$). All individuals in the prebiotic blend group received the same prebiotic supplement, while the control group was given bread croutons to match its calorie intake. The probiotic group was instead supplemented with a daily intake of *Lactocaseibacillus rhamnosus* 15 billion CFU. Overall, weight, waist circumference, metabolites, and gastrointestinal symptoms did not differ significantly between groups⁸⁸. In addition, we collected, curated, and analyzed $n = 1,124$ individuals from the ZOE PREDICT 3 UK23A RT (P3_UK23ART) longitudinal cohort, who voluntarily followed PDP suggestions provided via a mobile app and collected gut microbiome samples before and after enrolling in the program (samples $n = 2,248$, **Methods**). The interval between samples was, on average, six months.

We identified the microbiome species significantly impacted by the interventions in each of the three cohorts. Specifically, in the ZOE BIOME cohort, of the over 500 prevalent SGBs (>10% at both time points, from 538 to 571 depending on the arms), 57, four, and 14 SGBs, respectively, showed significant changes at the endpoint (Q value < 0.01) for the prebiotic blend, probiotic, and control arm (**Figure 4a**). Among the many species with a significant change found in the prebiotic blend arm, there are the beneficial fibre-degrading *Bifidobacterium adolescentis* (SGB 17244), *Bifidobacterium longum* (SGB 17248), and *Blautia obeum* (SGB 4811)^{89–91}, as well as the butyrate-producing *Agathobaculum butyriciproducens* (SGB 14993), *Anaerobutyricum hallii* (SGB 4532), and *Coprococcus catus* (SGB 4670)^{92,93}. In contrast, the species *Dysosmobacter welbionis* (SGB 15078), among the top unfavourably associated SGBs in our study, was significantly decreased by the same dietary intervention (**Supplementary Table 25**). In the ZOE METHOD cohort, we found 46 SGBs significantly different in their relative abundance in the PDP arm while only two in the control arm (**Figure 4b** and **Supplementary Table 25**, Wilcoxon rank-sum test Q value < 0.1). Of note, the prominent butyrate producers *Roseburia hominis* (SGB 4936) and *A. butyriciproducens* (SGB 14993) were also found increased in the PDP intervention. Finally, for the P3 UK23A RT cohort, among 797 SGBs with a prevalence of at least 5% at both time points, 404 SGBs had a significant change in relative abundance (Q value < 0.01), and 258 were still significant after stricter Bonferroni correction (P value < 1.255e-05, **Figure 4c**). For the significantly changing SGBs, those that increased were more prevalent at endpoint than the decreasing ones, suggesting that favorable microbes were also more prevalent post-intervention, while the unfavorable ones tended to be less present in the individuals' microbiome (**Figure 4d-f** and **Supplementary Fig. 15**).

For the increasing and decreasing SGBs identified in the three different longitudinal cohorts, we checked the values of their ZOE MB Health and Diet rankings, if present among the 661 ranked SGBs. The dietary intervention groups (BIOME, prebiotic blend and METHOD, PDP) and the ZOE customers (P3_UK23ART), which all aimed to improve diet, showed the highest number of

significantly changing SGBs, with the SGBs with increased relative abundance at endpoint showing significantly more favorable ranks than the decreasing SGBs (**Figure 4g-i**, Mann-Whitney U test P values = $7.78e-3$, $5.20e-5$, and $2.21e-11$, respectively). The same patterns were shown when considering the ZOE MB Diet-ranks (**Supplementary Fig. 15f**). No significant patterns were present for the probiotic group of the BIOME cohort or the control groups of BIOME and METHOD cohorts (**Figure 4g,h**). Focusing on the gut microbial SGBs that were significantly differentially abundant after dietary interventions across the BIOME and METHOD clinical trials as well as the P3_UK23ART longitudinal cohort, we performed a meta-analysis (fixed-effect with inverse variance of the \log_2 -ratio of SGBs' mean relative abundance values at endpoint over baseline; **Figure 5a** and **Supplementary Table 25, Methods**) and found the most increasing SGBs across cohorts (Q value < 0.1 , $n = 15$) to consistently show more favorable rankings than the most decreasing SGBs (Q value < 0.1 , $n = 15$) for both the ZOE MB Health and Diet ranks (**Figure 5b**, P values = 0.0381 and 0.0028 , respectively, **Supplementary Table 25**). Among the highest increasing and decreasing species retrieved via meta-analysis (**Figure 5a**), we found some overlapping with species specific to different dietary patterns from a recent work⁴⁴. Specifically, *Lachnospiraceae* bacterium (SGB 4953) and *R. hominis* (SGB 4936) were both associated with a vegan diet, while *R. torques* (SGB 4608) was found associated with omnivores. The *Streptococcus thermophilus* (SGB 8002) and *S. salivarius* (SGB 8007) species were found shared in food microbiomes,⁹⁴ and the *S. thermophilus* was also associated with non-vegans⁴⁴.

Together these results show how modification to dietary interventions or tailored probiotic blends, both aiming at improving diet quality, positively modulate the microbiome composition. The SGBs' rankings (ZOE MB Health and Diet) that were defined on cross-sectional independent cohorts, were strongly and consistently predictive of the SGBs most associated with the dietary interventions in independent cohorts and countries, supporting the direct, reproducible, and actionable link between diet and microbiome composition.”

Figure 4. Dietary interventions increase favorable and decrease unfavorable microbes for CMH, respectively. a-c Volcano plots showing the effect size (\log_2 -ratio of SGBs' mean relative abundance values at endpoint over baseline) and significance pre-post intervention variation levels for the prevalent gut microbial SGBs in each dietary intervention cohort. **a)** BIOME cohort (ClinicalTrials.gov registration: NCT06231706) volcano plot derived from healthy adults ($n = 321$ participants with both baseline and endpoint samples) from the UK ($n = 106$ prebiotic blend, $n = 106$ probiotic, and $n = 109$ control), whose SGBs were selected to be at least 10% prevalent at both time points and significance was set to Q values < 0.01 (FDR-BH corrected P values). **b)** METHOD cohort (ClinicalTrials.gov registration: NCT05273268) volcano plot of 347 individuals from the US ($n = 177$ personalized dietary intervention program [PDP], $n = 170$ control), whose SGBs were selected to be at least 10% prevalent at both time points and significance was set to Q values < 0.1 (FDR-BH corrected P values). **c)** P3_UK23ART cohort volcano plot of 1,124 healthy individuals from the UK, whose SGBs were selected to be at least 5% prevalent at both time points, and significance was set using strict Bonferroni correction (P value $< 1.255e-05$). **d-f)** The significant SGBs in the intervention arms highlighted in panels **a**, **b**, and **c** are separated here into those that increase from those that decrease from time point zero to

endpoint, and their relative abundance values are reported on the y-axis. **Supplementary Fig. 15a-e** reports the change in relative abundance of the control arms and the change in prevalence, respectively. **d)** The change in relative abundance for significant SGBs found in the Prebiotic blend group ($n = 57$) of the BIOME cohort. **e)** The change in relative abundance for significant SGBs found in the PDP group ($n = 46$) of the METHOD cohort. **f)** The change in relative abundance for significant SGBs found in the P3_UK23ART cohort ($n = 258$). **g-i)** The ZOE MB Health ranks of the significant gut microbial SGBs highlighted in panels **a-c** if they appeared among the 661 ranked SGBs. **Supplementary Figure 15f** reports the results for the ZOE MB Diet ranks. **g)** The significant SGBs in the prebiotic blend group show a significant difference between the ZOE MB Health ranks between the two time points (Mann-Whitney U test P value = $7.78e-3$), with the increasing SGBs showing more CMH-favorable ranks (values closer to zero) and the decreasing SGBs showing more unfavorable ranks (values closer to one). No clear patterns are visible for the probiotic or control groups. **h)** The significant SGBs in the PDP group show a significant difference between the ZOE MB Health-ranks between the two time points (Mann-Whitney U test P value = $5.20e-5$), with the increasing SGBs showing more CMH-favorable ranks (values closer to zero) and the decreasing SGBs showing more unfavorable ranks (values closer to one). No clear pattern is present for the control group. **i)** The significant SGBs of the P3_UK23ART cohort show a significant difference between the ZOE MB Health ranks between the two time points (Mann-Whitney U test P value = $2.21e-11$), with the increasing SGBs showing more CMH-favorable ranks (values closer to zero) and the decreasing SGBs showing more unfavorable ranks (values closer to one).

Figure 5. Gut microbial SGBs that increase after dietary interventions are linked with more favorable ZOE MB Health and Diet rank. a) The 30 gut microbial SGBs with the greatest effect sizes as determined by meta-analysis of the 300 SGBs with a significant change after dietary interventions from Figure 4 (x-axis shows the \log_2 -ratio of SGBs' mean relative abundance values at endpoint over baseline). Visualized in the panel, the 15 most increasing and decreasing SGBs (P value < 0.2 , meta-analysis colored if Q value < 0.1) were selected across the prebiotic blend BIOME and PDP METHOD intervention arms, as well as the P3_UK23ART longitudinal cohort (cohorts are indicated by shape). All values are reported in **Supplementary Tables 25 and 26**. b) The values for the ZOE MB Health and Diet rankings for each of the 30 SGBs, if present among the 661 ranked SGBs. The majority of increasing SGBs have a more favorable rank (closer to 0), whereas the most decreasing SGBs show more unfavorable ranks (closer to 1, Mann-Whitney U test P values = 0.0154 and $6.12e-4$ for the ZOE MB Health and Diet ranks, respectively).

6. While the authors highlight the strengths of their investigation, they did not mention any weaknesses. This is unusual and I urge the authors to insert a paragraph clearly stating the weaknesses of their study!

We thank the reviewer for pointing this out and apologize for not highlighting the limitations of our study in our original version. In this revised version of our manuscript, we have now included a paragraph in the **Conclusions** in which we clarify the main limitations of our study. We report below the texts in which these are mentioned:

“We acknowledge that the demographic composition of the cohorts may influence some associations, and we are continuing to expand in both population scale and precision of each host-associated readout.”

“Many health-associated host markers are correlated as they are the nutritional indicators, and disentangling their direct interactions from those mediated by the microbiome will remain elusive until large-scale microbiome interventions will be possible in humans. In this respect, one key limitation of our study design is that it does not allow disentangling directly the effect that diet exerts on the microbiome to improve cardiometabolic health from the impact of diet only. This is particularly important as diet-based ranks were more dependent on country-related differences compared to cardiometabolic ranks, and further studies should explore food-specific links with gut microbial species and cardiometabolic outcomes in greater detail ⁹⁷, although this would entail designing large scale studies in which both introduction of single foods and alterations of specific microbiome characteristics (e.g. by administration of specific microbiome members) are tested which is currently highly problematic.”

7. The authors should consider to provide a solution such as an online tool for the research community and individuals, where individual or batched datasets with metagenomic shotgun sequencing data can be uploaded (potentially restricted to specific strains to reduce data burden) to allow researchers to compute the rank scores for their purposes

We thank the reviewer for their suggestion and hope the reviewer understands that such an implementation requires extensive computational resources (both in terms of development, time, and financial) that are out of the scope of this work, and we are not able to allocate to such a purpose. We believe that the full table of rankings we provided in **Supplementary Table 5** is enough for researchers in the microbiome field to be reused for their analyses. The table of rankings is in a tab-separated standardized format, making it readily available to be exploited in computational analyses by the scientific community. Additionally, the ranks are also available on the webpage <https://zoe.com/our-science/microbiome-ranking>, which is hosted and maintained by ZOE Ltd.

8. Please name the NCT number for the PDP (The PREDICT 3 UK 23A RT cohort). Was this study registered? Is there an ethics vote on a protocol? The protocol should be added to the supplements.

We thank the reviewer for highlighting this. The **ZOE PREDICT cohorts definition** section of the **Methods** reported the protocol numbers for the different cohorts, summarized below for clarity:

- PREDICT 1 (NCT03479866)
- PREDICT 2 (NCT03983733)
- PREDICT 3 (NCT04735835)
- METHOD (NCT05273268)
- BIOME (NCT06231706)

We want to highlight that the PREDICT 3 UK 23A RT cohort was not a randomized controlled interventional study, but we have now included the METHOD and BIOME cohorts that are registered clinical trials with intervention and control arms.

9. How was the BMI in PDP assessed? Self-report?

We thank the reviewer for their question. BMI data was calculated using the self-reported height and weight values. As requested in the point above, summary information about BMI is available in the newly added **Supplementary Table 1**.

10. For an overview it would be helpful to have a consort diagram combining the cohorts.

We thank the reviewer for this suggestion. In the revised manuscript, we have now included the CONSORT diagrams for the five PREDICT cohorts in **Supplementary Figure 1**.

This is now mentioned at the beginning of the **Results** section, with the following sentence:

*“We used the previously published ZOE PREDICT 1 cohort ¹⁴ (n = 1,098) together with four newly defined and very large-scale microbiome cohorts from the ZOE PREDICT Studies (n = 33,596, **Figure 1a**, **Supplementary Fig. 1**, **Supplementary Table 1**, **Methods**), to assemble an extensive microbiome dataset of individuals with detailed dietary records along with anthropometric and cardiometabolic measures.”*

a PREDICT 2 (US)

b PREDICT 3 US 21

c PREDICT 3 US 22A

d PREDICT 3 UK 22A

e PREDICT 3 UK 23A RT

Supplementary Fig. 1. CONSORT diagrams for the ZOE cohorts. Description of the starting number of enrolled individuals in each of the new ZOE cohorts, with the details of the exclusion criteria and the number of individuals removed at each step. **a)** The PREDICT 2 cohort

comprised a total of 975 individuals from the US (NCT03983733). **b-e)** The PREDICT 3 cohorts (ClinicalTrials.gov ID: NCT04735835) are composed of ZOE customers who consented for their microbiome data to be used for research purposes. **b)** The PREDICT 3 US21 cohort comprised a total of 11,798 individuals from the US. **c)** The PREDICT 3 US22A cohort comprised a total of 8,470 individuals from the US. **d)** The PREDICT 3 UK22A cohort comprised a total of 13,353 individuals from the UK. **e)** The PREDICT 3 UK23A RT cohort comprised a total of 1,124 individuals from the UK with a second microbiome test, for a total of 2,248 samples.

11. Rephrase the title – currently it indicates that the authors investigated how diet modulates the gut microbiome in more than 34,000 individuals. However, this is not the case for most measurements since associations were investigated and mainly not interventions (such as PREDICT 1 and PDP). Thus, that the microbiome was “modulated” by the diet in over 34,000 individuals does not seem correct here.

We thank the reviewer for their suggestion, and we apologise for the title not being specific. However, considering Nature’s guidelines for the title format (maximum of 75 characters, including spaces), we propose below a revised version of the title that is within this limit.

“Gut microbes linked with metabolic health, nutrition and diet interventions”

However, we are losing some specific characteristics of our work with this very short alternative, for example the very large-scale of our investigation, as well as, the fact that since now we include two nutritional interventions studies that are registered clinical trials, we are showing that diet interventions modulate the microbiome composition, and the microbiome changes in the “right” direction according to our novel derived SGB-level rankings.

So, if we are allowed for a slightly longer title, we propose a second alternative below, asking advice from the Reviewer and Editor, if they should have any preference.

“Gut microbiome species indicative of cardiometabolic health change by diet interventions”

12. Since this work is strongly supported by ZOE, which is clearly stated, I wonder if this should be reflected in the title even. Such as “...data from a commercial diet intervention” or anything alike. But this is up to the editors to consider.

We thank the reviewer for their comment; however, considering the space limit for the title (maximum of 75 characters, including spaces), we believe this will be very difficult to include. However, if the reviewer has a strong opinion about this and the editor strongly

encourages us to include it, we will then be happy to propose a revised version of the title to include this information.

RESPONSE TO EDITOR'S AND REVIEWERS' COMMENTS for the manuscript: "*Gut microbes linked with health, nutrition and dietary interventions*" (2024-04-07955).

We thank the editor and the reviewers for their constructive feedback on our manuscript. We carefully considered all the comments and suggestions and extensively revised the manuscript to address the raised concerns. In particular, we paid special attention to the use of the term "cardiometabolic", for which we also proposed a slight modification of the title, and we removed any reference to causality and explicitly stated the caveats and limitations of the study. We provide below a point-by-point response, and we include a track-change version comparing this revision with the previous version of our manuscript so that the Editor and the Reviewers can easily compare the extent of our revision.

Comments from the Editor

The referees with computational expertise are now satisfied with the manuscript, and while those with expertise in metabolic disease, clinical trials and nutrition remain supportive, they are requesting a number of revisions to ensure that the data/claims are presented accurately and with full transparency, and these concerns are important to address to their satisfaction. Referee #3 continues to feel that the reported AUCs in the main manuscript are potentially misleading, and feels that this should be rectified, in addition to the need for more clarity in the approach used for defining ranks. Referee #4 feels that use of the term "cardiometabolic" as a qualifier for the health markers that are associated with the microbiome is unjustified, considering that the majority of them exhibit weak associations. This referee suggests a multivariate analysis that may help to alleviate this concern, but in the absence of stronger associations, they feel that this term should be avoided entirely, including any reference to metabolic health in the title. Referee #5, who was recruited in this round to comment on the nutrition/public health implications, feels that the claims are currently overstated with respect to the associations being inferred as causal, and feels that positioning of the microbiome as the key mediator through which diet influences cardiometabolic health is insufficiently supported. As suggested, it would be critical to remove any inferences of causality, and to be explicit about the main caveats (in the Abstract, Introduction and Discussion), in particular that the microbiome alterations may in fact be a reflection of diet rather than directly associating with health outcomes. We therefore invite you to revise your manuscript taking the full set of reservations into account.

We thank the editor for summarizing the main points from the reviewers. We want to highlight here that we have now amended the **Summary**, **Introduction**, and **Conclusions** sections of the manuscript to make sure we do not hint at causality with statements from our results. In the same direction, we revised the entire manuscript to avoid using the term "cardiometabolic" as a qualifier of the health markers we considered to define our microbial species ranks. We report these updates below in the specific answers to the reviewers' points, and these edits are also available in the track change version, comparing this with the previous version of the text.

You will also need to make some editorial changes to your paper so that it is as brief as possible and complies with our Guide to Authors. We also strongly suggest that your revised manuscript has tracked changes, which is increasingly requested by referees to aid in their re-review.

We would appreciate your careful attention to the following:

STATISTICS: When revising your manuscript, you should ensure that any statistical analysis used is sound and that it conforms to Nature's guidelines. A collection of articles explaining the basics of statistical analysis and advice on how to best present it can be found here.

We checked the adherence to Nature's guidelines on statistical analysis.

LENGTH: In print, biological sciences papers do not normally exceed 8 pages on average; the final print length, however, is at the editor's discretion. The typical length of an 8-page article with 5 modest (quarter-page) display items is 4300 words. If a composite figure (with multiple panels) must occupy at least half a page in order for all the elements to be visible, the text length may need to be reduced accordingly to accommodate such figures. Essential but technical details can be moved into the Methods or Supplementary Information (see below).

In this revised version, we took the opportunity to revise the text so as to accommodate reviewers' requests. This revised version is within the suggested limits.

TITLE: Titles cannot exceed 75 characters (including spaces); they must not contain punctuation.

According also to reviewers' comments, we revised the title of our manuscript, which we now propose to be:

"Gut microbes linked with health, nutrition and dietary interventions"

This new title is within the required limits.

SUMMARY PARAGRAPH: All Nature papers begin with a fully referenced paragraph, typically no longer than 200 words, aimed at readers in other disciplines. This paragraph starts with a 2- to 3-sentence, basic introduction to the field; continues with a 1-sentence statement of the main findings starting 'Here we show' or an equivalent phrase; and finally, concludes with 2 to 3 sentences putting the main findings into general context so it is clear how the results described in the paper have moved the field forward. A downloadable, annotated example is available here. In some cases it may be necessary to exceed this limit modestly in order to explain

complex material for readers in other fields. The extra length, however, is for introduction and context, and not for additional technical information.

We updated what was previously called “Abstract” to be now the “Summary Paragraph” and made sure it is within the required specifications and limits.

MAIN TEXT: If further introductory material is necessary, the main text can begin with up to 500 words of introduction expanding on the background to the work (some overlap with the summary is acceptable), before proceeding to a concise, focused account of the findings, and ending with 1 or 2 short paragraphs of discussion. Sections are separated with subheadings (up to 40 characters including spaces) to aid navigation.

The revised **Introduction** section is within the 500-word limits and is necessary to provide the reader with the right setting to contextualize the results presented in our work.

REFERENCES: As a guideline, most papers should include no more than 50 main text references; additional references can be cited in (and listed after) the Methods section, as detailed below.

This revised version of our manuscript currently reduces the references to 67. If strictly necessary, we can try to further reduce the number of references

FIGURE LEGENDS: These should be listed sequentially after the main text references and not in the figure files; they should not exceed 300 words each. Each legend should begin with a brief title for the whole figure and continue with a short description of each panel and the symbols used. Each figure legend should contain, for each panel where relevant, the following information:

- * the exact sample size (n) for each experimental group/condition, given as a number, not a range;
- * a description of the sample collection allowing the reader to understand whether the samples represent technical or biological replicates (including how many animals, litters, cultures, etc);
- * a statement of how many times the experiment shown was replicated;
- * definitions of statistical methods and measures:
 - * very common tests (e.g., t-test, simple Chi-square tests, Wilcoxon and Mann-Whitney tests) can be identified by name only, but more complex techniques should be described in the Methods;
 - * whether tests are one-sided or two-sided;
 - * whether there are adjustments for multiple comparisons;
 - * the statistical test results (e.g., P values);
 - * the definition of ‘center values’ as median or average;

* the definition of error bars as s.d. or s.e.m.

Any descriptions too long for the figure legend should be included in the Methods section; see here for further explanation.

All figures' legends are within the 300-word limit, and all panels describe what's shown according to the provided guidelines.

METHODS: After the main text figure legends there should be a section entitled "Methods", which provides the full, step-by-step instructions that would allow other researchers to replicate the results. The Methods section will not appear in print but will appear online in the full-text HTML and PDF versions. The Methods section should be written as concisely as possible but should contain all elements necessary to allow interpretation and reproduction of the results. If there are additional references in the Methods section, their numbering should continue from the last reference in the main text, and they should be listed following the Methods section. Specialized methods that require chemical structures, figures or tables, or methods requiring equations, cannot be accommodated in the Methods section of the main text file. If such information is part of the Methods, the entire Methods section must instead be included within a Supplementary Information text file.

We don't have Methods' specific references.

ETHICS STATEMENT: For research involving human research participants, the Methods section must include an ethics statement. This statement should provide the name of the committee that approved the study; confirm that the research was performed in accordance with all relevant guidelines and regulations; and confirm that informed consent was obtained from all participants. If the study was granted an exemption from requiring ethics approval, details of the committee granting the exemption must be included.

We added the "Ethical compliance" section within the Methods. Specifically, we included the names of the Research Ethics Committees where ethical approval was secured, confirmation that all studies complied with relevant guidelines and regulations, and that informed consent was obtained by participants for all trials. Below is reported the "Ethical compliance" section:

Ethical compliance

All study protocols are registered on clinicaltrials.gov and procedures are compliant with all relevant ethical regulations. Ethical approval for the PREDICT 1 study was obtained in the United Kingdom from the King's College London Research Ethics Committee (REC) and Integrated Research Application System (IRAS 236407), and in the United States from the institutional review board (Partners Healthcare Institutional Review Board (IRB) 2018P002078). Ethical approval for the PREDICT 2 study (Pro00033432) was obtained from Advarra

IRB. Ethical approval for the PREDICT 3 study (Pro00044316, HR/DP-21/22-28300 and HR/DP-24/25-45829) was obtained from Advarra IRB and King's College London REC. Ethical approval for the METHOD study (Pro00044316; protocol no. 00044316) was obtained from Advarra IRB. Ethical approval for the BIOME study (HR/DP-23/24-39673) was obtained through King's College London REC. All participants provided written informed consent and all studies were carried out in accordance with the Declaration of Helsinki and Good Clinical Practice.

MAIN TEXT STATEMENTS: Several statements (which will not appear in print but will appear online in the full-text HTML and PDF) are required after the Methods, and before the Extended Data legends. First, there should be an Acknowledgements section, listing grant/financial support. Next, we require a detailed Author Contribution statement; the specific contributions of each author, particularly in terms of which authors performed which specific experiments, must be listed. This is followed by a Competing Interest statement. Financial or non-financial interests should be noted here, as well as any patents; patent information should include at a minimum what is covered by the patent and who submitted the patent application. Finally, an Additional Information statement should include information regarding reprints and permissions and name the author(s) to whom correspondence and requests for materials should be addressed. For details of "end note" style and an example see here.

Acknowledgements, Author Contribution, and Competing Interest statements are provided and compliant.

DATA AND CODE AVAILABILITY STATEMENTS: All original research manuscripts published in Nature Portfolio journals must include a Data availability statement (DAS). This statement must make the conditions of access to the "minimum dataset" that is necessary to interpret, verify and extend the research in the article, transparent to readers. This minimum dataset may be provided through deposition in public community/discipline-specific repositories, custom proprietary repositories for certain types of datasets, or general repositories like Figshare, Zenodo and Dryad. Providing large datasets in supplementary information is strongly discouraged and the preferred approach is to make data available in repositories. More information on Nature Portfolio's reporting standards and preparing your Data availability statement can be found here.

For all studies using custom code or mathematical algorithms that are deemed central to the conclusions, a Code availability statement (CAS) must be included, indicating whether and how the code or algorithm can be accessed, including any restrictions to access. the CAS should be provided as a separate section after the DAS but before the references. Code should be deposited in a DOI-minting repository such as Zenodo, Gigantum or Code Ocean and cited in the reference list. We encourage you to manage subsequent code versions and to use a license approved by the open source initiative. Additional details can be found here.

Both Data and Code Availability Interest statements are provided and describe the accession IDs and DOI to access the data, as well as the GitHub repository links to the code used.

Wherever possible the data used in the paper should be placed into a public data repository or presented as Supplementary Information. If data can only be shared on request, please explain why in the DAS and in the cover letter. The DAS must list which data are included (e.g. by figure panels and data types) and mention any restrictions on availability. If a dataset generated or analysed during the study is publicly available and has a unique DOI, please include it in the reference list and cite the dataset in the Methods. Accession numbers for any newly determined sequences, structures, microarray or zoobank data; project IDs for MG-RAST data; accession numbers for X-ray crystallographic coordinates and structure factor files; or comparable NMR or cryoEM data, should be included only in the DAS.

Raw metagenomic data is publicly available in ENA, and accession IDs are listed in the Data Availability statement. Microbiome taxonomic profiles and metadata information (sex, age, BMI, country) are available in Zenodo, provided in the Data Availability statement

REVIEWER #1

The authors have adequately addressed all my concerns and in my opinion this manuscript is now ready for publication. I also agree with the authors that Rev. Fig. 1 and Rev. Fig. 2 do not need to be added to the manuscript.

We thank the reviewer for their valuable comments in the previous revision round and are happy to hear that we have satisfied all their concerns.

REVIEWER #2

Thank authors for having carefully clarified their methods. I am satisfied with their response and have no further comments.

We thank the reviewer for revisiting the modified version of our manuscript, and we are happy to hear that they are satisfied with our changes.

REVIEWER #3

Peer review of revised manuscript, “Gut microbiome species indicative of cardiometabolic health are modulated by diet in large and interventional cohorts of over 34,000 individuals”

The article describes an ambitious effort where five large metagenomic ZOE PREDICT cross-sectional studies totaling over 34,000 individuals from the UK and the US were analyzed to identify microbiome species consistently associated with dietary quality or markers of cardiometabolic health. I reviewed the original submission and consider this a valuable study worthy of consideration for Nature. I feel the revised manuscript has been made even stronger. However, before it is suitable for publication, there are two deficiencies that I would like to see addressed. One deficiency has to do with a concern I raised in my original review. The second has to do with clarifying the description of the “Rankings definition” section.

We thank the reviewer for taking the time to reconsider our manuscript and are happy to have fully addressed all but two concerns in our previous revision round. We hope to have filled any gaps left with this new revision. Please find point-by-point responses to the raised questions below.

1. Please clarify “Rankings Definition” description. From the text, we read that the process begins by computing partial correlations for each SRG and marker available in a cohort. The markers are assigned to categories. However, what is not clear is how the partial correlations are ranked together and normalized to address markers with opposite associations with health (e.g. high-density lipids vs. total triglycerides). I would guess that for each cohort and each marker, the partial correlations were ordered in the appropriate direction and then assigned normalized ranks from 0 to 1, with 0 being favorable to health and 1 being unfavorable to health. Then, for each cohort and SRG, all of the normalized ranks belonging to the same category were averaged together. Then, these cohort-, SRG-, and category averages were averaged together to yield cohort- and SRG-specific averages that were then averaged to yield SRG-specific averages that became the ZOE MB Health (or Dietary) Score. That’s what Figure 2 seems to imply. However, the description in lines 739 to 746 is not clear. I believe it could be cleared up by removing some of the unnecessary mentions of “each cohort”, such as on line “740” and in the sentence on lines 744 and 745, where this phrase appears twice and seems to raise the possibility that instead of ranking and normalizing the partial correlations by cohort and marker, the ranking and normalizing combined was done by cohort and category, combining together markers, which doesn’t make sense because how then would the conflicting sort order for, say, HDL and triglycerides be accounted for? I trust the authors can develop a clearer description of their approach.

We thank the reviewer for pointing this out, and we apologize for the lack of clarity in the definition of the two ranks. We reworded the **Rankings definition** section of the **Methods**, which we hope is now clearer in describing the step-by-step process we employed. As the reviewer summarized in their comment, we first calculated the partial

Spearman's correlations (adjusted for sex, age, and BMI) between the SGBs' relative abundances with the values of the identified markers. Then, partial correlations were ordered according to whether the marker was considered as 'positive' or 'negative' with respect to host health, and ranked and normalized accordingly. At this point, the normalized ranks of each SGB-marker pair were averaged across the available markers in each specific cohort, according to the marker categories ('Personal', 'Fasting', 'Postprandial', and 'Dietary'). To define the ZOE MB Health ranks, in each cohort, we averaged out the 'Personal', 'Fasting', and 'Postprandial' categories' averages, and then computed a final average of the five PREDICT cohort averages. To define the ZOE MB Diet ranks instead, we only consider the 'Dietary' category averages and average them across the five PREDICT cohorts. We hope that the reworded **Rankings definition** section, together with the description in the main text with **Figure 2a** and **Supplementary Fig. 6**, now explain the process more clearly.

Rankings definition

We first identified a subset of prevalent SGBs to ensure a minimum number of non-zero relative abundance values. In each PREDICT cohort, we selected markers that are intermediary measures of host health or diet health, and they were organized into four categories: Personal, Dietary, Fasting, and Postprandial (Supplementary Table 2). Second, we calculated the partial Spearman's correlation between each SGB and health markers, adjusting for sex, age, and BMI, using the "pingouin" Python package (v0.5.4, <https://github.com/raphaelvallat/pingouin>) (Supplementary Figs. 4 and 7). The SGB's relative abundance values (including zeros) were used as input for the correlations. Third, the SGB-marker partial correlations were sorted ascending if the marker was considered as positive with respect to health, or descending if the marker was considered as negative. These sorted partial correlations were ranked and normalized according to cohort sample sizes into percentiles ranging from 0 to 1 (function `pandas.DataFrame.rank` with param `pct=True`) (Figure 2b,c and Supplementary Figs. 3 and 6). Fourth, for each category of markers, we computed the average percentiles across markers (Figure 2a and Supplementary Fig. 6). SGBs were retained in the overall rankings if they were ranked in at least two different cohorts, leading to a final ranking of 661 SGBs. Lastly, the "ZOE Microbiome Health Ranking 2025" was defined by first averaging the 'Personal', 'Fasting', and 'Postprandial' category percentiles within each cohort, and then averaging these cohort-specific averages. The "ZOE Microbiome Diet Ranking 2025" instead was defined by averaging the 'Dietary' percentiles across all cohorts (Figure 2a, Supplementary Fig. 6, and Supplementary Table 5). The ZOE Microbiome Rankings are also available at <https://zoe.com/our-science/microbiome-ranking>.

2. The second deficiency is more important. It has to do with the fact that the authors are retaining the AUC values that concern the pairwise comparison of the extreme quartiles, which

is a concern because those AUC values are inflated and thus subject to misinterpretation because AUCs are effect sizes. That misinterpretation needs to be corrected, as I'll explain. The authors responded to my concerns about this AUC by reporting in the Supplemental Tables two additional AUC measures that are more clinically relevant, one contrasting the highest quartile to the bottom quartiles and the other contrasting the lowest quartile to the top three quartiles. I very much appreciate the authors providing this additional information. The AUCs for these latter tasks are more clinically relevant because in an actual prediction problem, one is unlikely to know that a patient whose status needs to be predicted belongs to the middle-half and should thus be excluded from the prediction task. By including these additional AUCs in the Supplemental Table, a reader is able to see that there is a high correlation among the three different AUC measures being reported. That supports the authors in their effort to prove that there ZOE index scores are correlated with health and dietary markers. However, AUCs are used widely enough that it is fair to assume that many readers are not just interested in the fact that the AUCs show correlation (by being above 0.50) but also by how much the amount the AUC are above 0.50. In particular, an AUC of 0.80 is regarded by many readers as qualitatively different from, say, 0.70. In their interpretation of Figure 1, the authors indeed feature the AUCs exceed 0.80. AUCs that contrast the highest and lowest quartiles are biased upward. Indeed, one can see from the author's supplemental table that they are about 0.07 to 0.09 higher on average, compared to the alternative AUCs reported there. It makes sense that the clinically irrelevant AUC contrasting the top and bottom quartiles would be inflated, because they concern an "easier" task, but I doubt most readers would realize that.

Fortunately, the authors combine their reporting of the AUCs with correlation measures. The correlation measures include everyone in the sample. Also, they do not involve coarsening a continuous marker into fourths, which also leads to an inflation in the AUC compared to what would be obtained by computing the Somers D(predictor|marker) for the uncoarsened marker and transforming that Somers D to the AUC scale, a measure known as the P-K (prediction probability) [see references below

So, here's what I believe would be a more appropriate rigorous and transparent way for the authors to report their results. In Figure 1b, replace the AUC value being reported there with a clinically relevant AUC, which could be the one of the two already available in the Supplemental Table (the authors could even choose either the top fourth vs. the lowest three-fourths or the opposite on a marker-specific basis) or it could be based on the P-K version of the AUC, the one that involves computing the Somers D (predictor|marker) and then transforming it to the 0 to 1 scale by adding one to its absolute value and then multiplying that sum by 0.5

Related to this, the authors should revise the Abstract and the text in the manuscript describing "strong" associations by instead describing them as "moderate to strong" or perhaps "moderately strong". In particular, the AUCs that are more clinically relevant will be of a magnitude more in line with the correlations being reported here and one can see by looking at the standard conversions among correlation, AUCs and Cohen's D values, that the effect sizes being featured in this valuable manuscript are generally between medium and strong. [see, for example, the effect size conversion website, <https://www.escale.site/>]

We thank the reviewer for their comment. We agree with the reviewer that we should avoid potential biased interpretations that could arise from this figure, and for this

reason, we decided to visualize the AUC of the comparison between the first three quartiles vs. the fourth quartile. As the reviewer suggested, we also believe that this is a more clinically relevant AUC by comparing against the 25% highest values of the different markers. We report below the new **Fig. 1b**, from which we also removed the horizontal (Spearman's = 0.4) and vertical (AUC = 0.8) lines, to avoid biased interpretation of the results presented in the figure.

Figure 1. The ZOE PREDICT studies comprise over 34,000 healthy individuals from five cross-sectional studies from the UK and the US, and one longitudinal study from the UK, with gut microbiome samples, detailed individual information, and dietary habits. a) In this study, we considered and harmonized five cross-sectional ZOE PREDICT cohorts with individuals from the UK and the US (**Supplementary Fig. 1**). For each cohort, sample size and the percentage of female participants (% F) are reported in the upper barplots, while sequencing depth (left-most, darker color, average Gbases) and the total number of detected species (right-most, lighter color) are reported in the middle barplots, showing that cohorts with lower sequencing depths do not have fewer total number of detected species. The bottom boxplots report distributions of age (left-most, darker color) and BMI (right-most, lighter color) in the six PREDICT cohorts (the PREDICT 1, P1, cohort is split into its UK and US parts, but is considered as a single cohort). **b)** Random Forest classification (discriminating the first three quartiles against the fourth quartile) and regression machine learning models (**Methods**) trained on the whole microbiome SGB-level composition with a cross-validation (CV) approach, show moderately strong and consistent associations with different categories of clinical data available across the five cross-sectional ZOE PREDICT cohorts (full ML results are reported in **Supplementary Fig. 2** and **Supplementary Tables 2** and **3**).

REFERENCES

Smith, WD, et al. "A MEASURE OF ASSOCIATION FOR ASSESSING PREDICTION ACCURACY THAT IS A GENERALIZATION OF NON-PARAMETRIC ROC AREA." *Statistics in Medicine.*, vol. 15, no. 11, 1996, pp. 1199–215, [https://doi.org/10.1002/\(SICI\)1097-0258\(19960615\)15:11<1199::AID-SIM218>3.0.CO;2-Y](https://doi.org/10.1002/(SICI)1097-0258(19960615)15:11<1199::AID-SIM218>3.0.CO;2-Y))

Van Calster, B., Van Belle, V., Vergouwe, Y. and Steyerberg, E.W. (2012), Discrimination ability of prediction models for ordinal outcomes: Relationships between existing measures and a new measure. *Biom. J.*, 54: 674-685. <https://doi.org/10.1002/bimj.201200026>

REVIEWER #4

I want to applaud the authors for the comprehensive revision according to the reviewers' comments.

Here are my statements according to the answers to my previous comments:

Many thanks for the positive feedback and kind words. We hope to have addressed the remaining comments with this revised version of our manuscript. The point-by-point responses are reported below.

Comment 1: no further questions

Comment 2: I find it not convincing to base the expression “metabolic health” or “cardiometabolic” solely on Spearman Correlations that are above the threshold of 0.3 of only age, weight, BMI and potentially visceral fat (which has not been cross-validated if I see this correctly), while all other parameters that have been established to assess cardiometabolic health are below the threshold (referring to Suppl. Fig. 2 and new Suppl. Table 1). One could also run a multivariate model and investigate individual parameter estimates. It is matter of current debate in the field and parameters of defining unhealthy obesity (i.e. cardiometabolic risk) have been identified (PMID: 39448862). This current debate is important to overcome rather traditional views that markers of obesity are generally markers of metabolic health and a panel of leading experts in the field has recently addressed this issue (PMID: 39824205). Thus, unless I oversee something here, I urge the authors to change the term cardiometabolic or metabolic health to e.g. anthropometry or anthropometry characteristics. Otherwise please convince me that the associations below the Spearman cut-off of 0.3 reflect clinically relevant associations to use the term cardiometabolic or metabolic health.

Comment 11: I find the title “Gut microbes linked with metabolic health, nutrition and diet interventions” much more adequate. However, as stated in my Comment 2 above, I am not convinced that the associations shown reflect associations with metabolic health. It rather reflects associations with adiposity or body mass index, since all other parameters reflecting the metabolic state such as blood pressure or HbA1c etc. seemed to exhibit only minor associations if at all. Thus I would exchange the term metabolic health with “body mass index”.

We thank the reviewer for their comments, both of which we would like to address together as their about the same argument on the use of the term “cardiometabolic” or “metabolic health”. We believe that the reviewer’s comment about the correlation strength originates from **Supplementary Fig. 2**, about which there seems to be some misunderstanding. **Supplementary Fig. 2** only shows the performance of the Random Forest regressor trained on the SGBs' relative abundances to predict the different clinical markers in different cohorts. Those correlations were *not* the ones used to define the two ranks, and age, weight, and BMI presented in **Supplementary Fig. 2** were *not*

considered to define the two SGB ranks. Instead, we used BMI, together with sex and age, to correct the correlations between the SGBs' relative abundances and the different clinical markers. Therefore, the ranks were derived from the "cardiometabolic markers" alone and not from BMI. For this reason, we find the suggestion to include "body mass index" in the title to not adequately reflect our work.

The markers we considered to define the ZOE MB Health ranks encompass several that are commonly used when assessing cardiometabolic health. We used, among others, those that were represented in all five cohorts: total cholesterol, HDL, HbA1C, triglycerides, glucose, glucose responses to three standardized metabolic challenge meals (iAUCs), and blood pressure, available in four out of five cohorts. Most of these markers were measured both at fasting and postprandially. These are detailed in **Figure 2** and **Supplementary Figs. 3** and **4**. The ranks' definition then was based on the coherent associations between all these markers (and *not* BMI) and the SGBs' relative abundance values.

However, we agree that the microbiome is not the only player linked to cardiometabolic health and that some of the associations are not particularly strong. As also pointed out by Reviewer #5, diet plays a major role in the levels of the markers that are used to define cardiometabolic health. Consequently, it is not surprising that the variation explainable by these markers is limited. Despite this, we observed similar correlation patterns across many clinical markers and different cohorts comprising individuals from different countries (**Supplementary Fig. 4**), and this is what we leveraged to define our two new microbial species rankings.

Finally, we agree with the reviewer that the term 'cardiometabolic' should be used wisely, and we amended the title and text to reflect this. As explained just above, we feel that mentioning "body mass index" in the title doesn't reflect the nature of our ranks, as we actually adjusted for BMI (together with sex and age). So, we propose the new title of our manuscript to be:

"Gut microbes linked with health, nutrition and dietary interventions"

Moreover, we specify now more clearly in the text that some of the markers we are considering are intermediary or surrogate measures of cardiometabolic health, and whenever possible, we specify that the clinical markers are representative of host health or of anthropometric characteristics. We kept in the **Introduction** the mention of the burden of cardiometabolic health and the fact that the microbiome and diet could play a role.

These changes are applied widely to the whole text, so we kindly ask the reviewer to check them in the track-changed version of the manuscript.

Comment 3: no further questions

Comment 4: E.g. in Suppl. Fig. 2 I would refer to the data on the x-axis as clinical data rather than metadata. But this is of course minor.

We thank the reviewer for their suggestions. We amended the caption of **Supplementary Fig. 2** as suggested, which we reported below for convenience.

Supplementary Fig. 2. Microbiome predictive potential for personal information, dietary indices, fasting, and postprandial metabolic markers, via classification and regression random forest models. Distributions of the random forest median AUCs (a) and median Spearman’s correlation coefficients (b) (**Methods**) in the five cross-sectional PREDICT studies for the different clinical data divided into four categories: ‘Personal’, ‘Dietary’, ‘Fasting’, and ‘Postprandial’. The AUC and Spearman’s index thresholds of 0.7 and 0.3, respectively, are indicated with a dashed line. **a)** Each point represents the median AUC value obtained in cross-validation for each cohort when testing the corresponding clinical marker on the x-axis. **b)** Each point represents the median Spearman’s correlation coefficient for the predicted values by the regressor and the true values in the cross-validation setting for each cohort.

Comment 5: no further comment

Comment 6: no further comment

Comment 7: no further comment

Comment 8: Please provide the study protocols or at least a link to the source.

We thank the reviewer for their comment. All studies successfully secured ethical approval following review of the study materials, including protocols. Study protocols for all studies are available on clinicaltrials.gov, where the studies were registered. We have now included in the manuscript associated ethical approval numbers, the clinical trial numbers, and the links to the protocol/clinical trial page for every study. These additions will make information more accessible and clearer to the readership and are reported below.

ZOE PREDICT cohorts definition

[..] The PREDICT 1 cohort (NCT03479866) was previously described^{8,50}. [..] The PREDICT 2 study (NCT03983733) [..] The PREDICT 3 cohorts (US21, US22A, UK22A) are research cohorts (NCT04735835) [..] Additionally, we considered and analyzed two registered clinical nutritional intervention studies, namely METHOD (NCT05273268)⁷¹ and BIOME (NCT06231706)⁷² [..] All study protocols are registered and available on clinicaltrials.gov through the clinical trials number and link affiliated with each trial.

Ethical compliance

All study protocols are registered on clinicaltrials.gov and procedures are compliant with all relevant ethical regulations. Ethical approval for the PREDICT 1 study was obtained in the United Kingdom from the King's College London Research Ethics Committee (REC) and Integrated Research Application System (IRAS 236407), and in the United States from the institutional review board (Partners Healthcare Institutional Review Board (IRB) 2018P002078). Ethical approval for the PREDICT 2 study (Pro00033432) was obtained from Advarra IRB. Ethical approval for the PREDICT 3 study (Pro00044316, HR/DP-21/22-28300 and HR/DP-24/25-45829) was obtained from Advarra IRB and King's College London REC. Ethical approval for the METHOD study (Pro00044316; protocol no. 00044316) was obtained from Advarra IRB. Ethical approval for the BIOME study (HR/DP-23/24-39673) was obtained through King's College London REC. All participants provided written informed consent and all

studies were carried out in accordance with the Declaration of Helsinki and Good Clinical Practice.

Comment 9: no further comment

Comment 10: no further comment

Comment 12: I have no strong opinion about this but rather believe this to be an editorial decision.

Considering the length requirements of the Journal for the title, we believe it will be difficult to fit it in. However, if the editor thinks this is appropriate and allows us, we will be happy to do so.

REVIEWER #5

The manuscript is well-written and of large scientific interest. It presents a rich dataset and offers a clear and comprehensive overview of the cross-sectional associations between diet, the gut microbiome, and cardiometabolic health. From a scientific standpoint, it makes a valuable contribution to the growing body of literature exploring the role of the microbiome in human health.

Findings from this study could be relevant for public health strategies to lower the burden of cardiometabolic diseases. However, I do have some concerns regarding the cross-sectional design and the strength of the conclusions drawn from the findings. There is a risk that the public health implications of the study are overstated, particularly the suggestion that targeting the microbiome may be an effective strategy to prevent or reduce cardiometabolic disease. These points are outlined in more detail below.

Public health relevance

The observed associations, if causal, are potentially important from both a nutritional science and public health perspective. By identifying specific gut microbial species linked to dietary patterns and health outcomes, the authors highlight the microbiome as a key mediator in the diet-disease relationship. The "ZOE Microbiome Ranking 2025" offers a novel framework for ranking microbes by cardiometabolic risk and may inform personalized nutritional interventions to improve metabolic health. The study also shows that dietary changes can beneficially alter gut microbiota composition, underscoring its modifiability through nutrition. These findings reinforce the role of diet in preventing cardiometabolic diseases such as cardiovascular disease and type 2 diabetes. The large sample size (>34,000 individuals) adds robustness, and the study supports the potential of precision nutrition based on individual microbiome profiles for effective disease prevention and health promotion. However, the cross-sectional design limits causal inference, reducing the applicability of the findings to dietary guideline development. The latter is usually based on stronger designs (prospective cohort studies and RCTs), because causality is key for effective public health strategies.

We thank the reviewer for their positive comments on our work. We agree with them that the cross-sectional study design can be a limitation, and in this revised version of our manuscript, we clearly state this in the **Introduction** and **Conclusions**. We would like to highlight that we sought to overcome the limitations of the cross-sectional design by validating our ranks both in publicly available microbiome cohorts, spanning more diverse microbiome configurations, countries, and lifestyles (despite the cross-sectional study designs), as well as in two dietary intervention studies (with longitudinal microbiome samplings). Our analysis on over 7,800 independent public samples showed a strong and reproducible association, where favorably-ranked species were negatively correlated with BMI and diseases, and unfavorably-ranked species were positively correlated with BMI and linked with disease. This consistency across a large, geographically diverse set of public cohorts supports the generalizability of our findings. As the reviewer noted, we also validated our rankings in two registered dietary

intervention clinical trials with a longitudinal study design. Results showed that dietary changes aiming to improve health, such as a personalized dietary program or a prebiotic blend, led to a significant and substantial increase in health-associated microbiome species and a reduction or depletion of unfavorably ranked species. These further support that the proposed ZOE MB Health and Diet ranks defined on cross-sectional cohorts were also validated in independent dietary intervention longitudinal studies.

In addition, as we stated in the **Conclusions**, it is currently not possible to disentangle the direct impact of diet and the microbiome on health markers, as many of the considered health markers are intrinsically co-correlating, likely each one impacting the other. So, the 'causality', as the reviewer was pointing to, would be extremely difficult to prove, both from the diet and microbiome perspectives. Nonetheless, we believe our study provides a foundational step in this direction by identifying specific microbial species that can be targeted, which can inform future research using more robust designs.

We hope that these clarifications, together with the revised text and the additional details from our validation efforts, address the reviewer's concerns and highlight the strength of our approach.

Role of gut microbiome

A key concern is that the authors position the gut microbiome as the primary mediator through which diet influences cardiometabolic health, while insufficiently acknowledging the independent effects of diet itself. Although they recognize the cross-sectional design and the potential for reverse causation, they pay limited attention to the possibility that the microbiome largely reflects dietary patterns rather than causally driving health outcomes. Evidence from intervention studies demonstrating that direct, long-term modification of the microbiome improves cardiometabolic health remains scarce. Moreover, global variation in microbiome composition contrasts with the widespread and consistent rise in obesity and cardiometabolic diseases across diverse populations, suggesting that Westernized diets and lifestyles, rather than the microbiome per se, are the main drivers. Further investigation into factors that disrupt the microbiome, such as antibiotic use, and their long-term cardiometabolic impacts could be valuable (although such studies are inherently challenged by confounding by indication). Overall, the article risks overstating the potential of microbiome-targeted interventions at the individual level, which may divert focus from broader public health strategies, including food reformulation and modifications to the food environment.

In my view, the authors give insufficient attention to these aspects, and I recommend they address this more thoroughly in the Discussion. They may elaborate on this statement in lines 578-580: "One key limitation of our study design is that it does not allow disentangling directly the effect that diet exerts on the microbiome to improve cardiometabolic health from the impact of diet only". They would do well to properly frame this in the Introduction, setting appropriate expectations for the reader.

We thank the reviewer for their comment. We agree that it is important not to overstate the microbiome's role as a primary mediator, and to properly frame the limitations of our study design. In our revised version of our manuscript, we now mention more clearly that it is currently not possible to disentangle the role of diet and microbiome in cardiometabolic health.

We believe this revision addresses your concerns by providing a more balanced perspective on the roles of diet and the microbiome, as we now anticipate this limitation already in both the **Summary** and **Introduction**, and throughout the manuscript (text snippets are reported below). We would like to clarify that with our ranks, we were not hinting at causal implications between the microbiome and cardiometabolic health from our cross-sectional findings. To address the lack of evidence from intervention studies, as the reviewer commented, we showed with the two registered dietary intervention clinical trials with a longitudinal study design, that dietary interventions shape microbiome configuration in a direction consistent with improved health outcomes. This supports the ranks defined cross-sectionally to be predictive of changes in targeted intervention studies. Diet, as well as Westernization, which the reviewer also pointed out, indeed have a huge impact on microbial composition and host health, but failing to recognize the role the microbiome plays in human health is also an oversight, as diet, or lifestyle, alone are not capable of fully answering, either. We specifically acknowledge the Westernization configuration of our cohorts as a limitation in the **Conclusions**:

Summary: *“In conclusion, we characterized gut microbial species based on their association with both human health markers and diet quality, and while the causal diet-microbiome-health links need validation, the resulting summary ranking can be leveraged as an indicator of species’ associations with gut microbiome health.”*

Introduction:

“Habitual diet is not only among the known risk factors associated with CMDs, but it is also the primary modifiable target for prevention and treatment⁴.”

“These species were organized into two microbiome rankings, representing host health and diet quality, respectively, that can be the basis for future causal and mechanistic studies.”

Conclusions: *“Second, the geographic diversity spanning all US states and UK regions, while confined to typical Westernized lifestyles and diets, allowed us to overcome local lifestyle-associated microbiome configurations.”*

While in our study, we primarily focus on the microbiome, we agree with the reviewer that it is important to acknowledge the broader public health context. This revision frames our work as complementary, rather than a replacement for broader public health

strategies. Instances in which we better highlight the interplay between diet, microbiome, and health:

“However, individual responses to dietary interventions vary, and precision nutrition aims at identifying host-specific factors that modulate the interaction between diet and host health ¹⁶, but it is currently not possible to disentangle the effects diet plays to improve cardiometabolic health via the microbiome.”

“A change in dietary patterns can shift the species-level composition of the microbiome, with knock-on effects on host health ¹⁵.”

“[.] indicating that precise dietary interventions aimed at stimulating beneficial bacterial growth can contribute to treating or managing metabolic disorders symptoms.”

“Together, these results show how dietary interventions or tailored prebiotic blends, both aiming at improving diet quality, positively modulate the microbiome composition.”

RESPONSE TO EDITOR'S AND REVIEWERS' COMMENTS for the manuscript: "*Gut microbes associated with health, nutrition and dietary interventions*" (2024-04-07955).

We thank the editor and the reviewers for their positive assessment of our manuscript. We thank all reviewers for their constructive comments during the peer review process; they made it possible to improve our manuscript, which they now find ready for publication.

Referee #3

The authors have adequately addressed all my analysis and reporting concerns. In my opinion this manuscript is now ready for publication.

We thank the reviewer for finding our latest version to have fulfilled all the raised points and be now ready for publication.

Referee #4

I have no further comments and recommend publication.

We thank the reviewer for finding our latest version to have fulfilled all the raised points and recommending it for publication.

Referee #5

I am satisfied with the given answers and the adjustments in the text. I have no further comments.

We thank the reviewer for finding our latest version to have fulfilled all the raised points.

Manuscript #2024-04-07955C: Gut microbes linked with metabolic health, nutrition and diet interventions.

Referee: Johanna M. Geleijnse, PhD. Wageningen University, The Netherlands.

Referee report

The manuscript is well-written and of large scientific interest. It presents a rich dataset and offers a clear and comprehensive overview of the cross-sectional associations between diet, the gut microbiome, and cardiometabolic health. From a scientific standpoint, it makes a valuable contribution to the growing body of literature exploring the role of the microbiome in human health.

Findings from this study could be relevant for public health strategies to lower the burden of cardiometabolic diseases. However, I do have some concerns regarding the cross-sectional design and the strength of the conclusions drawn from the findings. There is a risk that the public health implications of the study are overstated, particularly the suggestion that targeting the microbiome may be an effective strategy to prevent or reduce cardiometabolic disease. These points are outlined in more detail below.

Public health relevance

The observed associations, if causal, are potentially important from both a nutritional science and public health perspective. By identifying specific gut microbial species linked to dietary patterns and health outcomes, the authors highlight the microbiome as a key mediator in the diet-disease relationship. The "ZOE Microbiome Ranking 2025" offers a novel framework for ranking microbes by cardiometabolic risk and may inform personalized nutritional interventions to improve metabolic health. The study also shows that dietary changes can beneficially alter gut microbiota composition, underscoring its modifiability through nutrition. These findings reinforce the role of diet in preventing cardiometabolic diseases such as cardiovascular disease and type 2 diabetes. The large sample size (>34,000 individuals) adds robustness, and the study supports the potential of precision nutrition based on individual microbiome profiles for effective disease prevention and health promotion. However, the cross-sectional design limits causal inference, reducing the applicability of the findings to dietary guideline development. The latter is usually based on stronger designs (prospective cohort studies and RCTs), because causality is key for effective public health strategies.

Role of gut microbiome

A key concern is that the authors position the gut microbiome as the primary mediator through which diet influences cardiometabolic health, while insufficiently acknowledging the independent effects of diet itself. Although they recognize the cross-sectional design and the potential for reverse causation, they pay limited attention to the possibility that the microbiome largely reflects dietary patterns rather than causally driving health outcomes. Evidence from intervention studies demonstrating that direct, long-term modification of the microbiome improves cardiometabolic health remains scarce. Moreover, global variation in microbiome composition contrasts with the widespread and consistent rise in obesity and cardiometabolic diseases across diverse populations, suggesting that Westernized diets and lifestyles, rather than the microbiome per se, are the main drivers. Further investigation into factors that disrupt the microbiome, such as antibiotic use, and their long-term cardiometabolic impacts could be valuable (although such studies are inherently challenged by confounding by indication). Overall, the article risks overstating the potential of microbiome-targeted interventions at the individual level, which may divert focus from broader public health strategies, including food reformulation and modifications to the food environment.

In my view, the authors give insufficient attention to these aspects, and I recommend they address this more thoroughly in the Discussion. They may elaborate on this statement in lines 578-580: "One key limitation of our study design is that it does not allow disentangling directly the effect that diet exerts on the microbiome to improve cardiometabolic health from the impact of diet only". They would do well to properly frame this in the Introduction, setting appropriate expectations for the reader.